# Provably Efficient Risk-Sensitive Reinforcement Learning: Iterated CVaR and Worst Path

**Yihan Du**
Institute for Interdisciplinary Information Sciences
Tsinghua University
Beijing, China
duyh18@mails.tsinghua.edu.cn

**Siwei Wang**
Microsoft Research
Beijing, China
siweiwang@microsoft.com

**Longbo Huang** *
Institute for Interdisciplinary Information Sciences
Tsinghua University
Beijing, China
longbohuang@tsinghua.edu.cn

## Abstract

In this paper, we study a novel episodic risk-sensitive Reinforcement Learning (RL) problem, named Iterated CVaR RL, which aims to maximize the tail of the reward-to-go at each step, and focuses on tightly controlling the risk of getting into catastrophic situations at each stage. This formulation is applicable to real-world tasks that demand strong risk avoidance throughout the decision process, such as autonomous driving, clinical treatment planning and robotics. We investigate two performance metrics under Iterated CVaR RL, i.e., Regret Minimization and Best Policy Identification. For both metrics, we design efficient algorithms ICVaR-RM and ICVaR-BPI, respectively, and provide nearly matching upper and lower bounds with respect to the number of episodes $K$. We also investigate an interesting limiting case of Iterated CVaR RL, called Worst Path RL, where the objective becomes to maximize the minimum possible cumulative reward. For Worst Path RL, we propose an efficient algorithm with constant upper and lower bounds. Finally, our techniques for bounding the change of CVaR due to the value function shift and decomposing the regret via a distorted visitation distribution are novel, and can find applications in other risk-sensitive RL problems.

## 1 Introduction

Reinforcement Learning (RL) (Kaelbling et al., 1996; Szepesvári, 2010; Sutton & Barto, 2018) is a classic online decision-making formulation, where an agent interacts with an unknown environment with the goal of maximizing the obtained reward. Despite the empirical success and theoretical progress of recent RL algorithms, e.g., (Szepesvári, 2010; Agrawal & Jia, 2017; Azar et al., 2017; Zanette & Brunskill, 2019), they focus mainly on the risk-neutral criterion, i.e., maximizing the expected cumulative reward, and can fail to avoid rare but disastrous situations. As a result, existing algorithms cannot be applied to tackle real-world risk-sensitive tasks, such as autonomous driving (Wen et al., 2020) and clinical treatment planning (Coronato et al., 2020), where policies that ensure low risk of getting into catastrophic situations at all decision stages are strongly preferred.

Motivated by the above facts, we investigate Iterated CVaR RL, a novel episodic RL formulation equipped with an important risk-sensitive criterion, i.e., Iterated Conditional Value-at-Risk (CVaR) (Hardy & Wirch, 2004). Here, CVaR (Artzner et al., 1999) is a popular static (single-stage) risk measure which stands for the expected tail reward. Iterated CVaR is a dynamic (multi-stage) risk measure defined upon CVaR by backward iteration, and focuses on the worst portion of the reward-to-go at each stage. In the Iterated CVaR RL problem, an agent interacts with an *unknown*

---

* Corresponding author.

episodic Markov Decision Process (MDP) in order to maximize the worst $\alpha$-portion of the reward-to-go at each step, where $\alpha \in (0, 1]$ is a given risk level. Under this model, we investigate two important performance metrics, i.e., Regret Minimization (RM), where the goal is to minimize the cumulative regret over all episodes, and Best Policy Identification (BPI), where the performance is measured by the number of episodes required for identifying an optimal policy.

Compared to existing CVaR MDP model, e.g., (Boda & Filar, 2006; Ott, 2010; Bäuerle & Ott, 2011; Chow et al., 2015), which aims to maximize the CVaR (i.e., the worst $\alpha$-portion) of the *total* reward, our Iterated CVaR RL concerns the worst $\alpha$-portion of the reward-to-go *at each step*, and prevents the agent from getting into catastrophic states more carefully. Intuitively, CVaR MDP takes more cumulative reward into account and prefers actions which have better performance in general, but can have larger probabilities of getting into catastrophic states. Thus, CVaR MDP is suitable for scenarios where bad situations lead to a higher cost instead of fatal damage, e.g., finance. In contrast, our Iterated CVaR RL prefers actions which have smaller probabilities of getting into catastrophic states. Hence, Iterated CVaR RL is suitable for *safety-critical* applications, where catastrophic states are unacceptable and need to be carefully avoided, e.g., clinical treatment planning (Wang et al., 2019) and unmanned helicopter control (Johnson & Kannan, 2002). For example, consider the case where we fly an unmanned helicopter to complete some task. There is a small probability that, at each time during execution, the helicopter encounters a sensing or control failure and does not take the scheduled action. To guarantee the safety of surrounding workers and the helicopter, we need to make sure that even if the failure occurs, the taken policy ensures that the helicopter does not crash and cause fatal damage (see Appendix C.2, C.3 for more detailed comparisons with existing risk-sensitive MDP models).

Iterated CVaR RL faces several unique challenges as follows. (i) The importance (contribution to regret) of a state in Iterated CVaR RL is not proportional to its visitation probability. Specifically, there can be states which are critical (risky) but have a small visitation probability. As a result, the regret for Iterated CVaR RL cannot be decomposed into the estimation error at each step with respect to the visitation distribution, as in standard RL analysis (Jaksch et al., 2010; Azar et al., 2017; Zanette & Brunskill, 2019). (ii) In Iterated CVaR RL, the calculation of estimation error involves bounding the change of CVaR when the true value function shifts to optimistic value function, which is very different from typically bounding the change of expected rewards as in existing RL analysis (Jaksch et al., 2010; Azar et al., 2017; Jin et al., 2018). Therefore, Iterated CVaR RL demands brand-new algorithm design and analytical techniques. To tackle the above challenges, we design two efficient algorithms `ICVaR-RM` and `ICVaR-BPI` for the RM and BPI metrics, respectively, equipped with delicate CVaR-adapted value iteration and exploration bonuses to allocate more attention on rare but potentially dangerous states. We also develop novel analytical techniques, for bounding the change of CVaR due to the value function shift and decomposing the regret via a distorted visitation distribution. Lower bounds for both metrics are established to demonstrate the optimality of our algorithms with respect to the number of episodes $K$. Moreover, we present experiments to validate our theoretical results and show the performance superiority of our algorithm (see Appendix A).

We further study an interesting limiting case of Iterated CVaR RL when $\alpha$ approaches $0$, called Worst Path RL, where the goal becomes to maximize the minimum possible cumulative reward (optimize the worst path). This setting corresponds to the scenario where the decision maker is extremely risk-adverse and concerns the worst situation (e.g., in clinical treatment planning (Coronato et al., 2020), the worst case can be disastrous). We emphasize that Worst Path RL cannot be directly solved by taking $\alpha \to 0$ in Iterated CVaR RL's results, as the results there have a dependency on $\frac{1}{\alpha}$ in both upper and lower bounds. To handle this limiting case, we design a simple yet efficient algorithm `MaxWP`, and obtain constant upper and lower regret bounds which are independent of $K$.

The contributions of this paper are summarized as follows.

- We propose a novel Iterated CVaR RL formulation, where an agent interacts with an unknown environment, with the objective of maximizing the worst $\alpha$-percent tail of the reward-to-go at each step. This formulation enables one to tightly control risk throughout the decision process, and is most suitable for applications where such safety-at-all-time is critical.

- We investigate two important metrics of Iterated CVaR RL, i.e., Regret Minimization (RM) and Best Policy Identification (BPI), and propose efficient algorithms `ICVaR-RM`

and `ICVaR-BPI`. We establish nearly matching regret/sample complexity upper and lower bounds with respect to $K$. Moreover, we develop novel techniques to bound the change of CVaR due to the value function shift and decompose the regret via a distorted visitation distribution, which can be applied to other risk-sensitive decision making problems.

- We further investigate a limiting case of Iterated CVaR RL when $\alpha$ approaches $0$, called Worst Path RL, where the objective is to maximize the minimum possible cumulative reward. We develop a simple and efficient algorithm `MaxWP`, and provide constant regret upper and lower bounds (independent of $K$).

Due to space limit, we defer all proofs and experiments to Appendix.

## 2 RELATED WORK

Below we review the most related works, and defer a full literature review to Appendix B.

**CVaR-based MDPs (Known Transition).** Boda & Filar (2006); Ott (2010); Bäuerle & Ott (2011); Chow et al. (2015) study the CVaR MDP where the objective is to minimize the CVaR of the total cost, and show that the optimal policy for CVaR MDP is history-dependent (see Appendix C.2 for a detailed comparison with CVaR MDP). Hardy & Wirch (2004) firstly define the Iterated CVaR measure, and Osogami (2012); Chu & Zhang (2014); Bäuerle & Glauner (2022) consider iterated coherent risk measures (including Iterated CVaR) in MDPs, and demonstrate the existence of Markovian optimal policies. The above works focus mainly on the planning side, i.e., proposing algorithms and error guarantees for MDPs with *known* transition, while our work develops RL algorithms (interacting with the environment) and regret/sample complexity results for *unknown* transition.

**Risk-sensitive Reinforcement Learning (Unknown Transition).** Tamar et al. (2015); Keramati et al. (2020) study CVaR MDP with unknown transition and provide convergence analysis. Borkar & Jain (2014); Chow & Ghavamzadeh (2014); Chow et al. (2017) investigate RL with CVaR-based constraints. Heger (1994); Coraluppi & Marcus (1997; 1999) consider minimizing the worst-case cost in RL and design heuristic algorithms. Fei et al. (2020; 2021a;b) study risk-sensitive RL with the exponential utility criterion, which takes all successor states into account with an exponential reweighting scheme. In contrast, our Iterated CVaR RL primarily concerns the worst $\alpha$-portion successor states, and focuses on optimizing the performance under bad situations (see Appendix C.3 for a detailed comparison).

## 3 PROBLEM FORMULATION

In this section, we present the problem formulations of Iterated CVaR RL and Worst Path RL.

**Conditional Value-at-Risk (CVaR).** We first introduce two risk measures, i.e., Value-at-Risk (VaR) and Conditional Value-at-Risk (CVaR). Let $X$ be a random variable with cumulative distribution function $F(x) = \Pr[X \leq x]$. Given a risk level $\alpha \in (0, 1]$, the VaR at risk level $\alpha$ is the $\alpha$-quantile of $X$, i.e., $\text{VaR}^\alpha(X) = \min\{x | F(x) \geq \alpha\}$, and the CVaR at risk level $\alpha$ is defined as (Rockafellar et al., 2000):

$$\text{CVaR}^\alpha(X) = \sup_{x \in \mathbb{R}} \left\{ x - \frac{1}{\alpha} \mathbb{E}\left[ (x - X)^+ \right] \right\},$$

where $(x)^+ := \max\{x, 0\}$. If there is no probability atom at $\text{VaR}^\alpha(X)$, CVaR can also be written as $\text{CVaR}^\alpha(X) = \mathbb{E}[X | X \leq \text{VaR}^\alpha(X)]$ (Shapiro et al., 2021). Intuitively, $\text{CVaR}^\alpha(X)$ is a distorted expectation of $X$ conditioning on its $\alpha$-portion tail, which depicts the average value when bad situations happen. When $\alpha = 1$, $\text{CVaR}^\alpha(X) = \mathbb{E}[X]$, and when $\alpha \to 0$, $\text{CVaR}^\alpha(X)$ tends to $\min(X)$ (Chow et al., 2015).

**Iterated CVaR RL.** We consider an episodic Markov Decision Process (MDP) $\mathcal{M}(\mathcal{S}, \mathcal{A}, H, p, r)$. Here $\mathcal{S}$ is the state space, $\mathcal{A}$ is the action space, and $H$ is the length of horizon in each episode. $p$ is the transition distribution, i.e., $p(s'|s, a)$ gives the probability of transitioning to $s'$ when starting from state $s$ and taking action $a$. $r : \mathcal{S} \times \mathcal{A} \mapsto [0, 1]$ is a reward function, and $r(s, a)$ gives a deterministic reward for taking action $a$ in state $s$. A policy $\pi$ is defined as a collection of $H$ functions, i.e., $\pi = \{\pi_h : \mathcal{S} \mapsto \mathcal{A}\}_{h \in [H]}$, where $[H] := \{1, 2, ..., H\}$.

The *episodic* RL game is as follows. In each episode $k$, an agent chooses a policy $\pi^k$, and starts from a fixed initial state $s_1$, i.e., $s_1^k := s_1$, as assumed in many prior RL works (Fiechter, 1994; Kaufmann et al., 2021; Ménard et al., 2021). At each step $h \in [H]$, the agent observes the state $s_h^k$ and takes an action $a_h^k = \pi_h^k(s_h^k)$. After that, it receives a reward $r(s_h^k, a_h^k)$ and transitions to a next state $s_{h+1}^k$ according to the transition distribution $p(\cdot|s_h^k, a_h^k)$. The episode ends after $H$ steps and the agent enters the next episode.

In Iterated CVaR RL, for any risk level $\alpha \in (0, 1]$ and a policy $\pi$, we use value function $V_h^{\alpha,\pi} : \mathcal{S} \mapsto \mathbb{R}$ and Q-value function $Q_h^{\alpha,\pi} : \mathcal{S} \times \mathcal{A} \mapsto \mathbb{R}$ to denote the cumulative reward that can be obtained when the agent transitions to the worst $\alpha$-portion states at each step, starting from $s$ and $(s, a)$ at step $h$, respectively. For simplicity of notation, when the value of $\alpha$ is clear, we omit the superscript $\alpha$ and use the notations $V_h^\pi$ and $Q_h^\pi$. Formally, $Q_h^\pi$ and $V_h^\pi$ are recurrently defined in Eq. (i) below. Since $\mathcal{S}$, $\mathcal{A}$ and $H$ are finite and the maximization of $V_h^\pi(s)$ in Iterated CVaR RL satisfies the optimal substructure property, there exists an optimal policy $\pi^*$ which gives the optimal value $V_h^*(s) = \max_\pi V_h^\pi(s)$ for all $s \in \mathcal{S}$ and $h \in [H]$ (Chu & Zhang, 2014). Therefore, the Bellman equation and the Bellman optimality equation are given in Eqs. (i),(ii) below, respectively (Chu & Zhang, 2014).

$$
\begin{cases}
Q_h^\pi(s,a) = r(s,a) + \text{CVaR}_{s'\sim p(\cdot|s,a)}^\alpha(V_{h+1}^\pi(s')) \\
V_h^\pi(s) = Q_h^\pi(s, \pi_h(s)) \\
V_{H+1}^\pi(s) = 0, \ \forall s \in \mathcal{S},
\end{cases} \text{(i)}
\quad
\begin{cases}
Q_h^*(s,a) = r(s,a) + \text{CVaR}_{s'\sim p(\cdot|s,a)}^\alpha(V_{h+1}^*(s')) \\
V_h^*(s) = \max_{a\in\mathcal{A}} Q_h^*(s, a) \\
V_{H+1}^*(s) = 0, \ \forall s \in \mathcal{S},
\end{cases} \text{(ii)}
$$

where $\text{CVaR}_{s'\sim p(\cdot|s,a)}^\alpha(V_{h+1}^\pi(s'))$ denotes the CVaR value of random variable $V_{h+1}^\pi(s')$ with $s' \sim p(\cdot|s, a)$ at risk level $\alpha$. We also provide the expanded version of value function definitions for Iterated CVaR RL (Eqs. (i), (ii)) in Appendix C.1.

We consider two performance metrics for Iterated CVaR RL, i.e., Regret Minimization (RM) and Best Policy Identification (BPI). In Iterated CVaR RL-RM, the agent aims to minimize the cumulative regret in $K$ episodes, defined as

$$
\mathcal{R}(K) = \sum_{k=1}^{K} \left( V_1^*(s_1) - V_1^{\pi_k}(s_1) \right). \tag{1}
$$

In Iterated CVaR RL-BPI, given a confidence parameter $\delta \in (0, 1]$ and an accuracy parameter $\varepsilon > 0$, the agent needs to use as few trajectories (episodes) as possible to identify an $\varepsilon$-optimal policy $\hat{\pi}$, which satisfies $V_1^{\hat{\pi}}(s_1) \geq V_1^*(s_1) - \varepsilon$, with probability as least $1 - \delta$. That is, the performance of BPI is measured by the number of trajectories used, i.e., sample complexity.

**Worst Path RL.** Furthermore, we investigate an interesting limiting case of Iterated CVaR RL when $\alpha$ approaches 0, called Worst Path RL. In this case, the objective becomes maximizing the minimum possible reward (Heger, 1994). The Bellman (optimality) equations become

$$
\begin{cases}
Q_h^\pi(s,a) = r(s,a) + \min_{s'\sim p(\cdot|s,a)}(V_{h+1}^\pi(s')) \\
V_h^\pi(s) = Q_h^\pi(s, \pi_h(s)) \\
V_{H+1}^\pi(s) = 0, \ \forall s \in \mathcal{S},
\end{cases}
\quad
\begin{cases}
Q_h^*(s,a) = r(s,a) + \min_{s'\sim p(\cdot|s,a)}(V_{h+1}^*(s')) \\
V_h^*(s) = \max_{a\in\mathcal{A}} Q_h^*(s, a) \\
V_{H+1}^*(s) = 0, \ \forall s \in \mathcal{S},
\end{cases} \tag{2}
$$

where $\min_{s'\sim p(\cdot|s,a)}(V_{h+1}^\pi(s'))$ denotes the minimum value of random variable $V_{h+1}^\pi(s')$ with $s' \sim p(\cdot|s, a)$. From Eq. (2), one sees that

$$
Q_h^\pi(s,a) = \min_{(s_t,a_t)\sim\pi}\left[\sum_{t=h}^{H} r(s_t, a_t)\Big| s_h = s, a_h = a, \pi\right], \ V_h^\pi(s) = \min_{(s_t,a_t)\sim\pi}\left[\sum_{t=h}^{H} r(s_t, a_t)\Big| s_h = s, \pi\right].
$$

Thus, $Q_h^\pi(s, a)$ and $V_h^\pi(s)$ denote the minimum possible cumulative reward under policy $\pi$, starting from $(s, a)$ and $s$ at step $h$, respectively. The optimal policy $\pi^*$ maximizes the minimum possible cumulative reward (i.e., optimizes the worst path) for all starting states and steps. Formally, $\pi^*$ gives the optimal value $V_h^*(s) = \max_\pi V_h^\pi(s)$ for all $s \in \mathcal{S}$ and $h \in [H]$.

For Worst Path RL, in this paper we mainly consider the regret minimization setting, where the regret is defined the same as Eq. (1). Note that this case cannot be directly solved by taking $\alpha \to 0$ in Iterated CVaR RL, as the results there have a dependency on $\frac{1}{\alpha}$. Thus, changing from $\text{CVaR}(\cdot)$ to $\min(\cdot)$ in Worst Path RL requires a different algorithm design and analysis.

---

**Algorithm 1:** `ICVaR-RM`

---

**Input:** $\delta, \alpha, \delta' := \frac{\delta}{5}, L := \log(\frac{KHSA}{\delta'}), \bar{V}_{H+1}^k(s) = 0$ for any $k > 0$ and $s \in \mathcal{S}$

1 **for** $k = 1, 2, \ldots, K$ **do**
2      **for** $h = H, H - 1, \ldots, 1$ **do**
3          $\bar{Q}_h^k(s,a) \leftarrow \min\{r(s,a) + \text{CVaR}_{s' \sim \hat{p}^k(\cdot|s,a)}^\alpha(\bar{V}_{h+1}^k(s')) + \frac{H}{\alpha}\sqrt{\frac{L}{n_k(s,a)}}, H\}, \forall (s,a) \in \mathcal{S} \times \mathcal{A};$
4          $\bar{V}_h^k(s) \leftarrow \max_{a \in \mathcal{A}} \bar{Q}_h^k(s,a), \pi_h^k(s) \leftarrow \text{argmax}_{a \in \mathcal{A}} \bar{Q}_h^k(s,a), \forall s \in \mathcal{S};$
5      Play the episode $k$ with policy $\pi^k$, and update $n_{k+1}(s,a)$ and $\hat{p}^{k+1}(s'|s,a);$

---

The best policy identification setting of Worst Path RL, on the other hand, is very challenging. This is because we cannot establish confidence intervals under the $\min(\cdot)$ operation, and it is difficult to determine when the estimated optimal policy is accurate enough and when the algorithm should stop. We will further investigate this setting in future work.

## 4 ITERATED CVaR RL WITH REGRET MINIMIZATION

In this section, we consider regret minimization (Iterated CVaR RL-RM). We propose an algorithm `ICVaR-RM` with CVaR-adapted exploration bonuses, and demonstrate its sample efficiency.

### 4.1 ALGORITHM `ICVaR-RM` AND REGRET UPPER BOUND

We propose a value iteration-based algorithm `ICVaR-RM` (Algorithm 1), which adopts Brown-type (Brown, 2007) (CVaR-adapted) exploration bonuses and delicately pays more attention to rare but risky states. Specifically, in each episode $k$, `ICVaR-RM` computes the empirical CVaR for the values of next states $\text{CVaR}_{s' \sim \hat{p}^k(\cdot|s,a)}^\alpha(\bar{V}_{h+1}^k(s'))$ and Brown-type exploration bonuses $\frac{H}{\alpha}\sqrt{\frac{L}{n_k(s,a)}}$. Here $n^k(s,a)$ is the number of times $(s,a)$ was visited up to episode $k$, and $\hat{p}^k(s'|s,a)$ is the empirical estimate of transition probability $p(s'|s,a)$. Then, `ICVaR-RM` constructs optimistic Q-value function $\bar{Q}_h^k(s,a)$, optimistic value function $\bar{V}_h^k(s)$, and a greedy policy $\pi^k$ with respect to $\bar{Q}_h^k(s,a)$. After calculating the value functions and policy, `ICVaR-RM` plays episode $k$ with policy $\pi^k$, observes a trajectory, and updates $n_k(s,a)$ and $\hat{p}^{k+1}(s'|s,a)$. The calculation of CVaR (Line 3) can be implemented efficiently, and costs $O(S \log S)$ computation complexity (Shapiro et al., 2021).

We summarize the regret performance of `ICVaR-RM` as follows.

**Theorem 1** (Regret Upper Bound). *With probability at least $1 - \delta$, the regret of algorithm* `ICVaR-RM` *is bounded by*

$$O\left(\min\left\{\frac{1}{\sqrt{\min_{\pi,h,s: w_{\pi,h}(s)>0} w_{\pi,h}(s)}}, \frac{1}{\sqrt{\alpha^{H-1}}}\right\} \cdot \frac{HS\sqrt{KHA}}{\alpha} \log\left(\frac{KHSA}{\delta}\right)\right),$$

*where $w_{\pi,h}(s)$ denotes the probability of visiting state $s$ at step $h$ under policy $\pi$.*

**Remark 1.** The regret depends on the *minimum* between an MDP-intrinsic visitation factor $(\min_{\pi,h,s: w_{\pi,h}(s)>0} w_{\pi,h}(s))^{-\frac{1}{2}}$ and $\frac{1}{\sqrt{\alpha^{H-1}}}$. When $\alpha$ is small, the first term dominates the bound, which stands for the minimum probability of visiting an available state under any feasible policy. Note that $\min_{\pi,h,s: w_{\pi,h}(s)>0} w_{\pi,h}(s)$ takes the minimum over only the policies under which $s$ is reachable, and thus, this factor will never be zero. Indeed, this factor also exists in the lower bound (see Section 4.2). Thus, it characterizes the essential problem hardness, i.e., when the agent is highly risk-adverse, her regret will be heavily influenced by exploring critical but hard-to-reach states.

When $\alpha$ is large, $\frac{1}{\sqrt{\alpha^{H-1}}}$ instead dominates the bound. The intuition behind the factor $\frac{1}{\sqrt{\alpha^{H-1}}}$ is that for any state-action pair, the ratio of the visitation probability conditioning on transitioning to bad successor states over the original visitation probability can be upper bounded by $\frac{1}{\alpha^{H-1}}$. This ratio is critical and will appear in the regret bound (see Lemma 9 for a formal statement).

In the special case when $\alpha = 1$, our Iterated CVaR RL problem reduces to the classic RL formulation, and our regret bound becomes $\tilde{O}(HS\sqrt{KHA})$, which matches the result in existing classic

RL work (Jaksch et al., 2010). This bound has a gap of $\sqrt{HS}$ to the state-of-the-art regret bound for classic RL (Azar et al., 2017; Zanette & Brunskill, 2019). This is because our algorithm is mainly designed for general risk-sensitive cases (which require CVaR-adapted exploration bonuses), and does not use the Bernstein-type exploration bonuses (which only work for classic expectation maximization criterion). Such phenomenon also appears in existing risk-sensitive RL works (Fei et al., 2020; 2021a). Designing an algorithm which achieves an optimal regret simultaneously for both risk-sensitive cases and classic expectation maximization case is still an open problem, which we leave for future work. To validate our theoretical analysis, we also conduct experiments to exhibit the influences of parameters $\alpha$, $\delta$, $H$, $S$, $A$ and $K$ on the regret of ICVaR-RM in practice, and the empirical results well match our theoretical bound (see Appendix A).

*Challenges and Novelty in Regret Analysis.* The analysis of Iterated CVaR RL faces several challenges. (i) First of all, in Iterated CVaR RL, the contribution of a state to the regret is not proportional to its visitation probability as in standard RL analysis (Jaksch et al., 2010; Azar et al., 2017; Zanette & Brunskill, 2019). Instead, the regret is influenced more by risky but hard-to-reach states. Thus, the regret cannot be decomposed into estimation error with respect to visitation distribution. (ii) Second, unlike existing RL analysis (Jaksch et al., 2010; Azar et al., 2017; Jin et al., 2018) which typically calculates the change of expected rewards between optimistic and true value functions, in Iterated CVaR RL, we need to instead analyze the change of CVaR when the true value function shifts to an optimistic value function. To tackle these challenges, we develop a new analytical technique to bound the change of CVaR due to the value function shift via conditional transition probabilities, which can be applied to other CVaR-based RL problems. Furthermore, we establish a novel regret decomposition for Iterated CVaR RL via a distorted (conditional) visitation distribution, and quantify the deviation between this distorted visitation distribution and the original visitation distribution.

Below we present a proof sketch for Theorem 1 (see Appendix D.1 for a complete proof).

*Proof sketch of Theorem 1.* First, we introduce a key inequality (Eq. (3)) to bound the change of CVaR when the true value function shifts to an optimistic one. To this end, let $\beta^{\alpha,V}(\cdot|s,a) \in \mathbb{R}^S$ denote the conditional transition probability conditioning on transitioning to the worst $\alpha$-portion successor states $s'$, i.e., with the lowest values $V(s')$. It satisfies that $\sum_{s' \in \mathcal{S}} \beta^{\alpha,V}(s'|s,a) \cdot V(s') = \text{CVaR}^{\alpha}_{s' \sim p(\cdot|s,a)}(V(s'))$. Then, for any $(s,a)$ and value functions $\bar{V}, V$ such that $\bar{V}(s') \geq V(s')$ for any $s' \in \mathcal{S}$, we have

$$\text{CVaR}^{\alpha}_{s' \sim p(\cdot|s,a)}(\bar{V}(s')) - \text{CVaR}^{\alpha}_{s' \sim p(\cdot|s,a)}(V(s')) \leq \beta^{\alpha,V}(\cdot|s,a)^{\top}\left(\bar{V} - V\right). \tag{3}$$

Eq. (3) implies that the deviation of CVaR between optimistic and true value functions can be bounded by their value deviation under the conditional transition probability, which resolves the aforementioned challenge (ii), and serves as the basis of our recurrent regret decomposition.

Now, since $\bar{V}^k_h$ is an optimistic estimate of $V^*_h$, we decompose the regret in episode $k$ as

$$\bar{V}^k_1(s^k_1) - V^{\pi^k}_1(s^k_1) \overset{(a)}{\leq} \frac{H}{\alpha}\sqrt{\frac{L}{n_k(s^k_1, a^k_1)}} + \text{CVaR}^{\alpha}_{s' \sim \hat{p}^k(\cdot|s^k_1, a^k_1)}(\bar{V}^k_2(s')) - \text{CVaR}^{\alpha}_{s' \sim p(\cdot|s^k_1, a^k_1)}(\bar{V}^k_2(s'))$$

$$+ \text{CVaR}^{\alpha}_{s' \sim p(\cdot|s^k_1, a^k_1)}(\bar{V}^k_2(s')) - \text{CVaR}^{\alpha}_{s' \sim p(\cdot|s^k_1, a^k_1)}(V^{\pi^k}_2(s'))$$

$$\overset{(b)}{\leq} \frac{H}{\alpha}\sqrt{\frac{L}{n_k(s^k_1, a^k_1)}} + \frac{4H}{\alpha}\sqrt{\frac{SL}{n_k(s^k_1, a^k_1)}} + \beta^{\alpha,V^{\pi^k}_2}(\cdot|s^k_1, a^k_1)^{\top}\left(\bar{V}^k_2 - V^{\pi^k}_2\right)$$

$$\overset{(c)}{\leq} \sum_{h=1}^{H} \sum_{(s,a)} w^{\text{CVaR},\alpha,V^{\pi^k}}_{kh}(s,a) \cdot \frac{H\sqrt{L} + 4H\sqrt{SL}}{\alpha\sqrt{n_k(s,a)}} \tag{4}$$

Here $w^{\text{CVaR},\alpha,V^{\pi^k}}_{kh}(s,a)$ denotes the conditional probability of visiting $(s,a)$ at step $h$ of episode $k$, conditioning on transitioning to the worst $\alpha$-portion successor states $s'$ (i.e., with the lowest $\alpha$-portion values $V^{\pi^k}_{h'+1}(s')$) at each step $h' = 1, \ldots, h-1$. Intuitively, $w^{\text{CVaR},\alpha,V^{\pi^k}}_{kh}(s,a)$ is a distorted visitation probability under the conditional transition probability $\beta^{\alpha,V^{\pi^k}}(\cdot|\cdot,\cdot)$. Inequality (b) uses the concentration of CVaR and Eq. (3). Inequality (c) follows from recurrently applying steps (a)-(b) to unfold $\bar{V}^k_h(\cdot) - V^{\pi^k}_h(\cdot)$ for $h = 2, \ldots, H$, and the fact that $w^{\text{CVaR},\alpha,V^{\pi^k}}_{kh}(s,a)$ is the visitation

probability under conditional transition probability $\beta^{\alpha,V^{\pi^k}}(\cdot|\cdot,\cdot)$. Eq. (4) decomposes the regret into estimation error at all state-action pairs via the distorted (conditional) visitation distribution $w_{kh}^{\mathrm{CVaR},\alpha,V^{\pi^k}}(s,a)$, which overcomes the aforementioned challenge (i).

Summing Eq. (4) over all episodes $k \in [K]$ and using the Cauchy–Schwarz inequality, we have

$$\mathbb{E}[\mathcal{R}(K)] \leq \frac{5H\sqrt{SL}}{\alpha}\sqrt{\sum_{k=1}^{K}\sum_{h=1}^{H}\sum_{(s,a)}\frac{w_{kh}^{\mathrm{CVaR},\alpha,V^{\pi^k}}(s,a)}{n_k(s,a)}} \cdot \sqrt{\sum_{k=1}^{K}\sum_{h=1}^{H}\sum_{(s,a)}w_{kh}^{\mathrm{CVaR},\alpha,V^{\pi^k}}(s,a)}$$

$$\overset{(d)}{=}\frac{5H\sqrt{SL}\cdot\sqrt{KH}}{\alpha}\sqrt{\sum_{k=1}^{K}\sum_{h=1}^{H}\sum_{(s,a)}\frac{w_{kh}^{\mathrm{CVaR},\alpha,V^{\pi^k}}(s,a)}{w_{kh}(s,a)}\cdot\frac{w_{kh}(s,a)}{n_k(s,a)}\cdot\mathbb{1}\left\{w_{kh}(s,a)\neq 0\right\}}$$

$$\overset{(e)}{\leq}\frac{5H\sqrt{KHSL}}{\alpha}\sqrt{\min\left\{\frac{1}{\min\limits_{\pi,h,(s,a):\,w_{\pi,h}(s,a)>0}w_{\pi,h}(s,a)},\frac{1}{\alpha^{H-1}}\right\}\sum_{k=1}^{K}\sum_{h=1}^{H}\sum_{(s,a)}\frac{w_{kh}(s,a)}{n_k(s,a)}},$$

Here $w_{kh}(s,a)$ denotes the probability of visiting $(s,a)$ at step $h$ of episode $k$, and $w_{\pi,h}(s,a)$ denotes the probability of visiting $(s,a)$ at step $h$ under policy $\pi$. Equality (d) uses the facts that $\sum_{(s,a)}w_{kh}^{\mathrm{CVaR},\alpha,V^{\pi^k}}(s,a) = 1$, and if the visitation probability $w_{kh}(s,a) = 0$, the conditional visitation probability $w_{kh}^{\mathrm{CVaR},\alpha,V^{\pi^k}}(s,a)$ must be 0 as well. Inequality (e) is due to that $w_{kh}^{\mathrm{CVaR},\alpha,V^{\pi^k}}(s,a)/w_{kh}(s,a)$ can be bounded by both $1/\min_{\pi,h,(s,a):\,w_{\pi,h}(s,a)>0}w_{\pi,h}(s,a)$ and $1/\alpha^{H-1}$. Specifically, the bound $1/\min_{\pi,h,(s,a):\,w_{\pi,h}(s,a)>0}w_{\pi,h}(s,a)$ follows from $\min_{\pi,h,(s,a):\,w_{\pi,h}(s,a)>0}w_{\pi,h}(s,a) \leq w_{kh}(s,a)$, and the bound $1/\alpha^{H-1}$ comes from the fact that the conditional visitation probability $w_{kh}^{\mathrm{CVaR},\alpha,V^{\pi^k}}(s,a)$ is at most $1/\alpha^{H-1}$ times the visitation probability $w_{kh}(s,a)$. Having established the above, we can use a similar analysis as that in classic RL (Azar et al., 2017; Zanette & Brunskill, 2019) to bound $\sum_{k=1}^{K}\sum_{h=1}^{H}\sum_{(s,a)}\frac{w_{kh}(s,a)}{n_k(s,a)}$, and then, we can obtain Theorem 1. $\square$

## 4.2 REGRET LOWER BOUND

We now present a regret lower bound to demonstrate the optimality of algorithm `ICVaR-RM`.

**Theorem 2** (Regret Lower Bound). *There exists an instance of Iterated CVaR RL-RM, where $\min_{\pi,h,s:\,w_{\pi,h}(s)>0}w_{\pi,h}(s) > \alpha^{H-1}$ and the regret of any algorithm is at least*

$$\Omega\left(H\sqrt{\frac{AK}{\alpha\min_{\pi,h,s:\,w_{\pi,h}(s)>0}w_{\pi,h}(s)}}\right). \tag{5}$$

*In addition, there exists an instance of Iterated CVaR RL-RM, where $\alpha^{H-1} > \min_{\pi,h,s:\,w_{\pi,h}(s)>0}w_{\pi,h}(s)$ and the regret of any algorithm is at least $\Omega(\sqrt{\frac{AK}{\alpha^{H-1}}})$.*

**Remark 2.** Theorem 2 demonstrates that when $\alpha$ is small, the factor $\min_{\pi,h,s:\,w_{\pi,h}(s)>0}w_{\pi,h}(s)$ is inevitable in general. This reveals the intrinsic hardness of Iterated CVaR RL, i.e., when the agent is highly sensitive to bad situations, she must suffer a regret due to exploring risky but hard-to-reach states. This lower bound also validates that `ICVaR-RM` is near-optimal with respect to $K$.

*Lower Bound Analysis.* Here we provide the proof idea of the first lower bound (Eq. (5)) in Theorem 2, and defer the full proof to Appendix D.2. We construct an instance with a hard-to-reach bandit state (which has an optimal action and multiple sub-optimal actions), and show that this state is critical for minimizing the regret, but difficult for any algorithm to learn. As shown in Figure 1, we consider an MDP with $A$ actions, $n$ regular states $s_1, \ldots, s_n$ and three absorbing states $x_1, x_2, x_3$, where $n < \frac{1}{2}H$. The

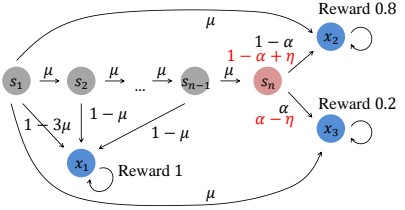

Figure 1: Instance for the lower bound.

---

**Algorithm 2:** `MaxWP`

---

**Input:** $\delta$, $\delta' := \frac{\delta}{2}$, $L := \log(\frac{SA}{\delta'})$, $\hat{V}_{H+1}^k(s) = 0$ for any $k > 0$ and $s \in \mathcal{S}$

1 **for** $k = 1, 2, \ldots, K$ **do**
2     **for** $h = H, H-1, \ldots, 1$ **do**
3         $\hat{Q}_h^k(s,a) \leftarrow r(s,a) + \min_{s' \sim \hat{p}^k(\cdot|s,a)}(\hat{V}_{h+1}^k(s'))$, $\forall (s,a) \in \mathcal{S} \times \mathcal{A}$;
4         $\hat{V}_h^k(s) \leftarrow \max_{a \in \mathcal{A}} \hat{Q}_h^k(s,a)$, $\pi_h^k(s) \leftarrow \mathrm{argmax}_{a \in \mathcal{A}} \hat{Q}_h^k(s,a)$, $\forall s \in \mathcal{S}$;
5     Play the episode $k$ with policy $\pi^k$, and update $n_{k+1}(s,a)$ and $\hat{p}^{k+1}(s'|s,a)$;

---

reward function $r(s,a)$ depends only on the states, i.e.,
$s_1, \ldots, s_n$ generate zero reward, and $x_1, x_2, x_3$ generate rewards 1, 0.8 and 0.2, respectively. Let $\mu$ be a parameter such that $0 < \alpha < \mu < \frac{1}{3}$. Under all actions, state $s_1$ transitions to $s_2, x_1, x_2, x_3$ with probabilities $\mu, 1 - 3\mu, \mu$ and $\mu$, respectively, and state $s_i$ ($2 \leq i \leq n-1$) transitions to $s_{i+1}, x_1$ with probabilities $\mu$ and $1 - \mu$, respectively. For the bandit state $s_n$, under the optimal action, $s_n$ transitions to $x_2, x_3$ with probabilities $1 - \alpha + \eta$ and $\alpha - \eta$, respectively. Under sub-optimal actions, $s_n$ transitions to $x_2, x_3$ with probabilities $1 - \alpha$ and $\alpha$, respectively.

In this MDP, under the Iterated CVaR criterion, the value function mainly depends on the path $s_1 \rightarrow s_2 \rightarrow \cdots \rightarrow s_n \rightarrow x_2/x_3$, and especially on the action choice in the bandit state $s_n$. Thus, to distinguish the optimal action in $s_n$, any algorithm must suffer a regret dependent on the probability of visiting $s_n$, which is exactly the minimum visitation probability over all reachable states $\min_{\pi,h,s:\, w_{\pi,h}(s)>0} w_{\pi,h}(s)$. Note that in this instance, $\min_{\pi,h,s:\, w_{\pi,h}(s)>0} w_{\pi,h}(s) = \mu^{n-1}$, which does not depend on $\alpha$ and $H$. This demonstrates that there is an essential dependency on $\min_{\pi,h,s:\, w_{\pi,h}(s)>0} w_{\pi,h}(s)$ in the lower bound.

## 5 ITERATED CVAR RL WITH BEST POLICY IDENTIFICATION

In this section, we design an efficient algorithm `ICVaR-BPI`, and establish sample complexity upper and lower bounds for Iterated CVaR RL with best policy identification (BPI).

### 5.1 ALGORITHM `ICVaR-BPI` AND SAMPLE COMPLEXITY UPPER BOUND

Algorithm `ICVaR-BPI` introduces a novel distorted (conditional) empirical transition probability to construct estimation error, which effectively assigns more attention to bad situations and fits the main focus of the Iterated CVaR criterion. Due to space limit, we defer the pseudo-code and detailed description of `ICVaR-BPI` to Appendix E.1. Below we present the sample complexity of `ICVaR-BPI`.

**Theorem 3** (Sample Complexity Upper Bound). *The number of trajectories used by algorithm* `ICVaR-BPI` *to return an $\varepsilon$-optimal policy with probability at least $1 - \delta$ is bounded by*

$$O\left( \min\left\{ \frac{1}{\min_{\pi,h,s:\, w_{\pi,h}(s)>0} w_{\pi,h}(s)}, \frac{1}{\alpha^{H-1}} \right\} \frac{H^3 S^2 A}{\varepsilon^2 \alpha^2} \cdot C \right),$$

*where* $C := \log^2(\min\{ \frac{1}{\min_{\pi,h,s:\, w_{\pi,h}(s)>0} w_{\pi,h}(s)}, \frac{1}{\alpha^{H-1}} \} \frac{HSA}{\varepsilon \alpha \delta})$.

Similar to Theorem 1, $\min_{\pi,h,s:\, w_{\pi,h}(s)>0} w_{\pi,h}(s)$ and $\alpha^{H-1}$ dominate the bound for a large $\alpha$ and a small $\alpha$, respectively. When $\alpha = 1$, the problem reduces to the classic RL formulation with best policy identification, and our sample complexity becomes $\tilde{O}(\frac{H^3 S^2 A}{\varepsilon^2})$, which recovers the result in prior classic RL work (Dann et al., 2017). Similar to Theorem 1, this bound has a gap of $HS$ to the state-of-the-art sample complexity for classic RL (Ménard et al., 2021). This gap is due to the fact that the result in (Ménard et al., 2021) is obtained using the Bernstein-type exploration bonuses, which are more fine-grained for the classic RL problem but do not work for general risk-sensitive cases, because it cannot be used to quantify the estimation error of CVaR.

To validate the tightness of Theorem 3, we further provide sample complexity lower bounds $\Omega(\frac{H^2 A}{\varepsilon^2 \alpha \min_{\pi,h,s:\, w_{\pi,h}(s)>0} w_{\pi,h}(s)} \log(\frac{1}{\delta}))$ and $\Omega(\frac{A}{\alpha^{H-1}\varepsilon^2} \log(\frac{1}{\delta}))$ for different instances, which demonstrate that the factor $\min\{ 1/\min_{\pi,h,s:\, w_{\pi,h}(s)>0} w_{\pi,h}(s), 1/\alpha^{H-1} \}$ is indispensable in general (see Appendix E.3 for a formal statement of lower bound).

# 6 WORST PATH RL

In this section, we investigate an interesting limiting case of Iterated CVaR RL when $\alpha \to 0$, called Worst Path RL, in which case the agent aims to maximize the minimum possible cumulative reward.

Worst Path RL has a *unique feature* that, the value function (Eq. (2)) concerns only the minimum value of successor states, which are independent of specific transition probabilities. Therefore, once we learn the connectivity among states, we can perform a planning to compute the optimal policy. Yet, this feature does not make the Worst Path RL problem trivial, because it is still challenging to distinguish whether a successor state is hard to reach or does not exist. As a result, a careful scheme is needed to both explore undetected successor states and exploit observations to minimize regret.

## 6.1 ALGORITHM MaxWP AND REGRET UPPER BOUND

We design an algorithm MaxWP (Algorithm 2) based on a simple and efficient empirical Q-value function, which makes full use of the unique feature of Worst Path RL, and simultaneously explores undetected successor states and exploits the current best action. Specifically, in episode $k$, MaxWP constructs empirical Q-value/value functions $\hat{Q}_h^k(s, a), \hat{V}_h^k(s)$ using the estimated lowest value of next states, and then, takes a greedy policy $\pi_h^k(s)$ with respect to $\hat{Q}_h^k(s, a)$ in this episode.

The intuition behind MaxWP is as follows. Since the Q-value function for Worst Path RL uses the $\min$ operator, if the Q-value function is not accurately estimated, it can only be over-estimated (not under-estimated). If over-estimation happens, MaxWP will be exploring an over-estimated action and urging its empirical Q-value to get back to its true Q-value. Otherwise, if the Q-value function is already accurate, MaxWP just selects the optimal action. In other words, MaxWP combines the exploration of over-estimated actions (which lead to undetected successor states) and exploitation of current best actions. Below we provide the regret guarantee for algorithm MaxWP.

**Theorem 4.** *With probability at least $1 - \delta$, the regret of algorithm* MaxWP *is bounded by*

$$O\left( \sum_{(s,a) \in \mathcal{S} \times \mathcal{A}} \frac{H}{\min_{\pi: \, \upsilon_\pi(s,a) > 0} \upsilon_\pi(s,a) \cdot \min_{s' \in \text{supp}(p(\cdot|s,a))} p(s'|s,a)} \log\left( \frac{SA}{\delta} \right) \right),$$

*where $\upsilon_\pi(s, a)$ denotes the probability $(s, a)$ is visited at least once in an episode under policy $\pi$.*

**Remark 3.** The factor $\min_{\pi: \, \upsilon_\pi(s,a) > 0} \upsilon_\pi(s, a)$ stands for the minimum probability of visiting $(s, a)$ at least once in an episode over all feasible policies, and $\min_{s' \in \text{supp}(p(\cdot|s,a))} p(s'|s, a)$ denotes the minimum transition probability over all successor states of $(s, a)$. Note that this result cannot be implied by Theorem 1, because the result for Iterated CVaR RL there depends on $\frac{1}{\alpha}$, and simply taking $\alpha \to 0$ leads to a vacuous bound.

Theorem 4 demonstrates that algorithm MaxWP enjoys a constant regret with respect to $K$. This constant regret is made possible by the unique feature of Worst Path RL that, under the worst path metric, once the agent determines the connectivity among states, she can accurately estimate the value function and find the optimal policy. Furthermore, determining the connectivity among states (with a given confidence) only requires a number of samples independent of $K$. MaxWP effectively utilizes this problem feature, and efficiently explores the connectivity among states.

To validate the optimality of our regret upper bound, we also provide a lower bound $\Omega(\max_{(s,a):\exists h, \, a \neq \pi_h^*(s)} \frac{H}{\min_{\pi: \upsilon_\pi(s,a) > 0} \upsilon_\pi(s,a) \cdot \min_{s' \in \text{supp}(p(\cdot|s,a))} p(s'|s,a)})$ for Worst Path RL, which demonstrates the tightness of the factors $\min_{\pi: \, \upsilon_\pi(s,a) > 0} \upsilon_\pi(s, a)$ and $\min_{s' \in \text{supp}(p(\cdot|s,a))} p(s'|s, a)$.

# 7 CONCLUSION

In this paper, we investigate a novel Iterated CVaR RL problem with the regret minimization and best policy identification metrics. We design two efficient algorithms ICVaR-RM and ICVaR-BPI, and provide nearly matching regret/sample complexity upper and lower bounds with respect to $K$. We also study an interesting limiting case called Worst Path RL, and propose a simple and efficient algorithm MaxWP with rigorous regret guarantees. There are several interesting directions for future work, e.g., further closing the gap between upper and lower bounds, and extending our model and results from the tabular setting to the function approximation framework.

## ACKNOWLEDGEMENTS

The work of Yihan Du and Longbo Huang is supported by the Technology and Innovation Major Project of the Ministry of Science and Technology of China under Grant 2020AAA0108400 and 2020AAA0108403, the Tsinghua University Initiative Scientific Research Program, and Tsinghua Precision Medicine Foundation 10001020109. The work of Siwei Wang was supported in part by the National Natural Science Foundation of China Grant 62106122.

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

# APPENDIX

## A EXPERIMENTS

In this section, we provide experimental results to evaluate the empirical performance of our algorithm ICVaR-RM, and compare it to the state-of-the-art algorithms EULER (Zanette & Brunskill, 2019) and RSVI2 (Fei et al., 2021a) for classic RL and risk-sensitive RL, respectively.

In our experiments, we consider an $H$-layered MDP with $S = 3(H - 1) + 1$ states and $A$ actions. There is a single state $s_0$ (initial state) in layer 1. For any $2 \leq h \leq H$, there are three states $s_{3(h-2)+1}, s_{3(h-2)+2}$ and $s_{3(h-2)+3}$ in layer $h$, which induce rewards $1, 0$ and $0.4$, respectively. The agent starts from $s_0$ in layer 1, and for each step $h \in [H]$, she takes an action from $\{a_1, \ldots, a_A\}$, and then transitions to one of three states in the next layer. For any $a \in \{a_1, \ldots, a_{A-1}\}$, action $a$ leads to $s_{3(h-1)+1}$ and $s_{3(h-1)+2}$ with probabilities $0.5$ and $0.5$, respectively. Action $a_A$ leads to $s_{3(h-1)+2}$ and $s_{3(h-1)+3}$ with probabilities $0.001$ and $0.999$, respectively.

We set $\alpha \in \{0.05, 0.1, 0.15\}$, $\delta \in \{0.5, 0.005, 0.00005\}$, $H \in \{2, 5, 10\}$, $S \in \{7, 13, 25\}$, $A \in \{3, 5, 12\}$ and $K \in [0, 10000]$ (the change of $K$ can be seen from the X-axis in Figure 2). We take $\alpha = 0.05$, $\delta = 0.005$, $H = 5$, $S = 13$, $A = 5$ and $K = 10000$ as the basic setting, and change parameters $\alpha$, $\delta$, $H$, $S$, $A$ and $K$ to see how they affect the empirical performance of algorithm ICVaR-RM. For each algorithm, we perform 20 independent runs and report the average regret across runs with $95\%$ confidence intervals.

As shown in Figure 2, our algorithm ICVaR-RM achieves a significantly lower regret than the other algorithms EULER (Zanette & Brunskill, 2019) and RSVI2 (Fei et al., 2021a), which demonstrates that ICVaR-RM can effectively control the risk under the Iterated CVaR criterion and shows performance superiority over the baselines. Moreover, the influences of parameters $\alpha$, $\delta$, $H$, $S$, $A$ and $K$ on the regret of algorithm ICVaR-RM match our theoretical bounds. Specifically, as $\alpha$ or $\delta$ increases, the regret of ICVaR-RM decreases. As $H, S$ or $A$ increases, the regret of ICVaR-RM increases as well. As the number of episodes $K$ increases, the regret of ICVaR-RM increases at a sublinear rate.

## B RELATED WORK

Below we present a complete review of related works.

**CVaR-based MDPs (Known Transition).** Boda & Filar (2006); Ott (2010); Bäuerle & Ott (2011); Haskell & Jain (2015); Chow et al. (2015) study the CVaR MDP problem where the objective is to minimize the CVaR of the total cost with known transition, and demonstrate that the optimal policy for CVaR MDP is history-dependent (not Markovian) and is inefficient to exactly compute. Hardy & Wirch (2004) firstly define the Iterated CVaR measure, and prove that it is a coherent dynamic risk measure, and applicable to equity-linked insurance. Osogami (2012); Chu & Zhang (2014); Bäuerle & Glauner (2022) investigate iterated coherent risk measures (including Iterated CVaR) in MDPs, and prove the existence of Markovian optimal policies for these MDPs. The above works focus mainly on designing planning algorithms and derive planning error guarantees for *known* transition, while our work develops RL algorithms (interacting with the environment online) and provides regret and sample complexity guarantees for *unknown* transition.

**Risk-Sensitive Reinforcement Learning (Unknown Transition).** Heger (1994); Coraluppi & Marcus (1997; 1999) consider minimizing the worst-case cost in RL, and present dynamic programming of value functions and heuristic algorithms without theoretical analysis. Borkar (2001; 2002) study risk-sensitive RL with the exponential utility measure, and design algorithms based on actor–critic learning and Q-learning, respectively. Di Castro et al. (2012); La & Ghavamzadeh (2013) investigate variance-related risk measures, and devise policy gradient and actor-critic-based algorithms with convergence analysis. Tamar et al. (2015) consider maximizing the CVaR of the total reward, and propose a sampling-based estimator for the CVaR gradient and a stochastic gradient decent algorithm to optimize CVaR. Keramati et al. (2020) also investigate optimizing the CVaR of the total reward, and design an algorithm based on an optimistic version of the distributional Bellman operator. Borkar & Jain (2014); Chow & Ghavamzadeh (2014); Chow et al. (2017) study how to minimize the expected total cost with CVaR-based constraints, and develop policy gradient, actor-critic and

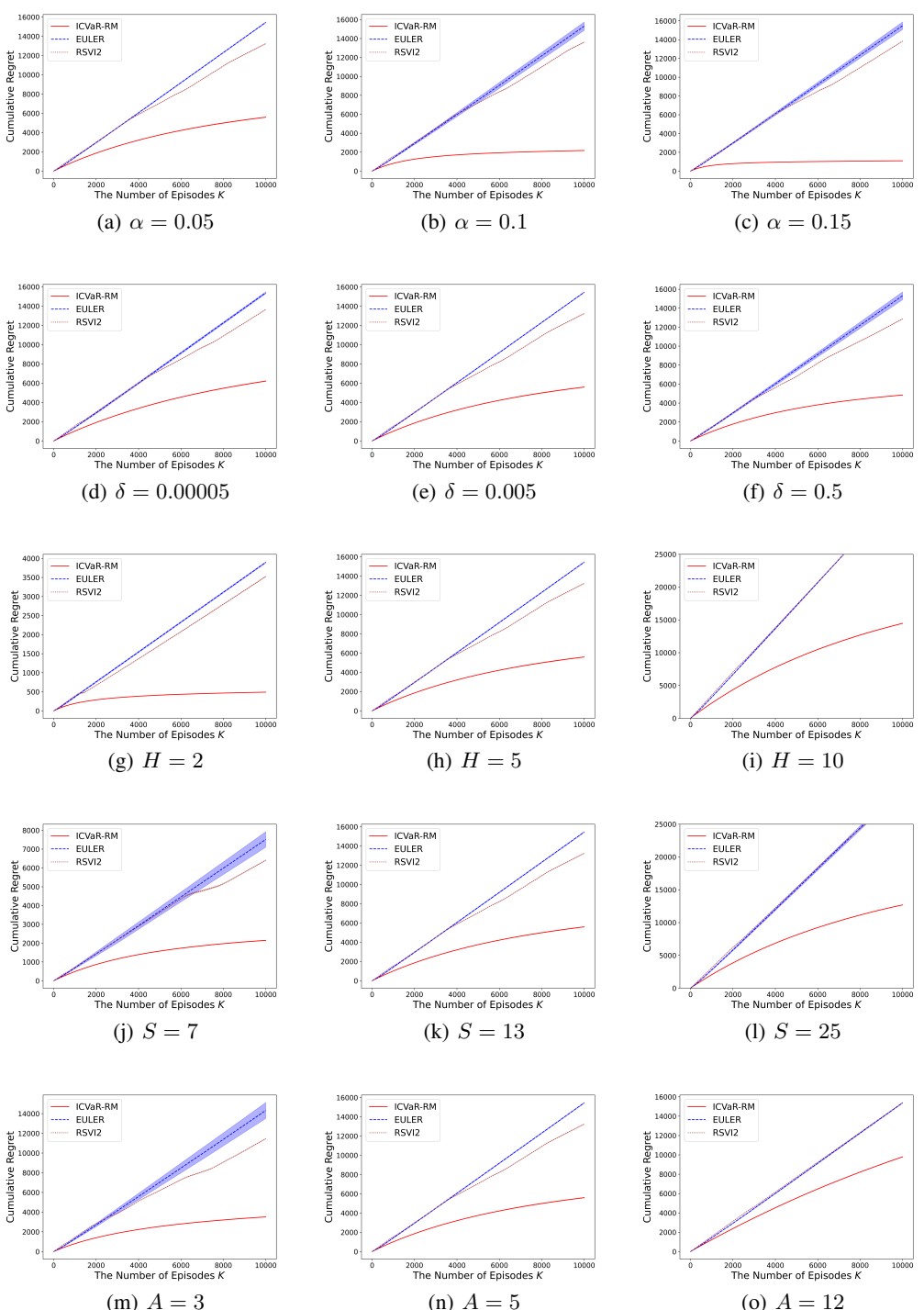

Figure 2: Experimental results for Iterated CVaR RL.

stochastic approximation-style algorithms. The above works mainly give convergence analysis, and do not provide finite-time regret and sample complexity guarantees as in our work.

To our best knowledge, there are only a few risk-sensitive RL works which provide finite-time regret analysis (Fei et al., 2020; 2021a;b). Fei et al. (2020) consider risk-sensitive RL with the exponential utility criterion, and propose algorithms based on logarithmic-exponential transformation and least-squares updates. Fei et al. (2021a) further improve the regret bound in (Fei et al., 2020) by

developing an exponential Bellmen equation and a Bellman backup analytical procedure. Fei et al. (2021b) extend the model and results in (Fei et al., 2020; 2021a) from the tabular setting to the function approximation framework. Our work is very different from the above works (Fei et al., 2020; 2021a;b) in formulation, algorithms and results. The above works (Fei et al., 2020; 2021a;b) use the exponential utility criterion to characterize the risk and take all successor states into account in decision making. They design algorithms based on exponential Bellmen equations and doubly decaying exploration bonuses. In contrast, we interpret the risk by the Iterated CVaR criterion, which primarily concerns the worst $\alpha$-portion successor states. We develop algorithms using CVaR-adapted exploration bonuses.

The works we discuss above fall in the literature of RL with risk-sensitive criteria. There are also other RL works which focus on state-wise safety. Cheng et al. (2019) utilize control barrier functions (CBFs) to ensure the agent within a set of safe sets and guide the learning by constraining explorable polices. Fatemi et al. (2019; 2021) define the notion of dead-end states (which lead to suboptimal terminal state with probability 1 in finite steps) and aim to avoid getting into dead-end states. The formulations and algorithms in these works greatly differ from ours, and they do not provide finite-time regret and sample complexity analysis as us. We refer interested readers to the survey (Garcıa & Fernández, 2015) for detailed categorization and discussion on safe RL.

## C   MORE DISCUSSION ON ITERATED CVAR RL

In this section, we first present the expanded value function definitions for Iterated CVaR RL. Then, we compare Iterated CVaR RL with existing risk-sensitive MDP models, including CVaR MDP (Boda & Filar, 2006; Ott, 2010; Bäuerle & Ott, 2011; Chow et al., 2015) and the exponential utility-based RL (Fei et al., 2020; 2021a).

### C.1   VALUE FUNCTION DEFINITIONS FOR ITERATED CVAR RL

The value function definition for Iterated CVaR RL, i.e., Eq. (i) in Section 3, can be expanded as

$$Q_h^\pi(s,a) = r(s,a) + \text{CVaR}_{s_{h+1} \sim p(\cdot|s,a)}^\alpha \Big( r(s_{h+1}, \pi_{h+1}(s_{h+1}))$$

$$+ \text{CVaR}_{s_{h+2} \sim p(\cdot|s_{h+1},\pi_{h+1}(s_{h+1}))}^\alpha \Big( \ldots \text{CVaR}_{s_H \sim p(\cdot|s_{H-1},\pi_{H-1}(s_{H-1}))}^\alpha (r(s_H, \pi_H(s_H))) \Big) \Big),$$

$$V_h^\pi(s) = r(s, \pi_h(s)) + \text{CVaR}_{s_{h+1} \sim p(\cdot|s,\pi_h(s))}^\alpha \Big( r(s_{h+1}, \pi_{h+1}(s_{h+1}))$$

$$+ \text{CVaR}_{s_{h+2} \sim p(\cdot|s_{h+1},\pi_{h+1}(s_{h+1}))}^\alpha \Big( \ldots \text{CVaR}_{s_H \sim p(\cdot|s_{H-1},\pi_{H-1}(s_{H-1}))}^\alpha (r(s_H, \pi_H(s_H))) \Big) \Big).$$

Similarly, the optimal value function definition, e.g., Eq. (ii) in Section 3, can be expanded as

$$Q_h^*(s,a) = \max_\pi \Big\{ r(s,a) + \text{CVaR}_{s_{h+1} \sim p(\cdot|s,a)}^\alpha \Big( r(s_{h+1}, \pi_{h+1}(s_{h+1}))$$

$$+ \text{CVaR}_{s_{h+2} \sim p(\cdot|s_{h+1},\pi_{h+1}(s_{h+1}))}^\alpha \Big( \ldots \text{CVaR}_{s_H \sim p(\cdot|s_{H-1},\pi_{H-1}(s_{H-1}))}^\alpha (r(s_H, \pi_H(s_H))) \Big) \Big) \Big\},$$

$$V_h^*(s) = \max_\pi \Big\{ r(s, \pi_h(s)) + \text{CVaR}_{s_{h+1} \sim p(\cdot|s,\pi_h(s))}^\alpha \Big( r(s_{h+1}, \pi_{h+1}(s_{h+1}))$$

$$+ \text{CVaR}_{s_{h+2} \sim p(\cdot|s_{h+1},\pi_{h+1}(s_{h+1}))}^\alpha \Big( \ldots \text{CVaR}_{s_H \sim p(\cdot|s_{H-1},\pi_{H-1}(s_{H-1}))}^\alpha (r(s_H, \pi_H(s_H))) \Big) \Big) \Big\}. \tag{6}$$

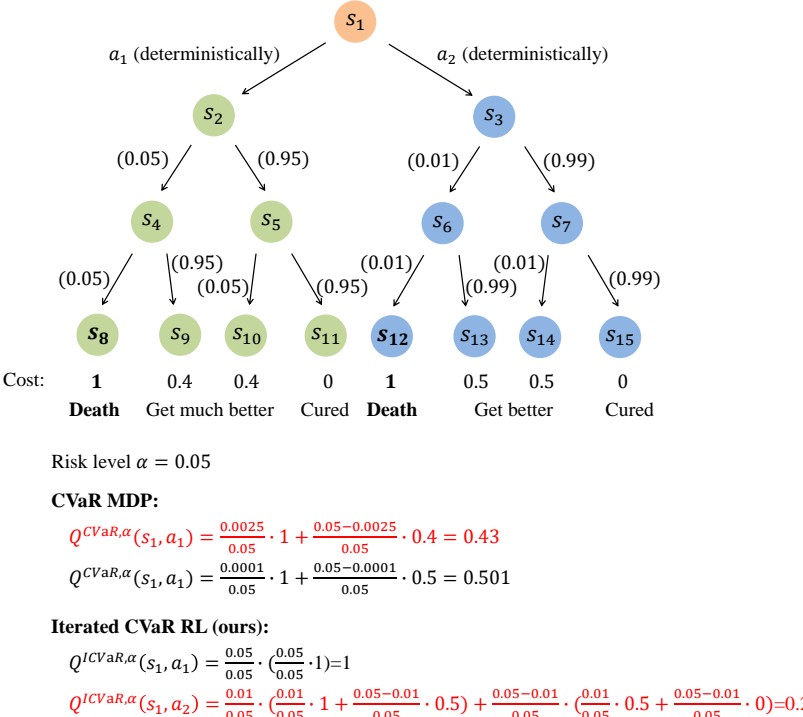

Cost:

| | | | | | | | |
|---|---|---|---|---|---|---|---|
| **1** | 0.4 | 0.4 | 0 | **1** | 0.5 | 0.5 | 0 |
| **Death** | Get much better | | Cured | **Death** | Get better | | Cured |

Risk level $\alpha = 0.05$

**CVaR MDP:**

$Q^{CVaR,\alpha}(s_1, a_1) = \frac{0.0025}{0.05} \cdot 1 + \frac{0.05-0.0025}{0.05} \cdot 0.4 = 0.43$

$Q^{CVaR,\alpha}(s_1, a_1) = \frac{0.0001}{0.05} \cdot 1 + \frac{0.05-0.0001}{0.05} \cdot 0.5 = 0.501$

**Iterated CVaR RL (ours):**

$Q^{ICVaR,\alpha}(s_1, a_1) = \frac{0.05}{0.05} \cdot (\frac{0.05}{0.05} \cdot 1) = 1$

$Q^{ICVaR,\alpha}(s_1, a_2) = \frac{0.01}{0.05} \cdot (\frac{0.01}{0.05} \cdot 1 + \frac{0.05-0.01}{0.05} \cdot 0.5) + \frac{0.05-0.01}{0.05} \cdot (\frac{0.01}{0.05} \cdot 0.5 + \frac{0.05-0.01}{0.05} \cdot 0) = 0.2$

Figure 3: Illustrating example for the comparison between CVaR MDP and Iterated CVaR RL.

From the above value function definitions, we can see that, Iterated CVaR RL aims to maximize the worst $\alpha$-portion tail of the reward-to-go at each step, i.e., taking the CVaR operator on the reward-to-go at each step. Intuitively, Iterated CVaR RL wants to optimize the performance even when bad situations happen at each decision stage.

## C.2 COMPARISON WITH CVaR MDP

The objective of CVaR MDP, e.g., (Boda & Filar, 2006; Ott, 2010; Bäuerle & Ott, 2011; Chow et al., 2015), is to maximize the worst $\alpha$-portion of the *total* reward, which is formally defined as

$$\max_{\pi} \text{CVaR}^{\alpha}_{(s_h, a_h) \sim p, \pi} \left( \sum_{h=1}^{H} r(s_h, a_h) \right).$$

Compared to our Iterated CVaR RL (Eq. (6) and Eq. (ii) in Section 3) which concerns bad situations *at each step*, CVaR MDP takes more cumulative reward into account and prefers actions which have better performance in general, but can have larger probabilities of getting into catastrophic states. Thus, CVaR MDP is suitable for scenarios where bad situations lead to a higher cost but not fatal damage, e.g., finance. In contrast, Iterated CVaR RL prefers actions which have smaller probabilities of getting into catastrophic states. Hence, Iterated CVaR RL is most suitable for safety-critical applications, where catastrophic states are unacceptable and need to be carefully avoid, e.g., clinical treatment planning.

We emphasize that Iterated CVaR is *not equivalent to* simply taking the worst $\alpha^H$-portion of the total reward. In fact, the good $(1 - \alpha^H)$-portion of the total reward also contributes to Iterated CVaR. This is because Iterated CVaR accounts bad situations for all states (both good and bad states) in its iterated computation, instead of just considering bad situations upon bad states.

Below we provide an example of clinical treatment planning to illustrate the difference between Iterated CVaR and CVaR MDP. Here we interpret the objective as cost minimization for ease of understanding, and set the risk level $\alpha = 0.05$.

Consider a 4-layered binary tree-structured MDP shown in Figure 3. The state sets in layers 1, 2, 3 and 4 are $\{s_1\}$, $\{s_2, s_3\}$, $\{s_4, \ldots, s_7\}$ and $\{s_8, \ldots, s_{15}\}$, respectively. There are two actions $a_1, a_2$ in each state, and $a_1, a_2$ have the same transition distribution in all states except the initial state $s_1$. Thus, a policy is to decide whether to choose $a_1$ or $a_2$ in state $s_1$, which leads to different subsequent costs.

The agent starts from the initial state $s_1$ in layer 1. If the agent takes action $a_1$, she will transition to state $s_2$ deterministically, and goes into the left sub-tree. On the other hand, if the agent takes action $a_2$ in state $s_1$, she will transition to state $s_3$ deterministically, and enters the right sub-tree.

If the agent goes into the left sub-tree (state $s_2$) in layer 2, she will transition to $s_4$ and $s_5$ in layer 3 with probabilities 0.05 and 0.95, respectively. Then, if she starts from state $s_4$ in layer 3, she will transition to $s_8$ and $s_9$ in layer 4 with probabilities 0.05 and 0.95, respectively. Otherwise, if she starts from state $s_5$ in layer 3, she will transition to $s_{10}$ and $s_{11}$ in layer 4 with probabilities 0.05 and 0.95, respectively.

On the other hand, if the agent goes into the right sub-tree (state $s_3$) in layer 2, she will transition to $s_6$ and $s_7$ in layer 3 with probabilities 0.01 and 0.99, respectively. Then, if she starts from state $s_6$ in layer 3, she will transition to $s_{12}$ and $s_{13}$ in layer 4 with probabilities 0.01 and 0.99, respectively. Otherwise, if she starts from state $s_7$ in layer 3, she will transition to $s_{14}$ and $s_{15}$ in layer 4 with probabilities 0.01 and 0.99, respectively.

The costs are state-dependent, and only the states in layer 4 produce non-zero costs. To be concrete, we use the clinical trial example and the costs represent the patient status. Specifically, in layer 4, $s_8$ and $s_{12}$ give costs 1, which denote *death*. $s_{13}$ and $s_{14}$ produce costs 0.5, which means the patient is *getting better*. $s_9$ and $s_{10}$ induce costs 0.4, which denote that the patient *gets much better*. $s_{11}$ and $s_{15}$ produce costs 0, which stand for that the patient is *fully cured*.

Under the CVaR criterion, we have that

$$Q^{\text{CVaR},\alpha}(s_1, a_1) = \frac{0.0025}{0.05} \cdot 1 + \frac{0.05 - 0.0025}{0.05} \cdot 0.4 = 0.43,$$

and

$$Q^{\text{CVaR},\alpha}(s_1, a_2) = \frac{0.0001}{0.05} \cdot 1 + \frac{0.05 - 0.0001}{0.05} \cdot 0.5 = 0.501.$$

Thus, CVaR MDP will choose action $a_1$ (and goes into the left sub-tree), since $a_1$ leads to better medium states $s_9$ and $s_{10}$, which give a lower cost 0.4 than the cost 0.5 produced by the right sub-tree.

On the other hand, under the Iterated CVaR criterion, we have that

$$Q^{\text{ICVaR},\alpha}(s_1, a_1) = \frac{0.05}{0.05} \cdot Q^{\text{ICVaR},\alpha}(s_4, \cdot) = \frac{0.05}{0.05} \cdot \left( \frac{0.05}{0.05} \cdot Q^{\text{ICVaR},\alpha}(s_8, \cdot) \right) = \frac{0.05}{0.05} \cdot \left( \frac{0.05}{0.05} \cdot 1 \right) = 1,$$

and

$$
\begin{aligned}
&Q^{\text{ICVaR},\alpha}(s_1, a_2) \\
=& \frac{0.01}{0.05} \cdot Q^{\text{ICVaR},\alpha}(s_6, \cdot) + \frac{0.05 - 0.01}{0.05} \cdot Q^{\text{ICVaR},\alpha}(s_7, \cdot) \\
=& \frac{0.01}{0.05} \cdot \left( \frac{0.01}{0.05} \cdot Q^{\text{ICVaR},\alpha}(s_{12}, \cdot) + \frac{0.05 - 0.01}{0.05} \cdot Q^{\text{ICVaR},\alpha}(s_{13}, \cdot) \right) \\
&+ \frac{0.05 - 0.01}{0.05} \cdot \left( \frac{0.01}{0.05} \cdot Q^{\text{ICVaR},\alpha}(s_{14}, \cdot) + \frac{0.05 - 0.01}{0.05} \cdot Q^{\text{ICVaR},\alpha}(s_{15}, \cdot) \right) \\
=& \frac{0.01}{0.05} \cdot \left( \frac{0.01}{0.05} \cdot 1 + \frac{0.05 - 0.01}{0.05} \cdot 0.5 \right) + \frac{0.05 - 0.01}{0.05} \cdot \left( \frac{0.01}{0.05} \cdot 0.5 + \frac{0.05 - 0.01}{0.05} \cdot 0 \right) \\
=& 0.2.
\end{aligned}
$$

Thus, Iterated CVaR RL will instead choose action $a_2$, because $a_2$ has a smaller probability of going into the bad left direction (which leads to the catastrophic state $s_{12}$).

The above example shows that, Iterated CVaR RL prefers actions with a smaller probability of getting into catastrophic states. In contrast, CVaR MDP favors actions with better average therapeutic effects, but has a larger probability of causing death.

Note that the above example also demonstrates that Iterated CVaR is *not equivalent* to the worst $\alpha^H$-portion of the total cost. To see this, we have that (here we consider $\alpha^3$ because there are 3 transition steps):

$$Q^{\text{CVaR},\alpha^3}(s_1, a_2) = \frac{0.0001}{0.000125} \cdot 1 + \frac{0.000125 - 0.0001}{0.000125} \cdot 0.5 = 0.9,$$

and

$$Q^{\text{ICVaR},\alpha}(s_1, a_2)$$
$$= \frac{0.01}{0.05} \cdot \left( \frac{0.01}{0.05} \cdot 1 + \frac{0.05 - 0.01}{0.05} \cdot 0.5 \right) + \frac{0.05 - 0.01}{0.05} \cdot \left( \frac{0.01}{0.05} \cdot 0.5 + \frac{0.05 - 0.01}{0.05} \cdot 0 \right)$$
$$= 0.2.$$

In addition, one can see that, the good state which gives cost 0 (i.e., $s_{15}$) also contributes to $Q^{\text{ICVaR},\alpha}(s_1, a_2)$, which shows that the good $(1 - \alpha^H)$-portion of the total cost also matters for Iterated CVaR.

### C.3 COMPARISON WITH EXPONENTIAL UTILITY-BASED RISK-SENSITIVE RL

The Bellman optimality equation for risk-sensitive RL with the exponential utility criterion (Fei et al., 2020; 2021a) is defined as

$$Q_h^*(s, a) = r_h(s, a) + \frac{1}{\beta} \log\{\mathbb{E}_{s' \sim p(\cdot|s,a)}[\exp(\beta \cdot V_{h+1}^*(s'))]\},$$

which takes all successor states $s'$ into account, i.e., all successor states $s'$ contribute to the computation of the Q-value. Here $\beta < 0$ is a risk-sensitivity parameter.

In contrast, in Iterated CVaR RL, the Bellman optimality equation is defined as

$$Q_h^*(s, a) = r(s, a) + \text{CVaR}_{s' \sim p(\cdot|s,a)}^\alpha(V_{h+1}^*(s')),$$

which focuses only on the worst $\alpha$-portion successor states $s'$ (i.e., with the lowest $\alpha$-portion values $V_{h+1}^*(s')$), i.e., only the worst $\alpha$-portion successor states $s'$ contribute to the computation of the Q-value.

Besides the formulation, our algorithm design and results are also very different from those in (Fei et al., 2020; 2021a). The algorithms in (Fei et al., 2020; 2021a) are based on exponential Bellman equations and doubly decaying exploration bonuses, and their results depend on $\exp(|\beta|H)$. In contrast, our algorithms are based on value iteration for Iterated CVaR with CVaR-adapted exploration bonuses, and our results depend on the minimum between an MDP-intrinsic visitation measure $1/\min_{\pi,h,s: w_{\pi,h}(s)>0} w_{\pi,h}(s)$ and a risk-level-dependent factor $1/\alpha^{H-1}$.

## D PROOFS FOR ITERATED CVAR RL WITH REGRET MINIMIZATION

In this section, we present the proofs of regret upper and lower bounds (Theorems 1 and 2) for Iterated CVaR RL-RM.

### D.1 PROOFS OF REGRET UPPER BOUND

#### D.1.1 CONCENTRATION

For any $k > 0$, $h \in [H]$ and $(s, a) \in \mathcal{S} \times \mathcal{A}$, let $n_{kh}(s, a)$ denote the number of times that $(s, a)$ was visited at step $h$ before episode $k$, and let $n_k(s, a) := \sum_{h=1}^{H} n_{kh}(s, a)$ denote the number of times that $(s, a)$ was visited before episode $k$.

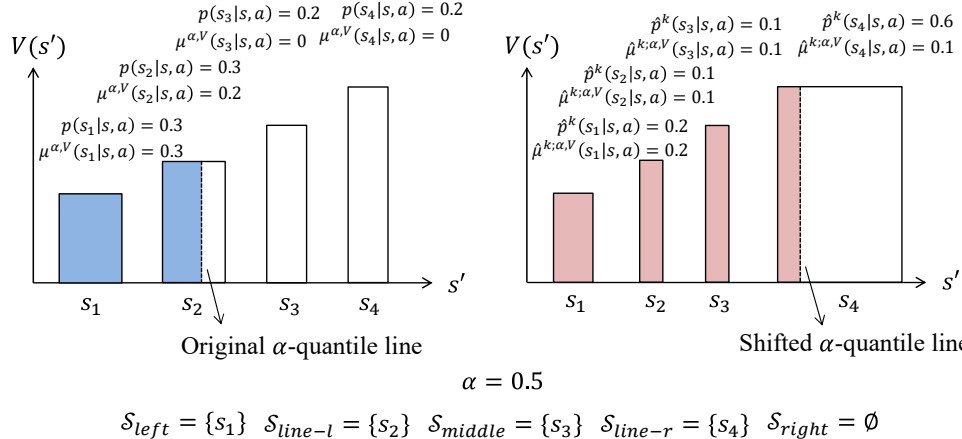

$$\alpha = 0.5$$

$$\mathcal{S}_{left} = \{s_1\} \quad \mathcal{S}_{line-l} = \{s_2\} \quad \mathcal{S}_{middle} = \{s_3\} \quad \mathcal{S}_{line-r} = \{s_4\} \quad \mathcal{S}_{right} = \emptyset$$

Figure 4: Illustrating example for Lemma 2. For each $s' \in \{s_1, s_2, s_3, s_4\}$, the height of the bar denotes the value $V(s')$ (fixed), and the width of the bar denotes the transition probability $p(s'|s, a)$ or $\hat{p}^k(s'|s, a)$. The colored part of the bars denotes the worst $\alpha$-portion successor states (i.e., with the lowest $\alpha$-portion values $V(s')$). In this example, $\alpha = 0.5$.

**Lemma 1** (Concentration for $V^*$). *It holds that*

$$\Pr\left[\left|\text{CVaR}^\alpha_{s'\sim\hat{p}^k(\cdot|s,a)}(V_h^*(s')) - \text{CVaR}^\alpha_{s'\sim p(\cdot|s,a)}(V_h^*(s'))\right| \leq \frac{H}{\alpha}\sqrt{\frac{\log\left(\frac{KHSA}{\delta'}\right)}{n_k(s,a)}},\right.$$

$$\left.\forall k \in [K],\ \forall h \in [H],\ \forall (s,a) \in \mathcal{S} \times \mathcal{A}\right] \geq 1 - 2\delta'.$$

*Proof of Lemma 1.* Using Brown's inequality (Brown, 2007) (Theorem 2 in (Thomas & Learned-Miller, 2019)) and a union bound over $(s, a) \in \mathcal{S} \times \mathcal{A}$ and $n_k(s, a) \in [KH]$, we can obtain this lemma. $\square$

For any risk level $\alpha \in (0, 1]$, function $V : \mathcal{S} \mapsto \mathbb{R}$ and $(s', s, a) \in \mathcal{S} \times \mathcal{S} \times \mathcal{A}$, $\beta^{\alpha,V}(s'|s, a)$ is the conditional transition probability from $(s, a)$ to $s'$, conditioning on transitioning to the worst $\alpha$-portion successor states $s'$ (i.e., with the lowest $\alpha$-portion values $V(s')$). Let $\mu^{\alpha,V}(s'|s, a)$ denote how large the transition probability of successor state $s'$ belongs to the worst $\alpha$-portion, which satisfies that $\frac{\mu^{\alpha,V}(s'|s,a)}{\alpha} = \beta^{\alpha,V}(s'|s, a)$ and $\sum_{s'\in\mathcal{S}}\mu^{\alpha,V}(s'|s, a) = \alpha$. In addition, for any risk level $\alpha \in (0, 1]$, function $V : \mathcal{S} \mapsto \mathbb{R}$ and $(s, a) \in \mathcal{S} \times \mathcal{A}$,

$$\text{CVaR}^\alpha_{s'\sim p(\cdot|s,a)}(V(s')) = \frac{\sum_{s'\in\mathcal{S}}\mu^{\alpha,V}(s'|s,a)\cdot V(s')}{\alpha} = \sum_{s'\in\mathcal{S}}\beta^{\alpha,V}(s'|s,a)\cdot V(s').$$

**Lemma 2** (Concentration for any $V$). *It holds that*

$$\left[\left|\text{CVaR}^\alpha_{s'\sim\hat{p}^k(\cdot|s,a)}(V(s')) - \text{CVaR}^\alpha_{s'\sim p(\cdot|s,a)}(V(s'))\right| \leq \frac{2H}{\alpha}\sqrt{\frac{2S\log\left(\frac{KHSA}{\delta'}\right)}{n_k(s,a)}},\right.$$

$$\left.\forall V : \mathcal{S} \mapsto [0, H],\ \forall k \in [K],\ \forall (s,a) \in \mathcal{S} \times \mathcal{A}\right] \geq 1 - 2\delta'.$$

*Proof of Lemma 2.* As shown in Figure 4, we sort all successor states $s' \in \mathcal{S}$ by $V(s')$ in ascending order (from the left to the right). Add a virtual line at the $\alpha$-quantile, denoted by $\alpha$-*quantile line*. Fix the value function $V(\cdot)$, and the transition probability changes from $p(\cdot|s, a)$ to $\hat{p}^k(\cdot|s, a)$.

Without loss of generality, below we consider the case where as the transition probability changes from $p(\cdot|s, a)$ to $\hat{p}^k(\cdot|s, a)$, the $\alpha$-quantile line shifts from left to right (the analysis of the contrary

case can also be obtained by interchanging $p(\cdot|s,a)$ and $\hat{p}^k(\cdot|s,a)$). We use *original $\alpha$-quantile line* and *shifted $\alpha$-quantile line* to denote the $\alpha$-quantile line before and after the shift, respectively.

We divide the successor states $s' \in \mathcal{S}$ into five subsets as follows. Let $\mathcal{S}_{left}$ and $\mathcal{S}_{right}$ denote the sets of states which are always on the left and right sides of the original and shifted $\alpha$-quantile lines, respectively. Let $\mathcal{S}_{middle}$ denote the set of states which are in the middle of the original and shifted $\alpha$-quantile lines. Let $s_{line\text{-}l}$ and $s_{line\text{-}r}$ denote the states which lie on the original and shifted $\alpha$-quantile lines, respectively.

For any $s' \in \mathcal{S}_{left}$, we have that $\mu^{\alpha,V}(s'|s,a) = p(s'|s,a)$ and $\hat{\mu}^{k;\alpha,V}(s'|s,a) = \hat{p}^k(s'|s,a)$.

For any $s' \in \mathcal{S}_{right}$, we have $\mu^{\alpha,V}(s'|s,a) = \hat{\mu}^{k;\alpha,V}(s'|s,a) = 0$.

For any $s' \in \mathcal{S}_{middle}$, we have that $\mu^{\alpha,V}(s'|s,a) = 0$ and $\hat{\mu}^{k;\alpha,V}(s'|s,a) = \hat{p}^k(s'|s,a)$.

For state $s_{line\text{-}l}$, we have that $\mu^{\alpha,V}(s_{line\text{-}l}|s,a) = p(s_{line\text{-}l}|s,a) - (\sum_{s'\in\mathcal{S}_{left}} p(s'|s,a) + p(s_{line\text{-}l}|s,a) - \alpha)$ and $\hat{\mu}^{k;\alpha,V}(s_{line\text{-}l}|s,a) = \hat{p}^k(s_{line\text{-}l}|s,a)$.

For state $s_{line\text{-}r}$, we have that $\mu^{\alpha,V}(s_{line\text{-}r}|s,a) = 0$ and $\hat{\mu}^{k;\alpha,V}(s_{line\text{-}r}|s,a) = \alpha - \sum_{s'\in\mathcal{S}_{left}} \hat{p}^k(s'|s,a) - \sum_{s'\in\mathcal{S}_{middle}} \hat{p}^k(s'|s,a) - \hat{p}^k(s_{line\text{-}l}|s,a)$.

Then, we obtain

$$
\sum_{s'\in\mathcal{S}} \left| \hat{\mu}^{k;\alpha,V}(s'|s,a) - \mu^{\alpha,V}(s'|s,a) \right|
$$

$$
\leq \sum_{s'\in\mathcal{S}_{left}} \left| \hat{\mu}^{k;\alpha,V}(s'|s,a) - \mu^{\alpha,V}(s'|s,a) \right| + \sum_{s'\in\mathcal{S}_{right}} \left| \hat{\mu}^{k;\alpha,V}(s'|s,a) - \mu^{\alpha,V}(s'|s,a) \right|
$$

$$
+ \sum_{s'\in\mathcal{S}_{middle}} \left| \hat{\mu}^{k;\alpha,V}(s'|s,a) - \mu^{\alpha,V}(s'|s,a) \right| + \left| \hat{\mu}^{k;\alpha,V}(s_{line\text{-}l}|s,a) - \mu^{\alpha,V}(s_{line\text{-}l}|s,a) \right|
$$

$$
+ \left| \hat{\mu}^{k;\alpha,V}(s_{line\text{-}r}|s,a) - \mu^{\alpha,V}(s_{line\text{-}r}|s,a) \right|
$$

$$
\leq \sum_{s'\in\mathcal{S}_{left}} \left| \hat{p}^k(s'|s,a) - p(s'|s,a) \right| + \sum_{s'\in\mathcal{S}_{middle}} \hat{p}^k(s'|s,a)
$$

$$
+ \left| \hat{p}^k(s_{line\text{-}l}|s,a) - p(s_{line\text{-}l}|s,a) + \left( \sum_{s'\in\mathcal{S}_{left}} p(s'|s,a) + p(s_{line\text{-}l}|s,a) - \alpha \right) \right|
$$

$$
+ \left( \alpha - \sum_{s'\in\mathcal{S}_{left}} \hat{p}^k(s'|s,a) - \sum_{s'\in\mathcal{S}_{middle}} \hat{p}^k(s'|s,a) - \hat{p}^k(s_{line\text{-}l}|s,a) \right)
$$

$$
\leq \sum_{s'\in\mathcal{S}_{left}} \left| \hat{p}^k(s'|s,a) - p(s'|s,a) \right| + \sum_{s'\in\mathcal{S}_{middle}} \hat{p}^k(s'|s,a)
$$

$$
+ \left| \hat{p}^k(s_{line\text{-}l}|s,a) - p(s_{line\text{-}l}|s,a) \right| + \left| \sum_{s'\in\mathcal{S}_{left}} p(s'|s,a) + p(s_{line\text{-}l}|s,a) - \alpha \right|
$$

$$
+ \left( \alpha - \sum_{s'\in\mathcal{S}_{left}} \hat{p}^k(s'|s,a) - \sum_{s'\in\mathcal{S}_{middle}} \hat{p}^k(s'|s,a) - \hat{p}^k(s_{line\text{-}l}|s,a) \right)
$$

$$
\overset{(a)}{\leq} \sum_{s'\in\mathcal{S}_{left}} \left| \hat{p}^k(s'|s,a) - p(s'|s,a) \right| + \sum_{s'\in\mathcal{S}_{middle}} \hat{p}^k(s'|s,a)
$$

$$
+ \left| \hat{p}^k(s_{line\text{-}l}|s,a) - p(s_{line\text{-}l}|s,a) \right| + \left( \sum_{s'\in\mathcal{S}_{left}} p(s'|s,a) + p(s_{line\text{-}l}|s,a) - \alpha \right)
$$

$$
+ \left( \alpha - \sum_{s'\in\mathcal{S}_{left}} \hat{p}^k(s'|s,a) - \sum_{s'\in\mathcal{S}_{middle}} \hat{p}^k(s'|s,a) - \hat{p}^k(s_{line\text{-}l}|s,a) \right)
$$

$$
\begin{aligned}
&= \sum_{s' \in \mathcal{S}_{left}} \left| \hat{p}^k(s'|s,a) - p(s'|s,a) \right| + \left| \hat{p}^k(s_{line\text{-}l}|s,a) - p(s_{line\text{-}l}|s,a) \right| \\
&\quad + \sum_{s' \in \mathcal{S}_{left}} p(s'|s,a) - \sum_{s' \in \mathcal{S}_{left}} \hat{p}^k(s'|s,a) + p(s_{line\text{-}l}|s,a) - \hat{p}^k(s_{line\text{-}l}|s,a) \\
&\leq \sum_{s' \in \mathcal{S}_{left}} \left| \hat{p}^k(s'|s,a) - p(s'|s,a) \right| + \left| \hat{p}^k(s_{line\text{-}l}|s,a) - p(s_{line\text{-}l}|s,a) \right| \\
&\quad + \sum_{s' \in \mathcal{S}_{left}} \left| p(s'|s,a) - \hat{p}^k(s'|s,a) \right| + \left| p(s_{line\text{-}l}|s,a) - \hat{p}^k(s_{line\text{-}l}|s,a) \right| \\
&\leq 2 \sum_{s' \in \mathcal{S}} \left| \hat{p}^k(s'|s,a) - p(s'|s,a) \right|,
\end{aligned}
\tag{7}
$$

where (a) is due to $\sum_{s' \in \mathcal{S}_{left}} p(s'|s,a) + p(s_{line\text{-}l}|s,a) - \alpha \geq 0$ by the definition of state $s_{line\text{-}l}$.

Thus, we have

$$
\begin{aligned}
&\left| \mathrm{CVaR}^\alpha_{s' \sim \hat{p}^k(\cdot|s,a)}(V(s')) - \mathrm{CVaR}^\alpha_{s' \sim p(\cdot|s,a)}(V(s')) \right| \\
&= \left| \frac{\sum_{s' \in \mathcal{S}} \hat{\mu}^{k;\alpha,V}(s'|s,a) \cdot V(s')}{\alpha} - \frac{\sum_{s' \in \mathcal{S}} \mu^{\alpha,V}(s'|s,a) \cdot V(s')}{\alpha} \right| \\
&= \frac{\left| \sum_{s' \in \mathcal{S}} \left( \hat{\mu}^{k;\alpha,V}(s'|s,a) - \mu^{\alpha,V}(s'|s,a) \right) \cdot V(s') \right|}{\alpha} \\
&\leq \frac{\sum_{s' \in \mathcal{S}} \left| \hat{\mu}^{k;\alpha,V}(s'|s,a) - \mu^{\alpha,V}(s'|s,a) \right| \cdot H}{\alpha} \\
&\leq \frac{2 \sum_{s' \in \mathcal{S}} \left| p^k(s'|s,a) - p(s'|s,a) \right| \cdot H}{\alpha}
\end{aligned}
\tag{8}
$$

Using Eq. (55) in (Zanette & Brunskill, 2019) (originated from (Weissman et al., 2003)), we have that with probability at least $1 - 2\delta'$, for any $k \in [K]$ and $(s,a) \in \mathcal{S} \times \mathcal{A}$,

$$
\sum_{s' \in \mathcal{S}} \left| \hat{p}^k(s'|s,a) - p(s'|s,a) \right| \leq \sqrt{\frac{2S \log\left( \frac{KHSA}{\delta'} \right)}{n_k(s,a)}}.
\tag{9}
$$

Plugging Eq. (9) into Eq. (8), we obtain that with probability at least $1 - 2\delta'$, for any $k \in [K]$, $(s,a) \in \mathcal{S} \times \mathcal{A}$ and function $V : \mathcal{S} \mapsto [0, H]$,

$$
\left| \mathrm{CVaR}^\alpha_{s' \sim \hat{p}^k(\cdot|s,a)}(V(s')) - \mathrm{CVaR}^\alpha_{s' \sim p(\cdot|s,a)}(V(s')) \right| \leq \frac{2H}{\alpha} \sqrt{\frac{2S \log\left( \frac{KHSA}{\delta'} \right)}{n_k(s,a)}}.
$$

$\square$

For any $k > 0$, $h \in [H]$ and $(s,a) \in \mathcal{S} \times \mathcal{A}$, let $w_{kh}(s,a)$ denote the probability of visiting $(s,a)$ at step $h$ of episode $k$. Then, it holds that for any $k > 0$, $h \in [H]$ and $(s,a) \in \mathcal{S} \times \mathcal{A}$, $w_{kh}(s,a) \in [0,1]$ and $\sum_{(s,a) \in \mathcal{S} \times \mathcal{A}} w_{kh}(s,a) = 1$.

**Lemma 3** (Concentration of Visitation). *It holds that*

$$
\Pr\left[ n_k(s,a) \geq \frac{1}{2} \sum_{k'=1}^{k-1} \sum_{h=1}^{H} w_{k'h}(s,a) - H \log\left( \frac{HSA}{\delta'} \right), \; \forall k > 0, \; \forall (s,a) \in \mathcal{S} \times \mathcal{A} \right] \geq 1 - \delta'.
$$

*Proof of Lemma 3.* Applying Lemma F.4 in (Dann et al., 2017), we have that for any fixed $h \in [H]$,

$$
\Pr\left[ n_{kh}(s,a) \geq \frac{1}{2} \sum_{k'=1}^{k-1} w_{k'h}(s,a) - \log\left( \frac{HSA}{\delta'} \right), \; \forall k > 0, \; \forall (s,a) \in \mathcal{S} \times \mathcal{A} \right] \geq 1 - \frac{\delta'}{H}
$$

By a union bound over $h \in [H]$, we have

$$\Pr \left[ n_k(s,a) \geq \frac{1}{2} \sum_{k'=1}^{k-1} \sum_{h=1}^{H} w_{k'h}(s,a) - H \log \left( \frac{HSA}{\delta'} \right) \right] \geq 1 - \delta'.$$

$\square$

To sum up, we define several concentration events which will be used in the following proof.

$$\mathcal{E}_1 := \left\{ \left| \text{CVaR}^{\alpha}_{s' \sim \hat{p}^k(\cdot|s,a)}(V_h^*(s')) - \text{CVaR}^{\alpha}_{s' \sim p(\cdot|s,a)}(V_h^*(s')) \right| \leq \frac{H}{\alpha} \sqrt{\frac{\log \left( \frac{KHSA}{\delta'} \right)}{n_k(s,a)}}, \right.$$

$$\left. \forall k \in [K], \ \forall h \in [H], \ \forall (s,a) \in \mathcal{S} \times \mathcal{A} \right\}$$

$$\mathcal{E}_2 := \left\{ \left| \text{CVaR}^{\alpha}_{s' \sim \hat{p}^k(\cdot|s,a)}(V(s')) - \text{CVaR}^{\alpha}_{s' \sim p(\cdot|s,a)}(V(s')) \right| \leq \frac{2H}{\alpha} \sqrt{\frac{2S \log \left( \frac{KHSA}{\delta'} \right)}{n_k(s,a)}}, \right.$$

$$\left. \forall V : \mathcal{S} \mapsto [0,H], \ \forall k \in [K], \ \forall (s,a) \in \mathcal{S} \times \mathcal{A} \right\}$$

$$\mathcal{E}_3 := \left\{ n_k(s,a) \geq \frac{1}{2} \sum_{k'=1}^{k-1} \sum_{h=1}^{H} w_{k'h}(s,a) - H \log \left( \frac{HSA}{\delta'} \right), \ \forall k > 0, \ \forall (s,a) \in \mathcal{S} \times \mathcal{A} \right\}$$

$$\mathcal{E} := \mathcal{E}_1 \cap \mathcal{E}_2 \cap \mathcal{E}_3$$

**Lemma 4.** *Letting $\delta' = \frac{\delta}{5}$, it holds that*

$$\Pr[\mathcal{E}] \geq 1 - \delta.$$

*Proof of Lemma 4.* This lemma can be obtained by combining Lemmas 1-3. $\square$

### D.1.2 OPTIMISM, VISITATION AND CVAR GAP

Recall that $L := \log \left( \frac{KHSA}{\delta'} \right)$.

**Lemma 5** (Optimism). *Suppose that event $\mathcal{E}$ holds. Then, for any $k \in [K]$, $h \in [H]$ and $s \in \mathcal{S}$, we have*

$$\bar{V}_h^k(s) \geq V_h^*(s).$$

*Proof of Lemma 5.* We prove this lemma by induction.

First, for any $k \in [K]$, $s \in \mathcal{S}$, it holds that $\bar{V}_{H+1}^k(s) = V_{H+1}^*(s) = 0$.

Then, for any $k \in [K]$, $h \in [H]$ and $(s,a) \in \mathcal{S} \times \mathcal{A}$, if $\bar{Q}_h^k(s,a) = H$, $\bar{Q}_h^k(s,a) \geq Q_h^*(s,a)$ trivially holds, and otherwise,

$$\bar{Q}_h^k(s,a) = r(s,a) + \text{CVaR}^{\alpha}_{s' \sim \hat{p}^k(\cdot|s,a)}(\bar{V}_{h+1}^k(s')) + \frac{H}{\alpha} \sqrt{\frac{L}{n_k(s,a)}}$$

$$\overset{(a)}{\geq} r(s,a) + \text{CVaR}^{\alpha}_{s' \sim \hat{p}^k(\cdot|s,a)}(V_{h+1}^*(s')) + \frac{H}{\alpha} \sqrt{\frac{L}{n_k(s,a)}}$$

$$\overset{(b)}{\geq} r(s,a) + \text{CVaR}^{\alpha}_{s' \sim p(\cdot|s,a)}(V_{h+1}^*(s'))$$

$$= Q_h^*(s,a),$$

where (a) uses the induction hypothesis and (b) comes from Lemma 1.

Thus, we have

$$\bar{V}_h^k(s) \geq \bar{Q}_h^k(s, \pi_h^*(s)) \geq Q_h^*(s, \pi_h^*(s)) = V_h^*(s),$$

which concludes the proof. $\square$

Following (Zanette & Brunskill, 2019), for any episode $k > 0$, we define the set of state-action pairs which have sufficient visitations in expectation as follows.

$$\mathcal{L}_k := \left\{ (s,a) \in \mathcal{S} \times \mathcal{A} : \frac{1}{4} \sum_{k'=1}^{k-1} \sum_{h=1}^{H} w_{k'h}(s,a) \geq H \log\left(\frac{HSA}{\delta'}\right) + H \right\}. \tag{10}$$

**Lemma 6** (Sufficient Visitation). *Suppose that event $\mathcal{E}$ holds. Then, for any $k > 0$ and $(s,a) \in \mathcal{L}_k$,*

$$n_k(s,a) \geq \frac{1}{4} \sum_{k'=1}^{k} \sum_{h=1}^{H} w_{k'h}(s,a).$$

*Proof of Lemma 6.* This proof is the same as that of Lemma 6 in (Zanette & Brunskill, 2019).

Using Lemma 3, we have

$$
\begin{aligned}
n_k(s,a) \geq & \frac{1}{2} \sum_{k'=1}^{k-1} \sum_{h=1}^{H} w_{k'h}(s,a) - H \log\left(\frac{HSA}{\delta'}\right) \\
= & \frac{1}{4} \sum_{k'=1}^{k-1} \sum_{h=1}^{H} w_{k'h}(s,a) + \frac{1}{4} \sum_{k'=1}^{k-1} \sum_{h=1}^{H} w_{k'h}(s,a) - H \log\left(\frac{HSA}{\delta'}\right) \\
\overset{(a)}{\geq} & \frac{1}{4} \sum_{k'=1}^{k-1} \sum_{h=1}^{H} w_{k'h}(s,a) + H \\
\overset{(b)}{\geq} & \frac{1}{4} \sum_{k'=1}^{k-1} \sum_{h=1}^{H} w_{k'h}(s,a) + \sum_{h=1}^{H} w_{kh}(s,a) \\
= & \frac{1}{4} \sum_{k'=1}^{k} \sum_{h=1}^{H} w_{k'h}(s,a)
\end{aligned}
$$

where (a) uses the fact that $(s,a) \in \mathcal{L}_k$ and the definition of $\mathcal{L}_k$, and (b) is due to that for any $k > 0$, $h \in [H]$ and $(s,a) \in \mathcal{S} \times \mathcal{A}$, $w_{kh}(s,a) \in [0,1]$. $\square$

**Lemma 7** (Standard Visitation Ratio). *For any $K > 0$, we have*

$$\sqrt{\sum_{k=1}^{K} \sum_{h=1}^{H} \sum_{(s,a) \in \mathcal{L}_k} \frac{w_{kh}(s,a)}{n_k(s,a)}} \leq 2\sqrt{SA \log\left(\frac{KHSA}{\delta'}\right)}.$$

*Proof of Lemma 7.* This proof is the same as that of Lemma 13 in (Zanette & Brunskill, 2019).

Recall that for any $k > 0$, let $w_k(s,a) := \sum_{(s,a) \in \mathcal{S} \times \mathcal{A}} w_{kh}(s,a)$. Then, we have

$$
\begin{aligned}
\sqrt{\sum_{k=1}^{K} \sum_{h=1}^{H} \sum_{(s,a) \in \mathcal{L}_k} \frac{w_{kh}(s,a)}{n_k(s,a)}} = & \sqrt{\sum_{k=1}^{K} \sum_{(s,a) \in \mathcal{L}_k} \frac{w_k(s,a)}{n_k(s,a)}} \\
= & \sqrt{\sum_{k=1}^{K} \sum_{(s,a) \in \mathcal{S} \times \mathcal{A}} \frac{w_k(s,a)}{n_k(s,a)} \cdot \mathbb{1}\left\{(s,a) \in \mathcal{L}_k\right\}} \\
\overset{(a)}{\leq} & 2\sqrt{\sum_{k=1}^{K} \sum_{(s,a) \in \mathcal{S} \times \mathcal{A}} \frac{w_k(s,a)}{\sum_{k'=1}^{k} w_{k'}(s,a)} \cdot \mathbb{1}\left\{(s,a) \in \mathcal{L}_k\right\}} \\
= & 2\sqrt{\sum_{(s,a) \in \mathcal{S} \times \mathcal{A}} \sum_{k=1}^{K} \frac{w_k(s,a)}{\sum_{k'=1}^{k} w_{k'}(s,a)} \cdot \mathbb{1}\left\{(s,a) \in \mathcal{L}_k\right\}}
\end{aligned}
$$

where (a) is due to Lemma 6.

According to the definition of $\mathcal{L}_k$ (Eq. (10)), for any $(s,a) \in \mathcal{S} \times \mathcal{A}$, once $(s,a)$ satisfies $(s,a) \in \mathcal{L}_k$ in some episode $k$, it will always satisfy $(s,a) \in \mathcal{L}_{k'}$ for all $k' \geq k$. For any $(s,a) \in \mathcal{S} \times \mathcal{A}$, let $k_0(s,a)$ denote the first episode $k$ where $(s,a) \in \mathcal{L}_k$.

Then, for any $k > 0$ and $(s,a) \in \mathcal{S} \times \mathcal{A}$, if $(s,a) \in \mathcal{L}_k$, we have

$$\sum_{k'=1}^{k} w_{k'}(s,a) = \sum_{k'=1}^{k_0(s,a)-1} w_{k'}(s,a) + \sum_{k'=k_0(s,a)}^{k} w_{k'}(s,a)$$

$$\overset{(a)}{\geq} H + \sum_{k'=k_0(s,a)}^{k} w_{k'}(s,a),$$

where (a) uses the fact that $(s,a) \in \mathcal{L}_{k_0(s,a)}$ and the definition of $\mathcal{L}_k$ (Eq. (10)).

Thus, we have

$$\sqrt{\sum_{k=1}^{K} \sum_{h=1}^{H} \sum_{(s,a) \in \mathcal{L}_k} \frac{w_{kh}(s,a)}{n_k(s,a)}} \leq 2 \sqrt{\sum_{(s,a) \in \mathcal{S} \times \mathcal{A}} \sum_{k=k_0(s,a)}^{K} \frac{w_k(s,a)}{H + \sum_{k'=k_0(s,a)}^{k} w_{k'}(s,a)}}.$$

Let $a_1 := w_{k_0(s,a)}(s,a)$, $a_2 := w_{k_0(s,a)+1}(s,a)$, $\ldots$, $a_{K-k_0(s,a)+1} := w_K(s,a)$. Define function $F(x) = \sum_{i=1}^{\lfloor x \rfloor} a_i + a_{\lceil x \rceil}(x - \lfloor x \rfloor)$, where $0 \leq x \leq K - k_0(s,a) + 1$. If $x$ is an integer, we have $F(x) = \sum_{i=1}^{x} a_i$, and otherwise, $F(x)$ interpolates between the function values for integers $x$. The derivative of $F(s)$ is $f(x) = a_{\lceil x \rceil}$.

Hence, we have

$$\sum_{k=k_0(s,a)}^{K} \frac{w_k(s,a)}{H + \sum_{k'=k_0(s,a)}^{k} w_{k'}(s,a)} = \sum_{k=1}^{K-k_0(s,a)+1} \frac{f(k)}{H + F(k)}$$

$$= \int_{0}^{K-k_0(s,a)+1} \frac{f(\lceil x \rceil)}{H + F(\lceil x \rceil)} dx$$

$$\overset{(a)}{\leq} \int_{0}^{K-k_0(s,a)+1} \frac{f(x)}{H + F(x)} dx$$

$$= \log\left(H + F(K - k_0(s,a) + 1)\right) - \log\left(H + F(0)\right)$$

$$\overset{(b)}{\leq} \log(KH)$$

$$\leq \log\left(\frac{KHSA}{\delta'}\right),$$

where (a) uses the fact that for any $0 \leq x \leq K - k_0(s,a) + 1$, $f(x) = f(\lceil x \rceil)$ and $F(x) \leq F(\lceil x \rceil)$, and (b) is due to that $k_0(s,a) \geq 2$ by the definitions of $\mathcal{L}_k$ (Eq. (10)) and $k_0(s,a)$.

Therefore, we have

$$\sqrt{\sum_{k=1}^{K} \sum_{h=1}^{H} \sum_{(s,a) \in \mathcal{L}_k} \frac{w_{kh}(s,a)}{n_k(s,a)}} \leq 2 \sqrt{\sum_{(s,a) \in \mathcal{S} \times \mathcal{A}} \log\left(\frac{KHSA}{\delta'}\right)}$$

$$\leq 2 \sqrt{SA \log\left(\frac{KHSA}{\delta'}\right)}$$

$\square$

Recall that for any $(s',s,a) \in \mathcal{S} \times \mathcal{S} \times \mathcal{A}$, $p(s'|s,a)$ is the transition probability from $(s,a)$ to $s'$. For any risk level $\alpha \in (0,1]$, function $V : \mathcal{S} \mapsto \mathbb{R}$ and $(s',s,a) \in \mathcal{S} \times \mathcal{S} \times \mathcal{A}$, $\beta^{\alpha,V}(s'|s,a)$ is the conditional probability of transitioning to $s'$ from $(s,a)$, conditioning on transitioning to the

worst $\alpha$-portion successor states $s'$ (i.e., with the lowest $\alpha$-portion values $V(s')$), and it holds that $\text{CVaR}^{\alpha}_{s' \sim p(s'|s,a)}(V(s')) = \sum_{s' \in \mathcal{S}} \beta^{\alpha,V}(s'|s,a) \cdot V(s')$.

For any $k > 0$, $h \in [H]$ and $(s,a) \in \mathcal{S} \times \mathcal{A}$, $w_{kh}(s,a)$ is the probability of visiting $(s,a)$ at step $h$ of episode $k$ (under transition probability $p(\cdot|\cdot,\cdot)$), and it holds that $w_{kh}(s,a) \in [0,1]$ and $\sum_{(s,a) \in \mathcal{S} \times \mathcal{A}} w_{kh}(s,a) = 1$. For any risk level $\alpha \in (0,1]$, $k > 0$, $h \in [H]$ and $(s,a) \in \mathcal{S} \times \mathcal{A}$, $w_{kh}^{CVaR,\alpha,V^{\pi^k}}(s,a)$ is the conditional probability of visiting $(s,a)$ at step $h$ of episode $k$, conditioning on transitioning to the worst $\alpha$-portion successor states $s'$ (i.e., with the lowest $\alpha$-portion values $V^{\pi^k}_{h'+1}(s')$) at each step $h' = 1, \ldots, h-1$. Here $\pi^k$ is the policy taken in episode $k$, and $V^{\pi^k}_h(\cdot) : \mathcal{S} \mapsto \mathbb{R}$ is the value function at step $h$ for policy $\pi^k$. Intuitively, $w_{kh}^{CVaR,\alpha,V^{\pi^k}}(s,a)$ is the probability of visiting $(s,a)$ at step $h$ of episode $k$ under conditional transition probability $\beta^{\alpha,V^{\pi^k}_{h'+1}}(\cdot|\cdot,\cdot)$ for each step $h' = 1, \ldots, h-1$. It holds that for any risk level $\alpha \in (0,1]$, $k > 0$, $h \in [H]$ and $(s,a) \in \mathcal{S} \times \mathcal{A}$, $w_{kh}^{CVaR,\alpha,V^{\pi^k}}(s,a) \in [0,1]$ and $\sum_{(s,a) \in \mathcal{S} \times \mathcal{A}} w_{kh}^{CVaR,\alpha,V^{\pi^k}}(s,a) = 1$.

**Lemma 8.** *For any risk level $\alpha \in (0,1]$, $k > 0$, $h \in [H]$ and $(s,a) \in \mathcal{S} \times \mathcal{A}$, if $w_{kh}(s,a) = 0$, then* $w_{kh}^{CVaR,\alpha,V^{\pi^k}}(s,a) = 0$.

*Proof of Lemma 8.* If $w_{kh}(s,a) = 0$, then the algorithm has zero probability to visit $(s,a)$ at step $h$ of episode $k$, which means that $(s,a)$ is unreachable under transition probability $p(\cdot|\cdot,\cdot)$.

Note that for each step $h' = 1, \ldots, h-1$, the conditional transition probability $\beta^{\alpha,V^{\pi^k}_{h'+1}}(s'|s,a)$ just renormalizes the transition probability and assigns more weights to the worst $\alpha$-portion successor states $s'$ (i.e., with the lowest $\alpha$-portion values $V^{\pi^k}_{h'+1}(s')$), but will not make an unreachable successor state reachable. Thus, $(s,a)$ is also unreachable under conditional transition probability $\beta^{\alpha,V^{\pi^k}_{h'+1}}(\cdot|\cdot,\cdot)$ for each step $h' = 1, \ldots, h-1$, and therefore $w_{kh}^{CVaR,\alpha,V^{\pi^k}}(s,a) = 0$. $\qquad\square$

**Lemma 9.** *For any functions $V_1, \ldots, V_H : \mathcal{S} \mapsto \mathbb{R}$, $k > 0$, $h \in [H]$ and $(s,a) \in \mathcal{S} \times \mathcal{A}$ such that $w_{kh}(s,a) > 0$,*

$$\frac{w_{kh}^{\text{CVaR},\alpha,V}(s,a)}{w_{kh}(s,a)} \leq \min\left\{ \frac{1}{\min\limits_{\pi,h,(s,a): w_{\pi,h}(s,a)>0} w_{\pi,h}(s,a)}, \frac{1}{\alpha^{h-1}} \right\},$$

*where $w_{kh}^{\text{CVaR},\alpha,V}(s,a)$ denotes the conditional probability of visiting $(s,a)$ at step $h$ of episode $k$, conditioning on transitioning to the worst $\alpha$-portion successor states $s'$ (i.e., with the lowest $\alpha$-portion values $V_{h'+1}(s')$) at each step $h' = 1, \ldots, h-1$.*

*Proof of Lemma 9.* Since $w_{kh}^{\text{CVaR},\alpha,V}(s,a)$ is the conditional probability of visiting $(s,a)$, we have $w_{kh}^{\text{CVaR},\alpha,V}(s,a) \in [0,1]$. Since $w_{kh}(s,a)$ is the probability of visiting $(s,a)$ at step $h$ under policy $\pi^k$ and $\min_{\pi,h,(s,a): w_{\pi,h}(s,a)>0} w_{\pi,h}(s,a)$ is the minimum probability of visiting any reachable $(s,a)$ at any step $h$ over all policies $\pi$, we have

$$w_{kh}(s,a) \geq \min_{\pi,h,(s,a): w_{\pi,h}(s,a)>0} w_{\pi,h}(s,a).$$

Hence, we have

$$\frac{w_{kh}^{\text{CVaR},\alpha,V}(s,a)}{w_{kh}(s,a)} \leq \frac{1}{\min\limits_{\pi,h,(s,a): w_{\pi,h}(s,a)>0} w_{\pi,h}(s,a)}. \tag{11}$$

Let $s_1$ be the initial state. Since $w_{kh}(s,a)$ and $w_{kh}^{\text{CVaR},\alpha,V}(s,a)$ are the probabilities of visiting $(s,a)$ at step $h$ with policy $\pi^k$ under transition probability $p(\cdot|\cdot,\cdot)$ and conditional transition probability $\beta^{\alpha,V_{h'+1}}(\cdot|\cdot,\cdot)$ for each step $h' = 1, \ldots, h-1$, respectively, we have that

$$w_{kh}(s,a) = \sum_{(s_2,\ldots,s_{h-1}) \in \mathcal{S}^{h-2}} \prod_{h'=1}^{h-1} p(s_{h'+1}|s_{h'}, a_{h'})$$

and

$$w_{kh}^{\text{CVaR},\alpha,V}(s,a) = \sum_{(s_2,\ldots,s_{h-1})\in\mathcal{S}^{h-2}} \prod_{h'=1}^{h-1} \beta^{\alpha,V_{h'+1}}(s_{h'+1}|s_{h'},a_{h'}),$$

where $s_1$ is the initial state, $s_h := s$, and $a_{h'} := \pi^k(s_{h'})$ for $h' = 1,\ldots,h-1$.

Recall that for any risk level $\alpha \in (0,1]$, function $V : \mathcal{S} \mapsto \mathbb{R}$ and $(s',s,a) \in \mathcal{S}\times\mathcal{S}\times\mathcal{A}$, $\mu^{\alpha,V}(s'|s,a)$ denotes how large the transition probability of successor state $s'$ belongs to the worst $\alpha$-portion successor states (i.e., with the lowest $\alpha$-portion values $V(\cdot)$), which satisfies that $\frac{\mu^{\alpha,V}(s'|s,a)}{\alpha} = \beta^{\alpha,V}(s'|s,a)$ and $0 \le \mu^{\alpha,V}(s'|s,a) \le p(s'|s,a)$.

Thus, we have

$$
\begin{aligned}
w_{kh}^{\text{CVaR},\alpha,V}(s,a) &= \sum_{(s_2,\ldots,s_{h-1})\in\mathcal{S}^{h-2}} \prod_{h'=1}^{h-1} \frac{\mu^{\alpha,V_{h'+1}}(s_{h'+1}|s_{h'},a_{h'})}{\alpha} \\
&\le \sum_{(s_2,\ldots,s_{h-1})\in\mathcal{S}^{h-2}} \prod_{h'=1}^{h-1} \frac{p(s_{h'+1}|s_{h'},a_{h'})}{\alpha} \\
&= \frac{1}{\alpha^{h-1}} \sum_{(s_2,\ldots,s_{h-1})\in\mathcal{S}^{h-2}} \prod_{h'=1}^{h-1} p(s_{h'+1}|s_{h'},a_{h'}) \\
&= \frac{1}{\alpha^{h-1}} \cdot w_{kh}(s,a)
\end{aligned}
$$

Therefore,

$$\frac{w_{kh}^{\text{CVaR},\alpha,V}(s,a)}{w_{kh}(s,a)} \le \frac{1}{\alpha^{h-1}}. \tag{12}$$

Combining Eqs. (11) and (12), we obtain this lemma. $\qquad\square$

**Lemma 10** (Insufficient Visitation). *It holds that*

$$
\sum_{k=1}^{K}\sum_{h=1}^{H}\sum_{(s,a)\notin\mathcal{L}_k} w_{kh}^{\text{CVaR},\alpha,V^{\pi^k}}(s,a) \le \min\left\{ \frac{1}{\min\limits_{\pi,h,s:\, w_{\pi,h}(s)>0} w_{\pi,h}(s)}, \frac{1}{\alpha^{H-1}} \right\}\cdot
$$
$$
\left( 4SAH\log\left(\frac{HSA}{\delta'}\right) + 5SAH \right).
$$

*Proof of Lemma 10.* According to the definition of $\mathcal{L}_k$ (Eq. (10)), for any $(s,a) \in \mathcal{S}\times\mathcal{A}$, once $(s,a)$ satisfies $(s,a)\in\mathcal{L}_k$ in some episode $k$, it will always satisfy $(s,a)\in\mathcal{L}_{k'}$ for all $k'\ge k$. For any $(s,a)\in\mathcal{S}\times\mathcal{A}$, let $\tilde{k}(s,a)$ denote the last episode $k$ where $(s,a)\notin\mathcal{L}_k$. Then, we have

$$
\begin{aligned}
\sum_{k=1}^{K}\sum_{h=1}^{H}\sum_{(s,a)\notin\mathcal{L}_k} w_{kh}(s,a) &= \sum_{(s,a)\in\mathcal{S}\times\mathcal{A}}\sum_{k=1}^{K}\sum_{h=1}^{H} w_{kh}(s,a)\cdot\mathbb{1}\{(s,a)\notin\mathcal{L}_k\} \\
&= \sum_{(s,a)\in\mathcal{S}\times\mathcal{A}}\sum_{k=1}^{\tilde{k}(s,a)}\sum_{h=1}^{H} w_{kh}(s,a) \\
&= \sum_{(s,a)\in\mathcal{S}\times\mathcal{A}}\left( \sum_{k=1}^{\tilde{k}(s,a)-1}\sum_{h=1}^{H} w_{kh}(s,a) + \sum_{h=1}^{H} w_{\tilde{k}(s,a),h}(s,a) \right) \\
&\overset{(a)}{<} \sum_{(s,a)\in\mathcal{S}\times\mathcal{A}}\left( 4H\log\left(\frac{HSA}{\delta'}\right) + 4H + H \right)
\end{aligned}
$$

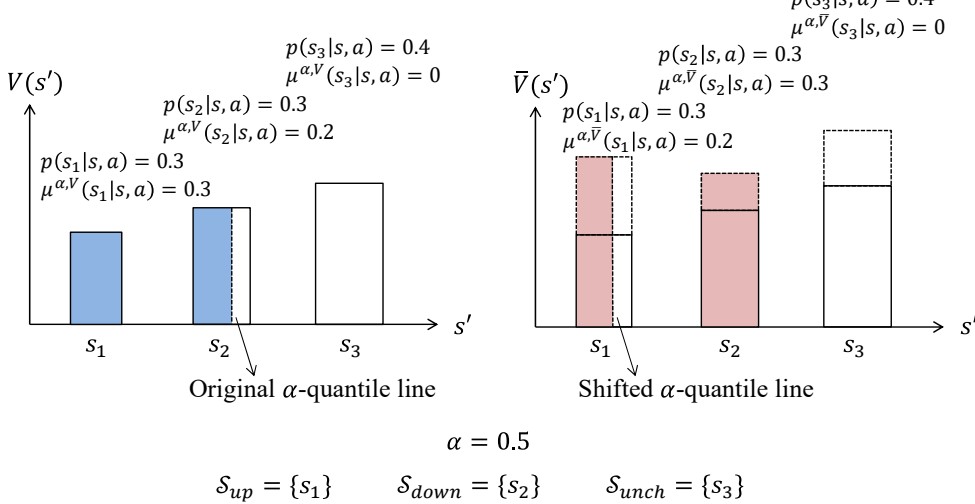

$$\alpha = 0.5$$

$$\mathcal{S}_{up} = \{s_1\} \qquad \mathcal{S}_{down} = \{s_2\} \qquad \mathcal{S}_{unch} = \{s_3\}$$

Figure 5: Illustrating example for Lemma 11. For each $s' \in \{s_1, s_2, s_3\}$, the height of the bar denotes the value $V(s')$ or $\bar{V}(s')$, and the width of the bar denotes the transition probability $p(s'|s, a)$ (fixed). The colored part of the bars denotes the worst $\alpha$-portion successor states (i.e., with the lowest $\alpha$-portion values $V(s')$ or $\bar{V}(s')$). In this example, $\alpha = 0.5$.

$$\leq 4SAH \log \left( \frac{HSA}{\delta'} \right) + 5SAH,$$

where (a) is due to that $(s, a) \notin \mathcal{L}_{\tilde{k}(s,a)}$, and for any $k > 0$, $h \in [H]$ and $(s, a) \in \mathcal{S} \times \mathcal{A}$, $w_{kh}(s, a) \in [0, 1]$.

For any policy $\pi$, $h \in [H]$ and $(s, a) \in \mathcal{S} \times \mathcal{A}$, let $w_{\pi,h}(s, a)$ and $w_{\pi,h}(s)$ denote the probabilities of visiting $(s, a)$ and $s$ at step $h$ under policy $\pi$, respectively. Then, we have

$$\sum_{k=1}^{K} \sum_{h=1}^{H} \sum_{(s,a) \notin \mathcal{L}_k} w_{kh}^{\mathrm{CVaR}, \alpha, V^{\pi^k}}(s, a)$$

$$\overset{(a)}{=} \sum_{k=1}^{K} \sum_{h=1}^{H} \sum_{(s,a) \notin \mathcal{L}_k} \frac{w_{kh}^{\mathrm{CVaR}, \alpha, V^{\pi^k}}(s, a)}{w_{kh}(s, a)} \cdot w_{kh}(s, a) \cdot \mathbb{1}\{w_{kh}(s, a) \neq 0\}$$

$$\overset{(b)}{\leq} \min \left\{ \frac{1}{\min_{\pi,h,(s,a):\ w_{\pi,h}(s,a)>0} w_{\pi,h}(s, a)}, \frac{1}{\alpha^{H-1}} \right\} \sum_{k=1}^{K} \sum_{h=1}^{H} \sum_{(s,a) \notin \mathcal{L}_k} w_{kh}(s, a)$$

$$\overset{(c)}{\leq} \min \left\{ \frac{1}{\min_{\pi,h,s:\ w_{\pi,h}(s)>0} w_{\pi,h}(s)}, \frac{1}{\alpha^{H-1}} \right\} \left( 4SAH \log \left( \frac{HSA}{\delta'} \right) + 5SAH \right),$$

Here (a) is due to Lemma 8. (b) comes from Lemma 9. (c) uses the fact that for any deterministic policy $\pi$, $h \in [H]$ and $(s, a) \in \mathcal{S} \times \mathcal{A}$, we have either $w_{\pi,h}(s, a) = w_{\pi,h}(s)$ or $w_{\pi,h}(s, a) = 0$, and thus $\min_{\pi,h,(s,a):\ w_{\pi,h}(s,a)>0} w_{\pi,h}(s, a) = \min_{\pi,h,s:\ w_{\pi,h}(s)>0} w_{\pi,h}(s)$. $\qquad \square$

Recall that for any risk level $\alpha \in (0, 1]$, function $V : \mathcal{S} \mapsto \mathbb{R}$ and $(s', s, a) \in \mathcal{S} \times \mathcal{S} \times \mathcal{A}$, $\beta^{\alpha,V}(s'|s, a)$ is the conditional probability of transitioning to $s'$ from $(s, a)$, conditioning on transitioning to the worst $\alpha$-portion successor states $s'$ (i.e., with the lowest $\alpha$-portion values $V(s')$), and it holds that $\mathrm{CVaR}_{s' \sim p(s'|s,a)}^{\alpha}(V(s')) = \sum_{s' \in \mathcal{S}} \beta^{\alpha,V}(s'|s, a) \cdot V(s')$.

**Lemma 11** (CVaR Gap due to Value Function Shift). *For any* $(s, a) \in \mathcal{S} \times \mathcal{A}$, *distribution* $p(\cdot|s, a) \in \triangle_{\mathcal{S}}$, *and functions* $V, \bar{V} : \mathcal{S} \mapsto [0, H]$ *such that* $\bar{V}(s') \geq V(s')$ *for any* $s' \in \mathcal{S}$,

$$\mathrm{CVaR}_{s' \sim p(\cdot|s,a)}^{\alpha}(\bar{V}(s')) - \mathrm{CVaR}_{s' \sim p(\cdot|s,a)}^{\alpha}(V(s')) \leq \beta^{\alpha,V}(\cdot|s, a)^{\top} \left( \bar{V} - V \right).$$

*Proof of Lemma 11.* Recall that for any risk level $\alpha \in (0, 1]$, function $V : \mathcal{S} \mapsto \mathbb{R}$ and $(s', s, a) \in \mathcal{S} \times \mathcal{S} \times \mathcal{A}$, $\beta^{\alpha,V}(s'|s,a)$ is the conditional transition probability from $(s, a)$ to $s'$, conditioning on transitioning to the worst $\alpha$-portion successor states $s'$ (i.e., with the lowest $\alpha$-portion values $V(s')$), and $\mu^{\alpha,V}(s'|s,a)$ denotes how large the transition probability of successor state $s'$ belongs to the worst $\alpha$-portion, which satisfies that $\frac{\mu^{\alpha,V}(s'|s,a)}{\alpha} = \beta^{\alpha,V}(s'|s,a)$ and $\sum_{s' \in \mathcal{S}} \mu^{\alpha,V}(s'|s,a) = \alpha$. Then, for any risk level $\alpha \in (0, 1]$, function $V : \mathcal{S} \mapsto \mathbb{R}$ and $(s', s, a) \in \mathcal{S} \times \mathcal{S} \times \mathcal{A}$,

$$\text{CVaR}^\alpha_{s' \sim p(\cdot|s,a)}(V(s')) = \frac{\sum_{s' \in \mathcal{S}} \mu^{\alpha,V}(s'|s,a) \cdot V(s')}{\alpha} = \sum_{s' \in \mathcal{S}} \beta^{\alpha,V}(s'|s,a) \cdot V(s'),$$

As shown in Figure 5, we sort all successor states $s' \in \mathcal{S}$ by their values $V(s')$ in ascending order (from left to right). Fix the transition probability $p(\cdot|s,a)$ and the value function shifts from $V(\cdot)$ to $\bar{V}(\cdot)$. Then, below we divide all successor states $s' \in \mathcal{S}$ into three subsets, i.e., $\mathcal{S}_{up}$, $\mathcal{S}_{down}$ and $\mathcal{S}_{unch}$, according to how $\mu^{\alpha,V}(s'|s,a)$ changes to $\mu^{\alpha,\bar{V}}(s'|s,a)$ as $V(s')$ shifts to $\bar{V}(s')$.

- For any $s' \in \mathcal{S}_{up}$, $\mu^{\alpha,\bar{V}}(s'|s,a) < \mu^{\alpha,V}(s'|s,a)$, the rank of $s'$ goes up, and the position of $s'$ moves to the right (here "rank" means to rank all successor states $s' \in \mathcal{S}$ by their values $V(s')$ or $\bar{V}(s')$ from highest to lowest).

- For any $s' \in \mathcal{S}_{down}$, $\mu^{\alpha,\bar{V}}(s'|s,a) > \mu^{\alpha,V}(s'|s,a)$, the rank of $s'$ goes down, and the position of $s'$ moves to the left.

- For any $s' \in \mathcal{S}_{unch}$, $\mu^{\alpha,\bar{V}}(s'|s,a) = \mu^{\alpha,V}(s'|s,a)$, the rank and position of $s'$ keep unchanged.

Then, it holds that

$$\sum_{s' \in \mathcal{S}_{up}} \left( \mu^{\alpha,\bar{V}}(s'|s,a) - \mu^{\alpha,V}(s'|s,a) \right) + \sum_{s' \in \mathcal{S}_{down}} \left( \mu^{\alpha,\bar{V}}(s'|s,a) - \mu^{\alpha,V}(s'|s,a) \right) = 0. \quad (13)$$

Next, we have

$$\text{CVaR}^\alpha_{s' \sim p(\cdot|s,a)}(\bar{V}(s')) - \text{CVaR}^\alpha_{s' \sim p(\cdot|s,a)}(V(s'))$$

$$= \frac{1}{\alpha} \cdot \left( \sum_{s' \in \mathcal{S}_{up}} \left( \mu^{\alpha,\bar{V}}(s'|s,a) \cdot \bar{V}(s') - \mu^{\alpha,V}(s'|s,a) \cdot V(s') \right) \right.$$

$$+ \sum_{s' \in \mathcal{S}_{down}} \left( \mu^{\alpha,\bar{V}}(s'|s,a) \cdot \bar{V}(s') - \mu^{\alpha,V}(s'|s,a) \cdot V(s') \right)$$

$$\left. + \sum_{s' \in \mathcal{S}_{unch}} \left( \mu^{\alpha,\bar{V}}(s'|s,a) \cdot \bar{V}(s') - \mu^{\alpha,V}(s'|s,a) \cdot V(s') \right) \right)$$

$$= \frac{1}{\alpha} \cdot \left( \sum_{s' \in \mathcal{S}_{up}} \left( \mu^{\alpha,V}(s'|s,a) \cdot \left( \bar{V}(s') - V(s') \right) \quad + \left( \mu^{\alpha,\bar{V}}(s'|s,a) - \mu^{\alpha,V}(s'|s,a) \right) \cdot \bar{V}(s') \right) \right.$$

$$+ \sum_{s' \in \mathcal{S}_{down}} \left( \mu^{\alpha,V}(s'|s,a) \cdot \left( \bar{V}(s') - V(s') \right) \quad + \left( \mu^{\alpha,\bar{V}}(s'|s,a) - \mu^{\alpha,V}(s'|s,a) \right) \cdot \bar{V}(s') \right)$$

$$\left. + \sum_{s' \in \mathcal{S}_{unch}} \mu^{\alpha,V}(s'|s,a) \cdot \left( \bar{V}(s') - V(s') \right) \right)$$

$$= \frac{1}{\alpha} \cdot \left( \sum_{s \in \mathcal{S}} \mu^{\alpha,V}(s'|s,a) \cdot \left( \bar{V}(s') - V(s') \right) - \sum_{s' \in \mathcal{S}_{up}} \left( \mu^{\alpha,V}(s'|s,a) - \mu^{\alpha,\bar{V}}(s'|s,a) \right) \cdot \bar{V}(s') \right.$$

$$\left. + \sum_{s' \in \mathcal{S}_{down}} \left( \mu^{\alpha,\bar{V}}(s'|s,a) - \mu^{\alpha,V}(s'|s,a) \right) \cdot \bar{V}(s') \right)$$

$$\overset{(a)}{\leq} \frac{1}{\alpha} \left( \sum_{s \in \mathcal{S}} \mu^{\alpha,V}(s'|s,a) \cdot \left( \bar{V}(s') - V(s') \right) - \min_{s' \in \mathcal{S}_{up}} \bar{V}(s') \cdot \sum_{s' \in \mathcal{S}_{up}} \left( \mu^{\alpha,V}(s'|s,a) - \mu^{\alpha,\bar{V}}(s'|s,a) \right) \right.$$

$$\left. + \min_{s' \in \mathcal{S}_{up}} \bar{V}(s') \cdot \sum_{s' \in \mathcal{S}_{down}} \left( \mu^{\alpha,\bar{V}}(s'|s,a) - \mu^{\alpha,V}(s'|s,a) \right) \right)$$

$$\overset{(b)}{=} \frac{1}{\alpha} \cdot \sum_{s \in \mathcal{S}} \mu^{\alpha,V}(s'|s,a) \cdot \left( \bar{V}(s') - V(s') \right)$$

$$= \beta^{\alpha,V}(\cdot|s,a)^{\top} \left( \bar{V} - V \right)$$

Here (a) is due to that for any $s' \in \mathcal{S}_{up}$, $\mu^{\alpha,\bar{V}}(s'|s,a) < \mu^{\alpha,V}(s'|s,a)$, and for any $s \in \mathcal{S}_{up}$, $s' \in \mathcal{S}_{down}$, $\bar{V}(s) \geq \bar{V}(s')$. (b) comes from Eq. (13).

$\square$

### D.1.3  PROOF OF THEOREM 1

*Proof of Theorem 1.* Suppose that event $\mathcal{E}$ holds. Then, for any $k \in [K]$,

$$V_1^*(s_1^k) - V_1^{\pi^k}(s_1^k)$$

$$\overset{(a)}{\leq} \bar{V}_1^k(s_1^k) - V_1^{\pi^k}(s_1^k)$$

$$= \min \left\{ r(s_1^k, a_1^k) + \text{CVaR}_{s' \sim \hat{p}^k(\cdot|s_1^k, a_1^k)}^{\alpha}(\bar{V}_2^k(s')) + \frac{H}{\alpha} \sqrt{\frac{L}{n_k(s_1^k, a_1^k)}},\ H \right\}$$

$$- \left( r(s_1^k, a_1^k) + \text{CVaR}_{s' \sim p(\cdot|s_1^k, a_1^k)}^{\alpha}(V_2^{\pi^k}(s')) \right)$$

$$\leq r(s_1^k, a_1^k) + \text{CVaR}_{s' \sim \hat{p}^k(\cdot|s_1^k, a_1^k)}^{\alpha}(\bar{V}_2^k(s')) + \min \left\{ \frac{H}{\alpha} \sqrt{\frac{L}{n_k(s_1^k, a_1^k)}},\ H \right\}$$

$$- \left( r(s_1^k, a_1^k) + \text{CVaR}_{s' \sim p(\cdot|s_1^k, a_1^k)}^{\alpha}(V_2^{\pi^k}(s')) \right)$$

$$= \min \left\{ \frac{H}{\alpha} \sqrt{\frac{L}{n_k(s_1^k, a_1^k)}},\ H \right\} + \text{CVaR}_{s' \sim \hat{p}^k(\cdot|s_1^k, a_1^k)}^{\alpha}(\bar{V}_2^k(s')) - \text{CVaR}_{s' \sim p(\cdot|s_1^k, a_1^k)}^{\alpha}(\bar{V}_2^k(s'))$$

$$+ \text{CVaR}_{s' \sim p(\cdot|s_1^k, a_1^k)}^{\alpha}(\bar{V}_2^k(s')) - \text{CVaR}_{s' \sim p(\cdot|s_1^k, a_1^k)}^{\alpha}(V_2^{\pi^k}(s'))$$

$$\overset{(b)}{\leq} \min \left\{ \frac{H}{\alpha} \sqrt{\frac{L}{n_k(s_1^k, a_1^k)}},\ H \right\} + \min \left\{ \frac{4H}{\alpha} \sqrt{\frac{SL}{n_k(s_1^k, a_1^k)}},\ H \right\} + \beta^{\alpha,V_2^{\pi^k}}(\cdot|s_1^k, a_1^k)^{\top}(\bar{V}_2^k - V_2^{\pi^k})$$

$$\overset{(c)}{\leq} \min \left\{ \frac{H\sqrt{L} + 4H\sqrt{SL}}{\alpha\sqrt{n_k(s_1^k, a_1^k)}},\ 2H \right\} + \sum_{s_2 \in \mathcal{S}} \beta^{\alpha,V_2^{\pi^k}}(s_2|s_1^k, a_1^k) \cdot (\bar{V}_2^k(s_2) - V_2^{\pi^k}(s_2))$$

$$\overset{(d)}{\leq} \min \left\{ \frac{H\sqrt{L} + 4H\sqrt{SL}}{\alpha\sqrt{n_k(s_1^k, a_1^k)}},\ 2H \right\} + \sum_{s_2 \in \mathcal{S}} \beta^{\alpha,V_2^{\pi^k}}(s_2|s_1^k, a_1^k) \cdot$$

$$\left( \min \left\{ \frac{H\sqrt{L} + 4H\sqrt{SL}}{\alpha\sqrt{n_k(s_2, a_2)}},\ 2H \right\} + \sum_{s_3 \in \mathcal{S}} \beta^{\alpha,V_3^{\pi^k}}(s_3|s_2, a_2) \cdot (\bar{V}_3^k(s_3) - V_3^{\pi^k}(s_3)) \right)$$

$$\overset{(e)}{\leq} \min \left\{ \frac{H\sqrt{L} + 4H\sqrt{SL}}{\alpha\sqrt{n_k(s_1^k, a_1^k)}},\ 2H \right\} + \sum_{s_2 \in \mathcal{S}} \beta^{\alpha,V_2^{\pi^k}}(s_2|s_1^k, a_1^k) \cdot$$

$$\left( \min \left\{ \frac{H\sqrt{L} + 4H\sqrt{SL}}{\alpha\sqrt{n_k(s_2, a_2)}},\ 2H \right\} + \sum_{s_3 \in \mathcal{S}} \beta^{\alpha,V_3^{\pi^k}}(s_3|s_2, a_2) \cdot$$

$$\left( \cdots \sum_{s_H \in \mathcal{S}} \beta^{\alpha, V_H^{\pi^k}}(s_H | s_{H-1}, a_{H-1}) \cdot \left( \min \left\{ \frac{H\sqrt{L} + 4H\sqrt{SL}}{\alpha \sqrt{n_k(s_H, a_H)}}, \, 2H \right\} \right) \right) \right)$$

$$\overset{(f)}{=} \sum_{h=1}^{H} \sum_{(s,a) \in \mathcal{S} \times \mathcal{A}} w_{kh}^{CVaR, \alpha, V^{\pi^k}}(s,a) \cdot \min \left\{ \frac{H\sqrt{L} + 4H\sqrt{SL}}{\alpha \sqrt{n_k(s,a)}}, \, 2H \right\}$$

$$\leq \sum_{h=1}^{H} \sum_{(s,a) \in \mathcal{L}_k} w_{kh}^{CVaR, \alpha, V^{\pi^k}}(s,a) \cdot \frac{H\sqrt{L} + 4H\sqrt{SL}}{\alpha \sqrt{n_k(s,a)}}$$

$$+ \sum_{h=1}^{H} \sum_{(s,a) \notin \mathcal{L}_k} w_{kh}^{CVaR, \alpha, V^{\pi^k}}(s,a) \cdot 2H \tag{14}$$

Here $a_h := \pi^k(s_h)$ for $h = 2, \ldots, H$. (a) is due to Lemma 5. (b) uses Lemma 2 and the fact that for any $k > 0$, $h \in [H]$ and $s \in \mathcal{S}$, $\bar{V}_h^k(s) \in [0, H]$, and thus for any $k > 0$, $h \in [H]$ and $(s,a) \in \mathcal{S} \times \mathcal{A}$, $\mathrm{CVaR}_{s' \sim \hat{p}^k(\cdot | s,a)}^{\alpha}(\bar{V}_{h+1}^k(s')) - \mathrm{CVaR}_{s' \sim p(\cdot | s,a)}^{\alpha}(\bar{V}_{h+1}^k(s')) \leq H$, and also uses Lemma 11. (c) comes from the property of $\min\{\cdot, \cdot\}$. (d) and (e) follow from recurrently applying steps (a)-(c). (f) is due to that $w_{kh}^{CVaR, \alpha, V^{\pi^k}}(s,a)$ is defined as the probability of visiting $(s,a)$ at step $h$ of episode $k$ under the conditional transition probability $\beta^{\alpha, V_{h'+1}^{\pi^k}}(\cdot | \cdot, \cdot)$ for each step $h' = 1, \ldots, h - 1$.

Since the second term in Eq. (14) can be bounded by Lemma 10, below we analyze the first term.

Recall that for any policy $\pi$, $h \in [H]$ and $(s,a) \in \mathcal{S} \times \mathcal{A}$, $w_{\pi,h}(s,a)$ and $w_{\pi,h}(s)$ denote the probabilities of visiting $(s,a)$ and $s$ at step $h$ under policy $\pi$, respectively. Summing the first term in Eq. (14) over $k \in [K]$, we have

$$\sum_{k=1}^{K} \sum_{h=1}^{H} \sum_{(s,a) \in \mathcal{L}_k} w_{kh}^{CVaR, \alpha, V^{\pi^k}}(s,a) \frac{H\sqrt{L} + 4H\sqrt{SL}}{\alpha \sqrt{n_k(s,a)}}$$

$$\leq \frac{H\sqrt{L} + 4H\sqrt{SL}}{\alpha} \sqrt{\sum_{k=1}^{K} \sum_{h=1}^{H} \sum_{(s,a) \in \mathcal{L}_k} \frac{w_{kh}^{CVaR, \alpha, V^{\pi^k}}(s,a)}{n_k(s,a)}} \cdot \sqrt{\sum_{k=1}^{K} \sum_{h=1}^{H} \sum_{(s,a) \in \mathcal{L}_k} w_{kh}^{CVaR, \alpha, V^{\pi^k}}(s,a)}$$

$$\overset{(a)}{=} \frac{H\sqrt{L} + 4H\sqrt{SL}}{\alpha} \sqrt{\sum_{k=1}^{K} \sum_{h=1}^{H} \sum_{(s,a) \in \mathcal{L}_k} \frac{w_{kh}^{CVaR, \alpha, V^{\pi^k}}(s,a)}{n_k(s,a)} \cdot \mathbb{1}\{w_{kh}(s,a) \neq 0\}} \cdot \sqrt{KH}$$

$$= \frac{(H\sqrt{L} + 4H\sqrt{SL})\sqrt{KH}}{\alpha} \sqrt{\sum_{k=1}^{K} \sum_{h=1}^{H} \sum_{(s,a) \in \mathcal{L}_k} \frac{w_{kh}^{CVaR, \alpha, V^{\pi^k}}(s,a)}{w_{kh}(s,a)} \cdot \frac{w_{kh}(s,a)}{n_k(s,a)} \cdot \mathbb{1}\{w_{kh}(s,a) \neq 0\}}$$

$$\overset{(b)}{\leq} \frac{(H\sqrt{L} + 4H\sqrt{SL})\sqrt{KH}}{\alpha} \sqrt{\min \left\{ \frac{1}{\min_{\pi, h, (s,a): w_{\pi,h}(s,a) > 0} w_{\pi,h}(s,a)}, \frac{1}{\alpha^{H-1}} \right\} \sum_{k=1}^{K} \sum_{h=1}^{H} \sum_{(s,a) \in \mathcal{L}_k} \frac{w_{kh}(s,a)}{n_k(s,a)}}$$

$$\overset{(c)}{\leq} \frac{(H\sqrt{L} + 4H\sqrt{SL})\sqrt{KH}}{\alpha} \cdot 2\sqrt{SAL} \cdot \min \left\{ \frac{1}{\sqrt{\min_{\pi, h, s: w_{\pi,h}(s) > 0} w_{\pi,h}(s)}}, \frac{1}{\sqrt{\alpha^{H-1}}} \right\}$$

$$\leq \frac{10HSL\sqrt{KHA}}{\alpha} \cdot \min \left\{ \frac{1}{\sqrt{\min_{\pi, h, s: w_{\pi,h}(s) > 0} w_{\pi,h}(s)}}, \frac{1}{\sqrt{\alpha^{H-1}}} \right\}.$$

Here (a) is due to Lemma 8 and the fact that for any $k > 0$ and $h \in [H]$, $\sum_{(s,a) \in \mathcal{S} \times \mathcal{A}} w_{kh}^{CVaR, \alpha, V^{\pi^k}}(s,a) = 1$. (b) comes from Lemma 9. (c) uses Lemma 7 and the fact that

for any deterministic policy $\pi$, $h \in [H]$ and $(s,a) \in \mathcal{S} \times \mathcal{A}$, we have either $w_{\pi,h}(s,a) = w_{\pi,h}(s)$ or $w_{\pi,h}(s,a) = 0$, and thus $\min_{\pi,h,(s,a):\ w_{\pi,h}(s,a)>0} w_{\pi,h}(s,a) = \min_{\pi,h,s:\ w_{\pi,h}(s)>0} w_{\pi,h}(s)$.

Then, summing the first and second terms in Eq. (14) over $k \in [K]$ and using Lemma 10, we have

$$
\begin{aligned}
\mathcal{R}(K) &= \sum_{k=1}^{K} \left( V_1^*(s_1^k) - V_1^{\pi^k}(s_1^k) \right) \\
&\leq \sum_{k=1}^{K} \sum_{h=1}^{H} \sum_{(s,a) \in \mathcal{L}_k} w_{kh}^{\mathrm{CVaR},\alpha,V^{\pi^k}}(s,a) \frac{H\sqrt{L} + 4H\sqrt{SL}}{\alpha\sqrt{n_k(s,a)}} \\
&\quad + \sum_{k=1}^{K} \sum_{h=1}^{H} \sum_{(s,a) \notin \mathcal{L}_k} w_{kh}^{\mathrm{CVaR},\alpha,V^{\pi^k}}(s,a) \cdot 2H \\
&\leq \min \left\{ \frac{1}{\sqrt{\min\limits_{\pi,h,s:\ w_{\pi,h}(s)>0} w_{\pi,h}(s)}}, \frac{1}{\sqrt{\alpha^{H-1}}} \right\} \frac{10HS\sqrt{KHA}}{\alpha} \log\left( \frac{KHSA}{\delta'} \right) \\
&\quad + \min \left\{ \frac{1}{\min\limits_{\pi,h,s:\ w_{\pi,h}(s)>0} w_{\pi,h}(s)}, \frac{1}{\alpha^{H-1}} \right\} \left( 8SAH^2 \log\left( \frac{HSA}{\delta'} \right) + 10SAH^2 \right)
\end{aligned}
$$

$\square$

When $K$ is large enough, the first term dominates the bound, and thus we obtain Theorem 1.

### D.2 Proof of Regret Lower Bound

Below we prove the regret lower bound (Theorem 2) for Iterated CVaR RL-RM.

*Proof of Theorem 2.* First, we construct an instance where $\min_{\pi,h,s:\ w_{\pi,h}(s)>0} w_{\pi,h}(s) > \alpha^{H-1}$, and prove that on this instance any algorithm must suffer a $\Omega\left( \frac{H}{\sqrt{\min_{\pi,h,s:\ w_{\pi,h}(s)>0} w_{\pi,h}(s)}} \sqrt{\frac{AK}{\alpha}} \right)$ regret.

Consider the instance shown in Figure 6 (the same as Figure 1 in the main text):

The state space is $\mathcal{S} = \{s_1, s_2, \ldots, s_n, x_1, x_2, x_3\}$, where $s_1$ is the initial state, and $n = S - 3 < S < \frac{1}{2}H$.

The reward functions are as follows. For any $a \in \mathcal{A}$, $r(x_1,a) = 1$, $r(x_2,a) = 0.8$ and $r(x_3,a) = 0.2$. For any $i \in [n]$ and $a \in \mathcal{A}$, $r(s_i,a) = 0$.

The transition distributions are as follows. Let $\mu$ be a parameter which satisfies that $0 < \alpha < \mu < \frac{1}{3}$. For any $a \in \mathcal{A}$, $p(s_2|s_1,a) = \mu$, $p(x_1|s_1,a) = 1 - 3\mu$, $p(x_2|s_1,a) = \mu$ and $p(x_3|s_1,a) = \mu$. For any $i \in \{2, \ldots, n-1\}$ and $a \in \mathcal{A}$, $p(s_{i+1}|s_i,a) = \mu$ and $p(x_1|s_i,a) = 1 - \mu$. $x_1$, $x_2$ and $x_3$ are absorbing states, i.e., for any $a \in \mathcal{A}$, $p(x_1|x_1,a) = 1$, $p(x_2|x_2,a) = 1$ and $p(x_3|x_3,a) = 1$. Let $a_J$ be the optimal action in state $s_n$, which is uniformly drawn from $\mathcal{A}$. For the optimal action $a_J$, $p(x_2|s_n,a_J) = 1 - \alpha + \eta$ and $p(x_3|s_n,a_J) = \alpha - \eta$, where $\eta$ is a parameter which satisfies $0 < \eta < \alpha$ and will be chosen later. For any suboptimal action $a \in \mathcal{A} \setminus \{a_J\}$, $p(x_2|s_n,a) = 1 - \alpha$ and $p(x_3|s_n,a) = \alpha$.

For any $a_j \in \mathcal{A}$, let $\mathbb{E}_j[\cdot]$ and $\mathrm{Pr}_j[\cdot]$ denote the expectation and probability operators under the instance with $a_J = a_j$. Let $\mathbb{E}_{unif}[\cdot]$ and $\mathrm{Pr}_{unif}[\cdot]$ denote the expectation and probability operators under the uniform instance where all actions $a \in \mathcal{A}$ in state $s_n$ have the same transition distribution, i.e., $p(x_2|s_n,a) = 1 - \alpha$ and $p(x_3|s_n,a) = \alpha$.

Fix an algorithm $\mathcal{A}$. Let $\pi^k$ denote the policy taken by algorithm $\mathcal{A}$ in episode $k$. Let $N_{s_n,a_j} = \sum_{k=1}^{K} \mathbb{1}\left\{ \pi^k(s_n) = a_j \right\}$ denote the number of episodes that the policy chooses $a_j$ in state $s_n$. Let $V_{s_n,a_j}$ denote the number of episodes that the algorithm $\mathcal{A}$ visits $(s_n,a_j)$. Let $w(s_n)$ denote the

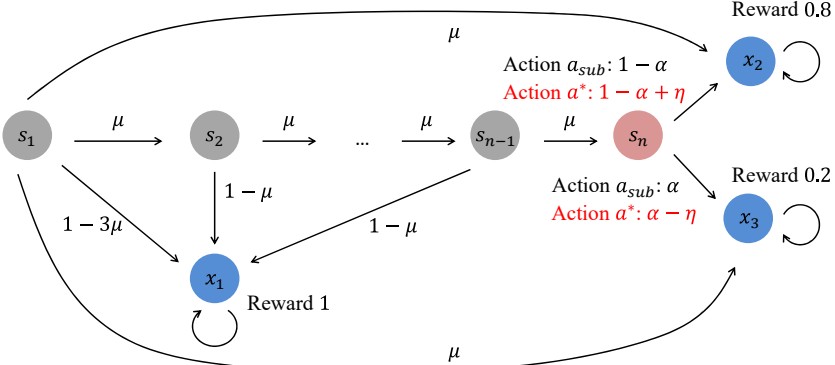

Figure 6: Instance of lower bounds (Theorems 2 and 5) for the $\min_{\pi,h,s:\ w_{\pi,h}(s)>0} w_{\pi,h}(s) > \alpha^{H-1}$ case.

probability of visiting $s_n$ in an episode (the probability of visiting $s_n$ is the same for all policies). Then, it holds that $\mathbb{E}[V_{s_n,a_j}] = w(s_n) \cdot \mathbb{E}[N_{s_n,a_j}]$.

Recall that $a_J$ is the optimal action in state $s_n$. According to the definition of the value function for Iterated CVaR RL, we have that

$$V_1^*(s_1) = \frac{(\alpha - \eta) \cdot 0.2(H - n) + \eta \cdot 0.8(H - n)}{\alpha},$$

and for any policy $\pi$,

$$\begin{aligned}
V_1^\pi(s_1) =& \frac{(\alpha - \eta) \cdot 0.2(H - n) + \eta \cdot 0.8(H - n)}{\alpha} \cdot \mathbb{1}\left\{\pi(s_n) = a_J\right\} \\
&+ 0.2(H - n) \cdot \left(1 - \mathbb{1}\left\{\pi(s_n) = a_J\right\}\right).
\end{aligned}$$

If $J = j$, for any policy $\pi$,

$$V_1^*(s_1) - V_1^\pi(s_1) = \frac{\eta \cdot 0.6(H - n)}{\alpha} \cdot \left(1 - \mathbb{1}\left\{\pi(s_n) = a_j\right\}\right), \tag{15}$$

and summing over all episodes $k \in [K]$, we have

$$\begin{aligned}
\mathbb{E}_j\left[\mathcal{R}(K)\right] =& \sum_{k=1}^K \left(V_1^*(s_1) - V_1^{\pi^k}(s_1)\right) \\
=& \frac{\eta \cdot 0.6(H - n)}{\alpha} \cdot \left(K - \sum_{k=1}^K \mathbb{1}\left\{\pi(s_n) = a_j\right\}\right) \\
=& \frac{\eta \cdot 0.6(H - n)}{\alpha} \cdot \left(K - \mathbb{E}_j[N_{s_n,a_j}]\right)
\end{aligned}$$

Therefore, we have

$$\begin{aligned}
\mathbb{E}\left[\mathcal{R}(K)\right] =& \frac{1}{A} \sum_{j=1}^A \sum_{k=1}^K \left(V_1^*(s_1) - V_1^{\pi^k}(s_1)\right) \\
=& \frac{1}{A} \sum_{j=1}^A \frac{\eta}{\alpha} \cdot 0.6(H - n)\left(K - \mathbb{E}_j[N_{s_n,a_j}]\right) \\
=& 0.6(H - n) \cdot \frac{\eta}{\alpha} \cdot \left(K - \frac{1}{A} \sum_{j=1}^A \mathbb{E}_j[N_{s_n,a_j}]\right) \tag{16}
\end{aligned}$$

For any $j \in [A]$, using Pinsker's inequality and $0 < \alpha < \frac{1}{3}$, we have that $\mathrm{KL}(p_{unif}(s_n, a_j)\|p_j(s_n, a_j)) = \mathrm{KL}(\mathtt{Ber}(\alpha)\|\mathtt{Ber}(\alpha - \eta)) \leq \frac{\eta^2}{(\alpha - \eta)(1 - \alpha + \eta)} \leq \frac{c_1\eta^2}{\alpha}$ for some

constant $c_1$ and small enough $\eta$. Then, using Lemma A.1 in (Auer et al., 2002), we have that for any $j \in [A]$,

$$
\begin{aligned}
\mathbb{E}_j[N_{s_n,a_j}] \leq & \mathbb{E}_{unif}[N_{s_n,a_j}] + \frac{K}{2}\sqrt{\mathbb{E}_{unif}[V_{s_n,a_j}] \cdot \mathrm{KL}\left(p_{unif}(s_n,a_j) \| p_j(s_n,a_j)\right)} \\
\leq & \mathbb{E}_{unif}[N_{s_n,a_j}] + \frac{K}{2}\sqrt{w(s_n) \cdot \mathbb{E}_{unif}[N_{s_n,a_j}] \cdot \frac{c_1\eta^2}{\alpha}}
\end{aligned}
$$

Then, using $\sum_{j=1}^{A} \mathbb{E}_{unif}[N_{s_n,a_j}] = K$ and the Cauchy–Schwarz inequality, we have

$$
\begin{aligned}
\frac{1}{A}\sum_{j=1}^{A} \mathbb{E}_j[N_{s_n,a_j}] \leq & \frac{1}{A}\sum_{j=1}^{A} \mathbb{E}_{unif}[N_{s_n,a_j}] + \frac{K\eta}{2A}\sum_{j=1}^{A}\sqrt{\frac{c_1}{\alpha} \cdot w(s_n) \cdot \mathbb{E}_{unif}[N_{s_n,a_j}]} \\
\leq & \frac{1}{A}\sum_{j=1}^{A} \mathbb{E}_{unif}[N_{s_n,a_j}] + \frac{K\eta}{2A}\sqrt{A\sum_{j=1}^{A}\frac{c_1}{\alpha} \cdot w(s_n) \cdot \mathbb{E}_{unif}[N_{s_n,a_j}]} \\
\leq & \frac{K}{A} + \frac{K\eta}{2}\sqrt{\frac{c_1 \cdot w(s_n)K}{\alpha A}}
\end{aligned}
\tag{17}
$$

By plugging Eq. (17) into Eq. (16), we have

$$
\mathbb{E}\left[\mathcal{R}(K)\right] \geq 0.6(H-n) \cdot \frac{\eta}{\alpha} \cdot \left(K - \frac{K}{A} - \frac{K\eta}{2}\sqrt{\frac{c_1 \cdot w(s_n)K}{\alpha A}}\right).
$$

Let $\eta = c_2\sqrt{\frac{\alpha A}{w(s_n)K}}$ for a small enough constant $c_2$. We have

$$
\begin{aligned}
\mathbb{E}\left[\mathcal{R}(K)\right] =& \Omega\left(H\sqrt{\frac{A}{\alpha \cdot w(s_n)K}} \cdot K\right) \\
=& \Omega\left(H\sqrt{\frac{AK}{\alpha \cdot w(s_n)}}\right)
\end{aligned}
$$

Recall that $n < \frac{1}{2}H$ and $0 < \alpha < \mu < \frac{1}{3}$. Thus, we have that $\min_{\pi,h,s:\, w_{\pi,h}(s)>0} w_{\pi,h}(s) = w(s_n) = \mu^{n-1} > \alpha^{H-1}$, and

$$
\mathbb{E}\left[\mathcal{R}(K)\right] = \Omega\left(H\sqrt{\frac{AK}{\alpha \cdot \min\limits_{\pi,h,s:\, w_{\pi,h}(s)>0} w_{\pi,h}(s)}}\right).
$$

Next, we construct another instance where $\alpha^{H-1} > \min_{\pi,h,s:\, w_{\pi,h}(s)>0} w_{\pi,h}(s)$, and prove that on this instance any algorithm must suffer a $\Omega(\sqrt{\frac{AK}{\alpha^{H-1}}})$ regret.

Consider the instance shown in Figure 7:

The state space is $\mathcal{S} = \{s_1, \ldots, s_n, s'_2, \ldots, s'_n, x_1, x_2, x_3, x_4\}$, where $n = H - 1$ and $s_1$ is the initial state. Let $0 < \alpha < \frac{1}{4}$.

The reward functions are as follows. For any $a \in \mathcal{A}$, $r(x_1, a) = r(x_4, a) = 1$, $r(x_2, a) = 0.8$ and $r(x_3, a) = 0.2$. For any $i \in [n]$ and $a \in \mathcal{A}$, $r(s_i, a) = 0$. For any $i \in \{2, \ldots, n\}$ and $a \in \mathcal{A}$, $r(s'_i, a) = 0$.

The transition distributions are as follows. For any $a \in \mathcal{A}$, $p(s_2|s_1, a) = \alpha$, $p(s'_2|s_1, a) = \gamma$ and $p(x_1|s_1, a) = 1 - \gamma - \alpha$. For any $i \in \{2, \ldots, n-1\}$ and $a \in \mathcal{A}$, $p(s_{i+1}|s_i, a) = \alpha$ and $p(x_1|s_i, a) = 1 - \alpha$. For any $i \in \{2, \ldots, n-1\}$ and $a \in \mathcal{A}$, $p(s'_{i+1}|s'_i, a) = \gamma$ and $p(x_1|s'_i, a) = 1 - \gamma$. For any $a \in \mathcal{A}$, $p(x_4|s'_n, a) = \gamma$ and $p(x_1|s'_n, a) = 1 - \gamma$. $x_1, x_2, x_3$ and $x_4$ are absorbing states, i.e., for any $a \in \mathcal{A}$ and $i \in [4]$, $p(x_i|x_i, a) = 1$. Let $a_J$ be the optimal action in state $s_n$, which is uniformly drawn from $\mathcal{A}$. For the optimal action $a_J$, $p(x_2|s_n, a_J) = 1 - \alpha + \eta$ and $p(x_3|s_n, a_J) = \alpha - \eta$,

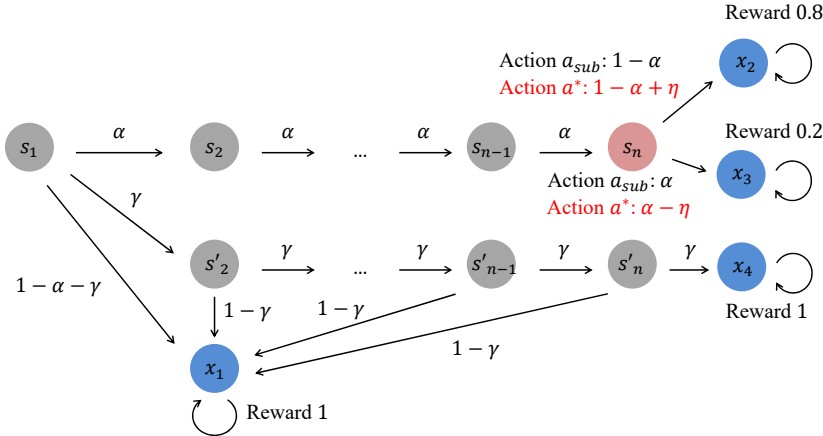

Figure 7: Instance of lower bounds (Theorems 2 and 5) for the $\alpha^{H-1} > \min_{\pi,h,s:\, w_{\pi,h}(s)>0} w_{\pi,h}(s)$ case.

where $\eta$ is a parameter which satisfies $0 < \eta < \alpha$ and will be chosen later. For any suboptimal action $a \in \mathcal{A} \setminus \{a_J\}$, $p(x_2|s_n, a) = 1 - \alpha$ and $p(x_3|s_n, a) = \alpha$.

According to the definition of the value function for Iterated CVaR RL, we have that

$$V_1^*(s_1) = \frac{0.2(\alpha - \eta) + 0.8\eta}{\alpha},$$

and for any policy $\pi$,

$$V_1^\pi(s_1) = \frac{0.2(\alpha - \eta) + 0.8\eta}{\alpha} \cdot \mathbb{1}\{\pi(s_n) = a_J\} + 0.2\left(1 - \mathbb{1}\{\pi(s_n) = a_J\}\right).$$

If $J = j$, for any policy $\pi$,

$$V_1^*(s_1) - V_1^\pi(s_1) = \frac{0.6\eta}{\alpha}\left(1 - \mathbb{1}\{\pi(s_n) = a_j\}\right), \tag{18}$$

and summing over all episodes $k \in [K]$, we have

$$\mathbb{E}_j\left[\mathcal{R}(K)\right] = \sum_{k=1}^{K}\left(V_1^*(s_1) - V_1^{\pi^k}(s_1)\right)$$

$$= \frac{0.6\eta}{\alpha} \cdot \left(K - \sum_{k=1}^{K}\mathbb{1}\{\pi(s_n) = a_j\}\right)$$

$$= \frac{0.6\eta}{\alpha} \cdot \left(K - \mathbb{E}_j[N_{s_n,a_j}]\right)$$

Therefore, we have

$$\mathbb{E}\left[\mathcal{R}(K)\right] = \frac{1}{A}\sum_{j=1}^{A}\sum_{k=1}^{K}\left(V_1^*(s_1) - V_1^{\pi^k}(s_1)\right)$$

$$= \frac{1}{A}\sum_{j=1}^{A}\frac{0.6\eta}{\alpha}\left(K - \mathbb{E}_j[N_{s_n,a_j}]\right)$$

$$= \frac{0.6\eta}{\alpha}\left(K - \frac{1}{A}\sum_{j=1}^{A}\mathbb{E}_j[N_{s_n,a_j}]\right) \tag{19}$$

Recall that $0 < \alpha < \frac{1}{4}$. For any $j \in [A]$, we have that $\mathrm{KL}(p_{unif}(s_n, a_j)\|p_j(s_n, a_j)) = \mathrm{KL}(\mathrm{Ber}(\alpha)\|\mathrm{Ber}(\alpha - \eta)) \le \frac{\eta^2}{(\alpha-\eta)(1-\alpha+\eta)} \le \frac{c_1\eta^2}{\alpha}$ for some constant $c_1$ and small enough $\eta$.

Then, using Lemma A.1 in (Auer et al., 2002), we have that for any $j \in [A]$,

$$\mathbb{E}_j[N_{s_n,a_j}] \leq \mathbb{E}_{unif}[N_{s_n,a_j}] + \frac{K}{2}\sqrt{\mathbb{E}_{unif}[V_{s_n,a_j}] \cdot \mathrm{KL}\left(p_{unif}(s_n,a_j)||p_j(s_n,a_j)\right)}$$

$$\leq \mathbb{E}_{unif}[N_{s_n,a_j}] + \frac{K}{2}\sqrt{w(s_n) \cdot \mathbb{E}_{unif}[N_{s_n,a_j}] \cdot \frac{c_1\eta^2}{\alpha}}$$

Then, using $\sum_{j=1}^{A} \mathbb{E}_{unif}[N_{s_n,a_j}] = K$ and the Cauchy–Schwarz inequality, we have

$$\frac{1}{A}\sum_{j=1}^{A}\mathbb{E}_j[N_{s_n,a_j}] \leq \frac{1}{A}\sum_{j=1}^{A}\mathbb{E}_{unif}[N_{s_n,a_j}] + \frac{K\eta}{2A}\sum_{j=1}^{A}\sqrt{\frac{c_1}{\alpha} \cdot w(s_n) \cdot \mathbb{E}_{unif}[N_{s_n,a_j}]}$$

$$\leq \frac{1}{A}\sum_{j=1}^{A}\mathbb{E}_{unif}[N_{s_n,a_j}] + \frac{K\eta}{2A}\sqrt{A\sum_{j=1}^{A}\frac{c_1}{\alpha} \cdot w(s_n) \cdot \mathbb{E}_{unif}[N_{s_n,a_j}]}$$

$$\leq \frac{K}{A} + \frac{K\eta}{2}\sqrt{\frac{c_1 \cdot w(s_n)K}{\alpha A}} \tag{20}$$

By plugging Eq. (20) into Eq. (19), we have

$$\mathbb{E}\left[\mathcal{R}(K)\right] \geq \frac{0.6\eta}{\alpha} \cdot \left(K - \frac{K}{A} - \frac{K\eta}{2}\sqrt{\frac{c_1 \cdot w(s_n)K}{\alpha A}}\right).$$

Let $\eta = c_2\sqrt{\frac{\alpha A}{w(s_n)K}}$ for a small enough constant $c_2$. We have

$$\mathbb{E}\left[\mathcal{R}(K)\right] = \Omega\left(\sqrt{\frac{A}{\alpha \cdot w(s_n)K}} \cdot K\right)$$

$$= \Omega\left(\sqrt{\frac{AK}{\alpha \cdot w(s_n)}}\right)$$

Recall that $0 < \gamma < \alpha$ and $n = H - 1$. Thus, we have $\min_{\pi,h,s:\, w_{\pi,h}(s)>0} w_{\pi,h}(s) = w(x_4) = \gamma^{H-1} < \alpha^{H-1}$. In addition, since $w(s_n) = \alpha^{n-1} = \alpha^{H-2}$, we have

$$\mathbb{E}\left[\mathcal{R}(K)\right] = \Omega\left(\sqrt{\frac{AK}{\alpha \cdot \alpha^{H-2}}}\right)$$

$$= \Omega\left(\sqrt{\frac{AK}{\alpha^{H-1}}}\right).$$

$\square$

# E  PROOFS FOR ITERATED CVAR RL WITH BEST POLICY IDENTIFICATION

In this section, we present the pseudo-code and detailed description of algorithm `ICVaR-BPI`, and formally state the sample complexity lower bound for Iterated CVaR-BPI (Theorem 5). We also give the proofs of sample complexity upper and lower bounds (Theorems 3 and 5).

## E.1  ALGORITHM `ICVaR-BPI`

Algorithm `ICVaR-BPI` (Algorithm 3) constructs optimistic and pessimistic value functions, estimation error, and a hypothesized optimal policy in each episode, and returns the hypothesized optimal policy when the estimation error shrinks within $\varepsilon$. Specifically, in each episode $k$, `ICVaR-BPI` calculates the empirical CVaR for values of next states $\mathrm{CVaR}_{s'\sim\hat{p}^k(\cdot|s,a)}^{\alpha}(\bar{V}_{h+1}^k(s')), \mathrm{CVaR}_{s'\sim\hat{p}^k(\cdot|s,a)}^{\alpha}(\underline{V}_{h+1}^k(s'))$ and exploration bonuses

---

**Algorithm 3:** `ICVaR-BPI`

---

**Input:** $\varepsilon, \delta, \alpha, \delta' := \frac{\delta}{7}, \tilde{L}(k) := \log(\frac{2HSAk^3}{\delta'})$ for any $k > 0$,

$\qquad J_{H+1}^k(s) = \bar{V}_{H+1}^k(s) = \underline{V}_{H+1}^k(s) = 0$ for any $k > 0$ and $s \in \mathcal{S}$.

1 **for** $k = 1, 2, \ldots, K$ **do**

2 $\quad$ **for** $h = H, H-1, \ldots, 1$ **do**

3 $\quad\quad$ **for** $s \in \mathcal{S}$ **do**

4 $\quad\quad\quad$ **for** $a \in \mathcal{A}$ **do**

5 $\quad\quad\quad\quad$ $\bar{Q}_h^k(s,a) \leftarrow \min \left\{ r(s,a) + \text{CVaR}_{s' \sim \hat{p}^k(\cdot|s,a)}^\alpha(\bar{V}_{h+1}^k(s')) + \frac{H}{\alpha}\sqrt{\frac{\tilde{L}(k)}{n_k(s,a)}}, \ H \right\}$;

6 $\quad\quad\quad\quad$ $\underline{Q}_h^k(s,a) \leftarrow \max \left\{ r(s,a) + \text{CVaR}_{s' \sim \hat{p}^k(\cdot|s,a)}^\alpha(\underline{V}_{h+1}^k(s')) - \frac{4H}{\alpha}\sqrt{\frac{S\tilde{L}(k)}{n_k(s,a)}}, \ 0 \right\}$;

7 $\quad\quad\quad\quad$ $G_h^k(s,a) \leftarrow \min \left\{ \frac{H(1+4\sqrt{S})\sqrt{\tilde{L}(k)}}{\alpha\sqrt{n_k(s,a)}} + \hat{\beta}^{k;\alpha,\underline{V}_{h+1}^k}(\cdot|s,a)^\top J_{h+1}^k, \ H \right\}$;

8 $\quad\quad\quad$ $\pi_h^k(s) \leftarrow \text{argmax}_{a \in \mathcal{A}} \bar{Q}_h^k(s,a)$. $\bar{V}_h^k(s) \leftarrow \max_{a \in \mathcal{A}} \bar{Q}_h^k(s,a)$.

$\quad\quad\quad$ $\underline{V}_h^k(s) \leftarrow \underline{Q}_h^k(s, \pi_h^k(s))$. $J_h^k(s) \leftarrow G_h^k(s, \pi_h^k(s))$;

9 $\quad$ **if** $J_1^k(s) \leq \varepsilon$ **then**

10 $\quad\quad$ **return** $\pi^k(s)$

11 $\quad$ **else**

12 $\quad\quad$ Play the episode $k$ with policy $\pi^k$, and update $n_{k+1}(s,a)$ and $\hat{p}^{k+1}(s'|s,a)$

---

$\frac{H}{\alpha}\sqrt{\frac{\tilde{L}(k)}{n_k(s,a)}}, \frac{4H}{\alpha}\sqrt{\frac{S\tilde{L}(k)}{n_k(s,a)}}$, to establish the optimistic and pessimistic Q-value functions $\bar{Q}_h^k(s,a)$ and $\underline{Q}_h^k(s,a)$, respectively. `ICVaR-BPI` further maintains a hypothesized optimal policy $\pi^k$, which is greedy with respect to $\bar{Q}_h^k(s,a)$. Let $\hat{\beta}^{k;\alpha,\underline{V}_{h+1}^k}(\cdot|s,a)$ denote the conditional empirical transition probability in episode $k$, conditioning on transitioning to the worst $\alpha$-portion successor states $s'$ (i.e., with the worst $\alpha$-portion values $\underline{V}_{h+1}^k(s')$), and it satisfies $\sum_{s' \in \mathcal{S}} \hat{\beta}^{k;\alpha,\underline{V}_{h+1}^k}(s'|s,a) \cdot \underline{V}_{h+1}^k(s') = \text{CVaR}_{s' \sim \hat{p}^k(\cdot|s,a)}^\alpha(\underline{V}_{h+1}^k(s'))$ (Line 7). Then, `ICVaR-BPI` computes estimation error $G_h^k(s,a)$ and $J_h^k(s)$ using conditional transition probability $\hat{\beta}^{k;\alpha,\underline{V}_{h+1}^k}(\cdot|s,a)$. Once estimation error $J_h^k(s)$ shrinks within accuracy parameter $\varepsilon$, `ICVaR-BPI` returns the hypothesized optimal policy $\pi^k$.

### E.2 Proofs of Sample Complexity Upper Bound

#### E.2.1 Concentration

In the best policy identification analysis, we introduce several useful lemmas and concentration events. Different from the regret minimization analysis where the logarithmic factor $\log(\frac{KHSA}{\delta'})$ in the exploration bonuses is an universal constant, here the logarithmic factor $\log(\frac{2k^3 HSA}{\delta'})$ will increase as the index of the episode $k$ increases.

**Lemma 12** (Concentration for $V^*$ – BPI). *It holds that*

$$\Pr\left[ \left| \text{CVaR}_{s' \sim \hat{p}^k(\cdot|s,a)}^\alpha(V_h^*(s')) - \text{CVaR}_{s' \sim p(\cdot|s,a)}^\alpha(V_h^*(s')) \right| \leq \frac{H}{\alpha}\sqrt{\frac{\log\left(\frac{2k^3 HSA}{\delta'}\right)}{n_k(s,a)}}, \right.$$

$$\left. \forall k > 0, \ \forall h \in [H], \ \forall (s,a) \in \mathcal{S} \times \mathcal{A} \right] \geq 1 - 2\delta'.$$

*Proof of Lemma 12.* Using the same analysis as Lemma 1, we have that for a fixed $k$,

$$\Pr\left[ \left| \text{CVaR}_{s' \sim \hat{p}^k(\cdot|s,a)}^\alpha(V_h^*(s')) - \text{CVaR}_{s' \sim p(\cdot|s,a)}^\alpha(V_h^*(s')) \right| \leq \frac{H}{\alpha}\sqrt{\frac{\log\left(\frac{2k^3 HSA}{\delta'}\right)}{n_k(s,a)}}, \right.$$

$$\left. \forall h \in [H], \ \forall (s,a) \in \mathcal{S} \times \mathcal{A} \right] \geq 1 - 2 \cdot \frac{\delta'}{2k^2}.$$

By a union bound over $k = 1, 2, \ldots$, we have

$$\Pr\left[\left|\text{CVaR}_{s' \sim \hat{p}^k(\cdot|s,a)}^\alpha(V_h^*(s')) - \text{CVaR}_{s' \sim p(\cdot|s,a)}^\alpha(V_h^*(s'))\right| \leq \frac{H}{\alpha}\sqrt{\frac{\log\left(\frac{2k^3 HSA}{\delta'}\right)}{n_k(s,a)}},\right.$$

$$\left.\forall k > 0, \ \forall h \in [H], \ \forall (s,a) \in \mathcal{S} \times \mathcal{A}\right]$$

$$\geq 1 - 2 \cdot \sum_{k=1}^\infty \left(\frac{\delta'}{2k^2}\right)$$

$$\geq 1 - 2\delta'.$$

$\square$

**Lemma 13** (Concentration for any $V$ − BPI). *It holds that*

$$\Pr\left[\left|\text{CVaR}_{s' \sim \hat{p}^k(\cdot|s,a)}^\alpha(V(s')) - \text{CVaR}_{s' \sim p(\cdot|s,a)}^\alpha(V(s'))\right| \leq \frac{2H}{\alpha}\sqrt{\frac{2S\log\left(\frac{2k^3 HSA}{\delta'}\right)}{n_k(s,a)}},\right.$$

$$\left.\forall V : \mathcal{S} \mapsto [0, H], \ \forall k > 0, \ \forall (s,a) \in \mathcal{S} \times \mathcal{A}\right] \geq 1 - 2\delta'.$$

*Proof of Lemma 13.* Using the same analysis as Lemma 2, we have that for a fixed $k$,

$$\Pr\left[\left|\text{CVaR}_{s' \sim \hat{p}^k(\cdot|s,a)}^\alpha(V(s')) - \text{CVaR}_{s' \sim p(\cdot|s,a)}^\alpha(V(s'))\right| \leq \frac{2H}{\alpha}\sqrt{\frac{2S\log\left(\frac{2k^3 HSA}{\delta'}\right)}{n_k(s,a)}},\right.$$

$$\left.\forall V : \mathcal{S} \mapsto [0, H], \ \forall (s,a) \in \mathcal{S} \times \mathcal{A}\right] \geq 1 - 2 \cdot \frac{\delta'}{2k^2}.$$

By a union bound over $k = 1, 2, \ldots$, we have

$$\Pr\left[\left|\text{CVaR}_{s' \sim \hat{p}^k(\cdot|s,a)}^\alpha(V(s')) - \text{CVaR}_{s' \sim p(\cdot|s,a)}^\alpha(V(s'))\right| \leq \frac{2H}{\alpha}\sqrt{\frac{2S\log\left(\frac{2k^3 HSA}{\delta'}\right)}{n_k(s,a)}},\right.$$

$$\left.\forall V : \mathcal{S} \mapsto [0, H], \ \forall k > 0, \ \forall (s,a) \in \mathcal{S} \times \mathcal{A}\right]$$

$$\geq 1 - 2 \cdot \sum_{k=1}^\infty \left(\frac{\delta'}{2k^2}\right)$$

$$\geq 1 - 2\delta'.$$

$\square$

For any risk level $\alpha \in (0, 1]$, function $V : \mathcal{S} \mapsto \mathbb{R}$, $k > 0$ and $(s', s, a) \in \mathcal{S} \times \mathcal{S} \times \mathcal{A}$, $\beta^{\alpha,V}(s'|s,a)$ and $\hat{\beta}^{k;\alpha,V}(s'|s,a)$ are the conditional transition probability from $(s,a)$ to $s'$ and the conditional empirical transition probability from $(s,a)$ to $s'$ in episode $k$, conditioning on transitioning to the worst $\alpha$-portion successor states $s'$ (i.e., with the lowest $\alpha$-portion values $V(s')$), respectively. $\mu^{\alpha,V}(s'|s,a)$ and $\hat{\mu}^{k;\alpha,V}(s'|s,a)$ denote how large the transition probability of successor state $s'$ and the empirical transition probability of successor state $s'$ in episode $k$ belong to the worst $\alpha$-portion, respectively. It holds that

$$\frac{\mu^{\alpha,V}(s'|s,a)}{\alpha} = \beta^{\alpha,V}(s'|s,a),$$

and

$$\frac{\hat{\mu}^{k;\alpha,V}(s'|s,a)}{\alpha} = \hat{\beta}^{k;\alpha,V}(s'|s,a).$$

**Lemma 14** (Concentration for conditional transition probability). *It holds that*

$$\Pr\left[\left|\hat{\beta}^{k;\alpha,V}(s'|s,a) - \beta^{\alpha,V}(s'|s,a)\right| \leq \frac{2}{\alpha}\sqrt{\frac{2S\log\left(\frac{2k^3HSA}{\delta'}\right)}{n_k(s,a)}},\right.$$

$$\left.\forall V : \mathcal{S} \mapsto \mathbb{R}, \ \forall k > 0, \ \forall(s,a) \in \mathcal{S} \times \mathcal{A}\right] \geq 1 - 2\delta'.$$

*Proof of Lemma 14.* Using the analysis of Eq. (7), we have that for any risk level $\alpha \in (0,1]$, function $V : \mathcal{S} \mapsto \mathbb{R}$, $k > 0$ and $(s,a) \in \mathcal{S} \times \mathcal{A}$,

$$\sum_{s'\in\mathcal{S}}\left|\hat{\mu}^{k;\alpha,V}(s'|s,a) - \mu^{\alpha,V}(s'|s,a)\right| \leq 2\sum_{s'\in\mathcal{S}}\left|\hat{p}^k(s'|s,a) - p(s'|s,a)\right|. \tag{21}$$

Using Eq. (55) in (Zanette & Brunskill, 2019) (originated from (Weissman et al., 2003)), we have that for any fixed $k$, with probability at least $1 - 2 \cdot (\frac{\delta'}{2k^2})$, for any $(s,a) \in \mathcal{S} \times \mathcal{A}$,

$$\sum_{s'\in\mathcal{S}}\left|\hat{p}^k(s'|s,a) - p(s'|s,a)\right| \leq \sqrt{\frac{2S\log\left(\frac{2k^3HSA}{\delta'}\right)}{n_k(s,a)}},$$

and thus,

$$\sum_{s'\in\mathcal{S}}\left|\hat{\beta}^{k;\alpha,V}(s'|s,a) - \beta^{\alpha,V}(s'|s,a)\right| = \sum_{s'\in\mathcal{S}}\left|\frac{\hat{\mu}^{k;\alpha,V}(s'|s,a)}{\alpha} - \frac{\mu^{\alpha,V}(s'|s,a)}{\alpha}\right|$$

$$= \frac{\sum_{s'\in\mathcal{S}}\left|\hat{\mu}^{k;\alpha,V}(s'|s,a) - \mu^{\alpha,V}(s'|s,a)\right|}{\alpha}$$

$$\leq \frac{2\sum_{s'\in\mathcal{S}}\left|\hat{p}^k(s'|s,a) - p(s'|s,a)\right|}{\alpha}$$

$$\leq \frac{2}{\alpha}\sqrt{\frac{2S\log\left(\frac{2k^3HSA}{\delta'}\right)}{n_k(s,a)}}$$

By a union bound over $k = 1, 2, \ldots$, we have

$$\Pr\left[\sum_{s'\in\mathcal{S}}\left|\hat{\beta}^{k;\alpha,V}(s'|s,a) - \beta^{\alpha,V}(s'|s,a)\right| \leq \frac{2}{\alpha}\sqrt{\frac{2S\log\left(\frac{2k^3HSA}{\delta'}\right)}{n_k(s,a)}},\right.$$

$$\left.\forall V : \mathcal{S} \mapsto \mathbb{R}, \ \forall k > 0, \ \forall(s,a) \in \mathcal{S} \times \mathcal{A}\right]$$

$$\geq 1 - 2 \cdot \sum_{k=1}^{\infty}\left(\frac{\delta'}{2k^2}\right)$$

$$\geq 1 - 2\delta'.$$

$\square$

To sum up, we define the following concentration events and recall event $\mathcal{E}_3$.

$$\mathcal{F}_1 := \left\{\left|\text{CVaR}^{\alpha}_{s'\sim\hat{p}^k(\cdot|s,a)}(V_h^*(s')) - \text{CVaR}^{\alpha}_{s'\sim p(\cdot|s,a)}(V_h^*(s'))\right| \leq \frac{H}{\alpha}\sqrt{\frac{\log\left(\frac{2k^3HSA}{\delta'}\right)}{n_k(s,a)}},\right.$$

$$\left.\forall k > 0, \ \forall h \in [H], \ \forall(s,a) \in \mathcal{S} \times \mathcal{A}\right\}$$

$$\mathcal{F}_2 := \left\{\left|\text{CVaR}^{\alpha}_{s'\sim\hat{p}^k(\cdot|s,a)}(V(s')) - \text{CVaR}^{\alpha}_{s'\sim p(\cdot|s,a)}(V(s'))\right| \leq \frac{2H}{\alpha}\sqrt{\frac{2S\log\left(\frac{2k^3HSA}{\delta'}\right)}{n_k(s,a)}},\right.$$

$$\forall V : \mathcal{S} \mapsto [0, H], \ \forall k > 0, \ \forall (s, a) \in \mathcal{S} \times \mathcal{A} \Big\}$$

$$\mathcal{F}_3 := \left\{ \left| \hat{\beta}^{k; \alpha, V}(s'|s, a) - \beta^{\alpha, V}(s'|s, a) \right| \leq \frac{2}{\alpha} \sqrt{\frac{2S \log\left(\frac{2k^3 HSA}{\delta'}\right)}{n_k(s, a)}}, \right.$$

$$\left. \forall V : \mathcal{S} \mapsto \mathbb{R}, \ \forall k > 0, \ \forall (s, a) \in \mathcal{S} \times \mathcal{A} \right\}$$

$$\mathcal{E}_3 := \left\{ n_k(s, a) \geq \frac{1}{2} \sum_{k'=1}^{k-1} \sum_{h=1}^{H} w_{k'h}(s, a) - H \log\left(\frac{HSA}{\delta'}\right), \ \forall k > 0, \ \forall (s, a) \in \mathcal{S} \times \mathcal{A} \right\}$$

$$\mathcal{F} := \mathcal{F}_1 \cap \mathcal{F}_2 \cap \mathcal{F}_3 \cap \mathcal{E}_3$$

**Lemma 15.** *Letting $\delta' = \frac{\delta}{7}$, it holds that*

$$\Pr[\mathcal{F}] \geq 1 - \delta.$$

*Proof of Lemma 15.* This lemma can be obtained by combining Lemmas 12-14 and 3. □

### E.2.2 OPTIMISM AND ESTIMATION ERROR

For any $k > 0$, let $\tilde{L}(k) := \log\left(\frac{2HSAk^3}{\delta'}\right)$.

**Lemma 16** (Optimism and Pessimism). *Suppose that event $\mathcal{F}$ holds. Then, for any $k > 0$, $h \in [H]$ and $s \in \mathcal{S}$,*

$$\bar{V}_h^k(s) \geq V_h^*(s),$$
$$\underline{V}_h^k(s) \leq V_h^{\pi^k}(s).$$

*Proof of Lemma 16.* The proof of $\bar{V}_h^k(s) \geq V_h^*(s)$ is similar to Lemma 5. Below we prove $\underline{V}_h^k(s) \leq V_h^{\pi^k}(s)$ by induction.

First, for any $k > 0$, $s \in \mathcal{S}$, it holds that $\underline{V}_{H+1}^k(s) = V_{H+1}^{\pi^k}(s) = 0$.

Then, for any $k > 0$, $h \in [H]$ and $(s, a) \in \mathcal{S} \times \mathcal{A}$, if $\underline{Q}_h^k(s, a) = 0$, $\underline{Q}_h^k(s, a) \leq Q^{\pi^k}(s, a)$ trivially holds, and otherwise,

$$\underline{Q}_h^k(s, a) = r(s, a) + \mathrm{CVaR}_{s' \sim \hat{p}^k(\cdot|s,a)}^\alpha(\underline{V}_{h+1}^k(s')) - \frac{4H}{\alpha}\sqrt{\frac{S\tilde{L}(k)}{n_k(s, a)}}$$

$$\overset{(a)}{\leq} r(s, a) + \mathrm{CVaR}_{s' \sim \hat{p}^k(\cdot|s,a)}^\alpha(V_{h+1}^{\pi^k}(s')) - \frac{4H}{\alpha}\sqrt{\frac{S\tilde{L}(k)}{n_k(s, a)}}$$

$$\overset{(b)}{\leq} r(s, a) + \mathrm{CVaR}_{s' \sim \hat{p}^k(\cdot|s,a)}^\alpha(V_{h+1}^{\pi^k}(s'))$$
$$\quad - \left( \mathrm{CVaR}_{s' \sim \hat{p}^k(\cdot|s,a)}^\alpha(V_{h+1}^{\pi^k}(s')) - \mathrm{CVaR}_{s' \sim p(\cdot|s,a)}^\alpha(V_{h+1}^{\pi^k}(s')) \right)$$

$$= r(s, a) + \mathrm{CVaR}_{s' \sim p(\cdot|s,a)}^\alpha(V_{h+1}^{\pi^k}(s'))$$

$$= Q^{\pi^k}(s, a),$$

where (a) uses the induction hypothesis, and (b) comes from Lemma 2.

Thus, we have

$$\underline{V}_h^k(s) = \underline{Q}_h^k(s, \pi_h^k(s)) \leq Q^{\pi^k}(s, \pi_h^k(s)) = V_h^{\pi^k}(s),$$

which concludes the proof. □

For any risk level $\alpha \in (0,1]$, $k > 0$, $h \in [H]$ and $(s', s, a) \in \mathcal{S} \times \mathcal{S} \times \mathcal{A}$, $\hat{\beta}^{k;\alpha,\underline{V}_{h+1}^k}(s'|s,a)$ is the conditional empirical transition probability from $(s,a)$ to $s'$ in episode $k$, conditioning on transitioning to the worst $\alpha$-portion successor states $s'$ (i.e., with the lowest $\alpha$-portion values $\underline{V}_{h+1}^k(s')$). It holds that $\mathrm{CVaR}_{s' \sim \hat{p}^k(\cdot|s,a)}^\alpha(\underline{V}_{h+1}^k(s')) = \sum_{s' \in \mathcal{S}} \hat{\beta}^{k;\alpha,\underline{V}_{h+1}^k}(s'|s,a) \cdot \underline{V}_{h+1}^k(s')$.

**Lemma 17** (Estimation Error). *Suppose that event $\mathcal{F}$ holds. Then, for any $k > 0$,*

$$V_1^*(s_1) - V_1^{\pi^k}(s_1) \le J_1^k(s_1).$$

*Proof of Lemma 17.* In the following, we prove by induction that for any $k > 0$, $h \in [H]$ and $s \in \mathcal{S}$,

$$\bar{V}_h^k(s) - \underline{V}_h^k(s) \le J_h^k(s). \tag{22}$$

First, for any $k > 0$ and $s \in \mathcal{S}$, it holds that $\bar{V}_{H+1}^k(s) - \underline{V}_{H+1}^k(s) = J_{H+1}^k(s) = 0$.

Then, for any $k > 0$, $h \in [H]$ and $(s,a) \in \mathcal{S} \times \mathcal{A}$, if $G_h^k(s,a) = H$, $\bar{Q}_h^k(s,a) - \underline{Q}_h^k(s,a) \le G_h^k(s,a)$ holds trivially, and otherwise,

$$
\begin{aligned}
\bar{Q}_h^k(s,a) - \underline{Q}_h^k(s,a) =& \frac{H}{\alpha}\sqrt{\frac{\tilde{L}(k)}{n_k(s,a)}} + \frac{4H}{\alpha}\sqrt{\frac{S\tilde{L}(k)}{n_k(s,a)}} \\
&+ \mathrm{CVaR}_{s' \sim \hat{p}^k(\cdot|s,a)}^\alpha(\bar{V}_{h+1}^k(s')) - \mathrm{CVaR}_{s' \sim \hat{p}^k(\cdot|s,a)}^\alpha(\underline{V}_{h+1}^k(s')) \\
\overset{(a)}{\le}& \frac{H\sqrt{\tilde{L}(k)}(1 + 4\sqrt{S})}{\alpha\sqrt{n_k(s,a)}} + \hat{\beta}^{k;\alpha,\underline{V}_{h+1}^k}(\cdot|s,a)^\top \left(\bar{V}_{h+1}^k - \underline{V}_{h+1}^k\right) \\
\overset{(b)}{\le}& \frac{H\sqrt{\tilde{L}(k)}(1 + 4\sqrt{S})}{\alpha\sqrt{n_k(s,a)}} + \hat{\beta}^{k;\alpha,\underline{V}_{h+1}^k}(\cdot|s,a)^\top J_{h+1}^k \\
=& G_h^k(s,a),
\end{aligned}
$$

where (a) uses Lemma 11 with empirical transition probability $\hat{p}^k(\cdot|s,a)$, conditional empirical transition probability $\hat{\beta}^{k;\alpha,\underline{V}_{h+1}^k}(\cdot|s,a)$, and values $\bar{V}_{h+1}^k, \underline{V}_{h+1}^k$, and (b) is due to the induction hypothesis.

Thus,

$$\bar{V}_h^k(s) - \underline{V}_h^k(s) = \bar{Q}_h^k(s, \pi_h^k(s)) - \underline{Q}_h^k(s, \pi_h^k(s)) \le G_h^k(s, \pi_h^k(s)) = J_h^k(s),$$

which completes the proof of Eq. (22).

Hence, for any $k > 0$,

$$\bar{V}_1^k(s_1) - \underline{V}_1^k(s_1) \le J_1^k(s_1).$$

Using Lemma 16, we have

$$V_1^*(s) - V_1^{\pi^k}(s_1) \le \bar{V}_1^k(s_1) - \underline{V}_1^k(s_1) \le J_1^k(s_1).$$

$\square$

### E.2.3   PROOF OF THEOREM 3

*Proof of Theorem 3.* Suppose that event $\mathcal{F}$ holds.

First, we prove the correctness. Using Lemma 17, when algorithm ICVaR-BPI stops, we have

$$V_1^*(s_1) - V_1^{\pi^k}(s_1) \le J_1^k(s_1) \le \varepsilon.$$

Thus, the output policy $\pi^k$ is $\varepsilon$-optimal.

Next, we prove the sample complexity.

Unfolding $J_1^k(s_1)$, we have

$$J_1^k(s_1) \stackrel{(a)}{=} \min\left\{\frac{H(1+4\sqrt{S})\sqrt{\tilde{L}(k)}}{\alpha\sqrt{n_k(s_1,a_1)}} + \sum_{s_2\in\mathcal{S}}\hat{\beta}^{k;\alpha,\underline{V}_2^k}(s_2|s_1,a_1)\cdot J_2^k(s_2),\ H\right\}$$

$$= \min\left\{\frac{H(1+4\sqrt{S})\sqrt{\tilde{L}(k)}}{\alpha\sqrt{n_k(s_1,a_1)}} + \sum_{s_2\in\mathcal{S}}\beta^{\alpha,\underline{V}_2^k}(s_2|s_1,a_1)\cdot J_2^k(s_2)\right.$$
$$\left. + \sum_{s_2\in\mathcal{S}}\left(\hat{\beta}^{k;\alpha,\underline{V}_2^k}(s_2|s_1,a_1) - \beta^{\alpha,\underline{V}_2^k}(s_2|s_1,a_1)\right)\cdot J_2^k(s_2),\ H\right\}$$

$$\stackrel{(b)}{\leq} \min\left\{\frac{H(1+4\sqrt{S})\sqrt{\tilde{L}(k)}}{\alpha\sqrt{n_k(s_1,a_1)}} + \sum_{s_2\in\mathcal{S}}\beta^{\alpha,\underline{V}_2^k}(s_2|s_1,a_1)\cdot J_2^k(s_2)\right.$$
$$\left. + H\sum_{s_2\in\mathcal{S}}\left|\left(\hat{\beta}^{k;\alpha,\underline{V}_2^k}(s_2|s_1,a_1) - \beta^{\alpha,\underline{V}_2^k}(s_2|s_1,a_1)\right)\right|,\ H\right\}$$

$$\stackrel{(c)}{\leq} \min\left\{\frac{H(1+4\sqrt{S})\sqrt{\tilde{L}(k)}}{\alpha\sqrt{n_k(s_1,a_1)}} + \sum_{s_2\in\mathcal{S}}\beta^{\alpha,\underline{V}_2^k}(s_2|s_1,a_1)\cdot J_2^k(s_2) + \frac{4H}{\alpha}\sqrt{\frac{S\cdot\tilde{L}(k)}{n_k(s_1,a_1)}},\ H\right\}$$

$$\stackrel{(d)}{\leq} \min\left\{\frac{H(1+8\sqrt{S})\sqrt{\tilde{L}(k)}}{\alpha\sqrt{n_k(s_1,a_1)}},\ H\right\} + \sum_{s_2\in\mathcal{S}}\beta^{\alpha,\underline{V}_2^k}(s_2|s_1,a_1)\cdot J_2^k(s_2)$$

$$\stackrel{(e)}{\leq} \min\left\{\frac{H(1+8\sqrt{S})\sqrt{\tilde{L}(k)}}{\alpha\sqrt{n_k(s_1,a_1)}},\ H\right\} + \sum_{s_2\in\mathcal{S}}\beta^{\alpha,\underline{V}_2^k}(s_2|s_1,a_1)\cdot$$
$$\left(\min\left\{\frac{H(1+8\sqrt{S})\sqrt{\tilde{L}(k)}}{\alpha\sqrt{n_k(s_2,a_2)}},\ H\right\} + \sum_{s_3\in\mathcal{S}}\beta^{\alpha,\underline{V}_3^k}(s_3|s_2,a_2)\cdot J_3^k(s_3)\right)$$

$$\stackrel{(f)}{\leq} \min\left\{\frac{H(1+8\sqrt{S})\sqrt{\tilde{L}(k)}}{\alpha\sqrt{n_k(s_1,a_1)}},\ H\right\} + \sum_{s_2\in\mathcal{S}}\beta^{\alpha,\underline{V}_2^k}(s_2|s_1,a_1)\cdot$$
$$\left(\min\left\{\frac{H(1+8\sqrt{S})\sqrt{\tilde{L}(k)}}{\alpha\sqrt{n_k(s_2,a_2)}},\ H\right\} + \sum_{s_3\in\mathcal{S}}\beta^{\alpha,\underline{V}_3^k}(s_3|s_2,a_2)\cdot\right.$$
$$\left.\left(\cdots\sum_{s_H\in\mathcal{S}}\beta^{\alpha,\underline{V}_H^k}(s_H|s_{H-1},a_{H-1})\cdot\min\left\{\frac{H(1+8\sqrt{S})\sqrt{\tilde{L}(k)}}{\alpha\sqrt{n_k(s_H,a_H)}},\ H\right\}\right)\right)$$

$$\stackrel{(g)}{=} \sum_{h=1}^H\sum_{(s,a)\in\mathcal{S}\times\mathcal{A}} w_{kh}^{\text{CVaR},\alpha,\underline{V}^k}(s,a)\cdot\min\left\{\frac{H(1+8\sqrt{S})\sqrt{\tilde{L}(k)}}{\alpha\sqrt{n_k(s_H,a_H)}},\ H\right\}$$

$$\leq \sum_{h=1}^H\sum_{(s,a)\in\mathcal{L}_k} w_{kh}^{\text{CVaR},\alpha,\underline{V}^k}(s,a)\cdot\frac{H(1+8\sqrt{S})\sqrt{\tilde{L}(k)}}{\alpha\sqrt{n_k(s,a)}} + \sum_{h=1}^H\sum_{(s,a)\notin\mathcal{L}_k} w_{kh}^{\text{CVaR},\alpha,\underline{V}^k}(s,a)\cdot H$$

Here (b) is due to that for any $k > 0$, $h \in [H]$ and $s \in \mathcal{S}$, $J_h^k(s) \in [0, H]$. (c) comes from Lemma 14. (e) and (f) follow from recurrently applying steps (a)-(d). (g) uses the fact that $w_{kh}^{\text{CVaR},\alpha,\underline{V}^k}(s,a)$ is defined as the probability of visiting $(s,a)$ at step $h$ of episode $k$ under the conditional transition probability $\beta^{\alpha,\underline{V}_{h'+1}^k}(\cdot|\cdot,\cdot)$ for each step $h' = 1,\ldots,h-1$.

Let $\tau$ denote the episode in which algorithm `ICVaR-BPI` stops (`ICVaR-BPI` will not sample any trajectory in the stopping episode $\tau$). Then, for any $k < \tau$, we have $\varepsilon < J_1^k(s_1)$. Summing over $k < \tau$, we have

$$(\tau - 1) \cdot \varepsilon < \sum_{k=1}^{\tau-1} J_1^k(s_1)$$

$$\leq \sum_{k=1}^{\tau-1} \sum_{h=1}^{H} \sum_{(s,a)\in\mathcal{L}_k} w_{kh}^{\text{CVaR},\alpha,\underline{V}^k}(s,a) \cdot \frac{H(1+8\sqrt{S})\sqrt{\tilde{L}(k)}}{\alpha\sqrt{n_k(s,a)}}$$

$$+ \sum_{k=1}^{\tau-1} \sum_{h=1}^{H} \sum_{(s,a)\notin\mathcal{L}_k} w_{kh}^{\text{CVaR},\alpha,\underline{V}^k}(s,a) \cdot H$$

$$\overset{(a)}{\leq} \frac{H(1+8\sqrt{S})\sqrt{\tilde{L}(\tau-1)}}{\alpha} \sqrt{\sum_{k=1}^{\tau-1} \sum_{h=1}^{H} \sum_{(s,a)\in\mathcal{L}_k} \frac{w_{kh}^{\text{CVaR},\alpha,\underline{V}^k}(s,a)}{n_k(s,a)}} \cdot$$

$$\sqrt{\sum_{k=1}^{\tau-1} \sum_{h=1}^{H} \sum_{(s,a)\in\mathcal{L}_k} w_{kh}^{\text{CVaR},\alpha,\underline{V}^k}(s,a)}$$

$$+ \min\left\{ \frac{1}{\min\limits_{\pi,h,s:\ w_{\pi,h}(s)>0} w_{\pi,h}(s)}, \frac{1}{\alpha^{H-1}} \right\} \left( 4SAH^2 \log\left(\frac{HSA}{\delta'}\right) + 5SAH^2 \right)$$

$$\overset{(b)}{\leq} \frac{H(1+8\sqrt{S})\sqrt{\tilde{L}(\tau-1)}}{\alpha} \cdot \sqrt{(\tau-1)H} \cdot \sqrt{\sum_{k=1}^{\tau-1} \sum_{h=1}^{H} \sum_{(s,a)\in\mathcal{L}_k} \frac{w_{kh}^{\text{CVaR},\alpha,\underline{V}^k}(s,a)}{n_k(s,a)}}$$

$$+ \min\left\{ \frac{1}{\min\limits_{\pi,h,s:\ w_{\pi,h}(s)>0} w_{\pi,h}(s)}, \frac{1}{\alpha^{H-1}} \right\} \left( 4SAH^2 \log\left(\frac{HSA}{\delta'}\right) + 5SAH^2 \right)$$

$$\overset{(c)}{\leq} \frac{(1+8\sqrt{S})H\sqrt{H \cdot \tilde{L}(\tau-1) \cdot (\tau-1)}}{\alpha} \cdot$$

$$\sqrt{\sum_{k=1}^{\tau-1} \sum_{h=1}^{H} \sum_{(s,a)\in\mathcal{L}_k} \frac{w_{kh}^{\text{CVaR},\alpha,\underline{V}^k}(s,a)}{w_{kh}(s,a)} \cdot \frac{w_{kh}(s,a)}{n_k(s,a)} \mathbb{1}\left\{w_{kh}(s,a) \neq 0\right\}}$$

$$+ \min\left\{ \frac{1}{\min\limits_{\pi,h,s:\ w_{\pi,h}(s)>0} w_{\pi,h}(s)}, \frac{1}{\alpha^{H-1}} \right\} \left( 4SAH^2 \log\left(\frac{HSA}{\delta'}\right) + 5SAH^2 \right)$$

$$\leq \frac{(1+8\sqrt{S})H\sqrt{H \cdot \tilde{L}(\tau-1) \cdot (\tau-1)}}{\alpha} \cdot \min\left\{ \frac{1}{\sqrt{\min\limits_{\pi,h,s:\ w_{\pi,h}(s)>0} w_{\pi,h}(s)}}, \frac{1}{\sqrt{\alpha^{H-1}}} \right\} \cdot$$

$$\sqrt{\sum_{k=1}^{\tau-1} \sum_{h=1}^{H} \sum_{(s,a)\in\mathcal{L}_k} \frac{w_{kh}(s,a)}{n_k(s,a)}} + \min\left\{ \frac{1}{\min\limits_{\pi,h,s:\ w_{\pi,h}(s)>0} w_{\pi,h}(s)}, \frac{1}{\alpha^{H-1}} \right\} \cdot$$

$$\left( 4SAH^2 \log\left(\frac{HSA}{\delta'}\right) + 5SAH^2 \right)$$

$$\overset{(d)}{\leq} \frac{(1+8\sqrt{S})H\sqrt{H \cdot \tilde{L}(\tau-1) \cdot (\tau-1)}}{\alpha} \min\left\{ \frac{1}{\sqrt{\min\limits_{\pi,h,s:\ w_{\pi,h}(s)>0} w_{\pi,h}(s)}}, \frac{1}{\sqrt{\alpha^{H-1}}} \right\} \cdot$$

$$2\sqrt{SA\tilde{L}(\tau-1)} + \min\left\{\frac{1}{\min\limits_{\pi,h,s:\,w_{\pi,h}(s)>0} w_{\pi,h}(s)},\ \frac{1}{\alpha^{H-1}}\right\}\cdot$$

$$\left(4SAH^2\log\left(\frac{HSA}{\delta'}\right) + 5SAH^2\right)$$

$$\leq \min\left\{\frac{1}{\sqrt{\min\limits_{\pi,h,s:\,w_{\pi,h}(s)>0} w_{\pi,h}(s)}},\ \frac{1}{\sqrt{\alpha^{H-1}}}\right\}\frac{18SH\cdot\tilde{L}(\tau-1)\sqrt{HA(\tau-1)}}{\alpha}$$

$$+ \min\left\{\frac{1}{\min\limits_{\pi,h,s:\,w_{\pi,h}(s)>0} w_{\pi,h}(s)},\ \frac{1}{\alpha^{H-1}}\right\}\left(4SAH^2\log\left(\frac{HSA}{\delta'}\right) + 5SAH^2\right),$$

where (a) is due to Lemma 10, (b) uses the fact that for any risk level $\alpha\in(0,1]$, $k>0$ and $h\in[H]$, $\sum_{(s,a)\in\mathcal{S}\times\mathcal{A}} w_{kh}^{CVaR,\alpha,\underline{V}^k}(s,a)=1$, (c) comes from Lemma 8, and (d) is due to Lemma 7.

Thus, when $\log\left(\frac{HSA}{\delta'}\right)\geq 1$, we have

$$\tau-1 \leq \min\left\{\frac{1}{\sqrt{\min\limits_{\pi,h,s:\,w_{\pi,h}(s)>0} w_{\pi,h}(s)}},\ \frac{1}{\sqrt{\alpha^{H-1}}}\right\}\frac{18SH\sqrt{HA}}{\varepsilon\alpha}\cdot\sqrt{\tau-1}\cdot\log\left(\frac{2HSA(\tau-1)^3}{\delta'}\right)$$

$$+ \min\left\{\frac{1}{\min\limits_{\pi,h,s:\,w_{\pi,h}(s)>0} w_{\pi,h}(s)},\ \frac{1}{\alpha^{H-1}}\right\}\frac{4SAH^2\log\left(\frac{HSA}{\delta'}\right)+5SAH^2}{\varepsilon}$$

$$\leq \min\left\{\frac{1}{\sqrt{\min\limits_{\pi,h,s:\,w_{\pi,h}(s)>0} w_{\pi,h}(s)}},\ \frac{1}{\sqrt{\alpha^{H-1}}}\right\}\frac{54SH\sqrt{HA}}{\varepsilon\alpha}\cdot\sqrt{\tau-1}\cdot\log\left(\frac{2HSA(\tau-1)}{\delta'}\right)$$

$$+ \min\left\{\frac{1}{\min\limits_{\pi,h,s:\,w_{\pi,h}(s)>0} w_{\pi,h}(s)},\ \frac{1}{\alpha^{H-1}}\right\}\frac{9SAH^2}{\varepsilon}\log\left(\frac{HSA}{\delta'}\right)$$

Using Lemma 24 with $A=1$, $B=0$, $C=\min\{\frac{1}{\sqrt{\min_{\pi,h,s:\,w_{\pi,h}(s)>0} w_{\pi,h}(s)}},\ \frac{1}{\sqrt{\alpha^{H-1}}}\}\frac{54SH\sqrt{HA}}{\varepsilon\alpha}$, $D=\min\{\frac{1}{\min_{\pi,h,s:\,w_{\pi,h}(s)>0} w_{\pi,h}(s)},\ \frac{1}{\alpha^{H-1}}\}\frac{9SAH^2}{\varepsilon}$, $E=0$, $\beta=\frac{2HSA}{\delta'}$ and $T=\tau-1$, and recalling that algorithm `ICVaR-BPI` does not sample any trajectory in the stopping episode $\tau$, we have that the number of used trajectories is bounded by

$$\tau-1 = O\left(\min\left\{\frac{1}{\min\limits_{\pi,h,s:\,w_{\pi,h}(s)>0} w_{\pi,h}(s)},\ \frac{1}{\alpha^{H-1}}\right\}\frac{H^3S^2A}{\varepsilon^2\alpha^2}\cdot\right.$$

$$\left.\log^2\left(\min\left\{\frac{1}{\min\limits_{\pi,h,s:\,w_{\pi,h}(s)>0} w_{\pi,h}(s)},\ \frac{1}{\alpha^{H-1}}\right\}\frac{HSA}{\varepsilon\alpha\delta}\right)\right).$$

$\square$

### E.3 SAMPLE COMPLEXITY LOWER BOUND

Below we present the sample complexity lower bound for Iterated CVaR RL-BPI and provide its proof.

We say algorithm $\mathcal{A}$ is $(\delta,\varepsilon)$-correct if $\mathcal{A}$ returns an $\varepsilon$-optimal policy $\hat{\pi}$ such that $V_1^{\hat{\pi}}(s_1)\geq V_1^*(s_1)-\varepsilon$ with probability $1-\delta$.

**Theorem 5** (Sample Complexity Lower Bound). *There exists an instance of Iterated CVaR RL-BPI, where $\min_{\pi,h,s:\ w_{\pi,h}(s)>0} w_{\pi,h}(s) > \alpha^{H-1}$ and the number of trajectories used by any $(\delta,\varepsilon)$-correct algorithm is at least*

$$\Omega\left(\frac{H^2 A}{\varepsilon^2\alpha \min\limits_{\pi,h,s:\ w_{\pi,h}(s)>0} w_{\pi,h}(s)} \log\left(\frac{1}{\delta}\right)\right).$$

*In addition, there also exists an instance of Iterated CVaR RL-BPI, where $\alpha^{H-1} > \min_{\pi,h,s:\ w_{\pi,h}(s)>0} w_{\pi,h}(s)$ and the number of trajectories used by any $(\delta,\varepsilon)$-correct algorithm is at least*

$$\Omega\left(\frac{A}{\alpha^{H-1}\varepsilon^2} \log\left(\frac{1}{\delta}\right)\right).$$

Theorem 5 corroborates that when $\alpha$ is small, the factor $\min_{\pi,h,s:\ w_{\pi,h}(s)>0} w_{\pi,h}(s)$ is unavoidable in general. This reveals the intrinsic hardness of Iterated CVaR RL, i.e., when the agent is highly risk-sensitive, she needs to spend a number of trajectories on exploring critical but hard-to-reach states in order to identify an optimal policy.

*Proof of Theorem 5.* This proof uses a similar analytical procedure as Theorem 2 in (Dann & Brunskill, 2015).

First, we consider the instance in Figure 6 as in the proof of Theorem 2, where $\min_{\pi,h,s:\ w_{\pi,h}(s)>0} w_{\pi,h}(s) > \alpha^{H-1}$. Below we prove that on this instance any algorithm must suffer a $O(\frac{1}{\min\limits_{\pi,h,s:\ w_{\pi,h}(s)>0} w_{\pi,h}(s)} \cdot \frac{H^3 S^2 A}{\varepsilon^2\alpha^2} \log(\frac{1}{\delta}))$ regret.

Fix an algorithm $\mathcal{A}$. Define $\mathcal{E}_{s_n} := \{\hat{\pi}(s_n) = a_J\}$ as the event that the output policy $\hat{\pi}$ of algorithm $\mathcal{A}$ chooses the optimal action in state $s_n$.

Using the similar analysis as in the proof of Theorem 2 (Eq. (15)), we have

$$V_1^*(s_1) - V_1^\pi(s_1) = 0.6(H-n) \cdot \frac{\eta}{\alpha} \cdot (1 - \mathbb{1}\{\mathcal{E}_{s_n}\}).$$

For $\pi$ to be $\varepsilon$-optimal, we need

$$\varepsilon \geq V_1^*(s_1) - V_1^\pi(s_1) = 0.6(H-n) \cdot \frac{\eta}{\alpha} \cdot (1 - \mathbb{1}\{\mathcal{E}_{s_n}\}),$$

which is equivalent to

$$\mathbb{1}\{\mathcal{E}_{s_n}\} \geq 1 - \frac{\varepsilon\alpha}{0.6(H-n)\cdot\eta}.$$

Let $\eta = \frac{8e^4\varepsilon\alpha}{0.6c_0(H-n)}$ for some constant $c_0$ and small enough $\varepsilon$. Then, for $\pi$ to be $\varepsilon$-optimal, we need

$$\mathbb{1}\{\mathcal{E}_{s_n}\} \geq 1 - \frac{c_0}{8e^4}.$$

Let $\phi := 1 - \frac{c_0}{8e^4}$. For algorithm $\mathcal{A}$ to be $(\varepsilon,\delta)$-correct, we need

$$\begin{aligned}
1 - \delta &\leq \Pr[V^* - V^\pi \geq \varepsilon] \\
&\leq \Pr[\mathbb{1}\{\mathcal{E}_{s_n}\} \geq \phi] \\
&\leq \frac{\mathbb{E}[\mathcal{E}_{s_n}]}{\phi} \\
&\leq \frac{1}{\phi}\Pr[\mathcal{E}_{s_n}],
\end{aligned}$$

which is equivalent to

$$\Pr[\bar{\mathcal{E}}_{s_n}] = 1 - \Pr[\mathcal{E}_{s_n}] \leq 1 - \phi + \phi\delta.$$

Recall that $0 < \alpha < \frac{1}{3}$. For any $j \in [A]$, $\text{KL}(p_{unif}(s_n, a_j)\|p_j(s_n, a_j)) = \text{KL}(\text{Ber}(\alpha)\|\text{Ber}(\alpha - \eta)) \leq \frac{\eta^2}{(\alpha - \eta)(1 - \alpha + \eta)} \leq \frac{c_1 \cdot \eta^2}{\alpha}$ for some constant $c_1$ and small enough $\eta$. Let $V_{s_n}$ be the number of times that algorithm $\mathcal{A}$ visited state $s_n$. To ensure $\Pr[\bar{\mathcal{E}}_{s_n}] \leq 1 - \phi + \phi\delta$, we need

$$\mathbb{E}[V_{s_n}] \geq \sum_{j=1}^{A} \frac{1}{\text{KL}(p_{unif}(s_n, a_j)\|p_j(s_n, a_j))} \log\left(\frac{c_2}{1 - \phi + \phi\delta}\right)$$

$$\geq \frac{\alpha A}{c_1 \cdot \eta^2} \log\left(\frac{c_2}{1 - \phi + \phi\delta}\right)$$

$$= \frac{\alpha A \cdot 0.6^2 c_0^2 (H - n)^2}{c_1 \cdot 64 e^8 \varepsilon^2 \alpha^2} \log\left(\frac{c_2}{\frac{c_0}{8e^4} + \delta}\right)$$

for some constant $c_2$.

Let $c_0$ be a small constant such that $\frac{c_0}{8e^4} < \delta$. Let $w(s_n)$ denote the probability of visiting $s_n$ in an episode, and this probability is the same for all policies. Let $\tau$ denote the number of trajectories required by $\mathcal{A}$ to be $(\varepsilon, \delta)$-correct. Then, $\tau$ must satisfy

$$\tau \geq \frac{A \cdot 0.6^2 c_0^2 (H - n)^2}{c_1 \cdot 64 e^8 \varepsilon^2 \alpha \cdot w(s_n)} \log\left(\frac{c_2}{\frac{c_0}{8e^4} + \delta}\right)$$

$$= \Omega\left(\frac{H^2 A}{\varepsilon^2 \alpha \cdot w(s_n)} \log\left(\frac{1}{\delta}\right)\right).$$

Recall that $n < \frac{1}{2}H$ and $0 < \alpha < \mu < \frac{1}{3}$. Thus, in the constructed instance (Figure 6), we have that $\min_{\pi, h, s: \, w_{\pi, h}(s) > 0} w_{\pi, h}(s) = w(s_n) = \mu^{n-1} > \alpha^{H-1}$, and

$$\tau = \Omega\left(\frac{H^2 A}{\varepsilon^2 \alpha \cdot \min_{\pi, h, s: \, w_{\pi, h}(s) > 0} w_{\pi, h}(s)} \log\left(\frac{1}{\delta}\right)\right).$$

Next, we consider the instance in Figure 7 as in the proof of Theorem 2, where $\alpha^{H-1} > \min_{\pi, h, s: \, w_{\pi, h}(s) > 0} w_{\pi, h}(s)$. Below we prove that on this instance any algorithm must suffer a $O(\frac{1}{\alpha^{H-1}} \cdot \frac{H^3 S^2 A}{\varepsilon^2 \alpha^2} \log(\frac{1}{\delta}))$ regret.

Define $\mathcal{E}_{s_n} := \{\hat{\pi}(s_n) = a_J\}$ as the event that the output policy $\hat{\pi}$ of algorithm $\mathcal{A}$ chooses the optimal action in state $s_n$.

Using the similar analysis as in the proof of Theorem 2 (Eq. (18)), we have

$$V_1^*(s_1) - V_1^\pi(s_1) = \frac{0.6\eta}{\alpha} \cdot (1 - \mathbb{1}\{\mathcal{E}_{s_n}\}).$$

For $\pi$ to be $\varepsilon$-optimal, we need

$$\varepsilon \geq V_1^*(s_1) - V_1^\pi(s_1) = \frac{0.6\eta}{\alpha} \cdot (1 - \mathbb{1}\{\mathcal{E}_{s_n}\}),$$

which is equivalent to

$$\mathbb{1}\{\mathcal{E}_{s_n}\} \geq 1 - \frac{\varepsilon\alpha}{0.6\eta}.$$

Let $\eta = \frac{8e^4 \varepsilon\alpha}{0.6 c_0}$ for some constant $c_0$ and small enough $\varepsilon$. Then, for $\pi$ to be $\varepsilon$-optimal, we need

$$\mathbb{1}\{\mathcal{E}_{s_n}\} \geq 1 - \frac{c_0}{8e^4}.$$

Let $\phi := 1 - \frac{c_0}{8e^4}$. For algorithm $\mathcal{A}$ to be $(\varepsilon, \delta)$-correct, we need

$$1 - \delta \leq \Pr[V^* - V^\pi \geq \varepsilon]$$

$$\leq \Pr[\mathbb{1}\{\mathcal{E}_{s_n}\} \geq \phi]$$

$$\leq \frac{\mathbb{E}[\mathcal{E}_{s_n}]}{\phi}$$

$$\leq \frac{1}{\phi} \Pr[\mathcal{E}_{s_n}],$$

which is equivalent to

$$\Pr[\bar{\mathcal{E}}_{s_n}] = 1 - \Pr[\mathcal{E}_{s_n}] \leq 1 - \phi + \phi\delta.$$

Recall that $0 < \alpha < \frac{1}{4}$. For any $j \in [A]$, $\mathrm{KL}(p_{unif}(s_n, a_j) \| p_j(s_n, a_j)) = \mathrm{KL}(\mathtt{Ber}(\alpha) \| \mathtt{Ber}(\alpha - \eta)) \leq \frac{\eta^2}{(\alpha - \eta)(1 - \alpha + \eta)} \leq \frac{c_1 \cdot \eta^2}{\alpha}$ for some constant $c_1$ and small enough $\eta$. Let $V_{s_n}$ be the number of times that algorithm $\mathcal{A}$ visited state $s_n$. To ensure $\Pr[\bar{\mathcal{E}}_{s_n}] \leq 1 - \phi + \phi\delta$, we need

$$\mathbb{E}[V_{s_n}] \geq \sum_{j=1}^{A} \frac{1}{\mathrm{KL}(p_{unif}(s_n, a_j) \| p_j(s_n, a_j))} \log\left(\frac{c_2}{1 - \phi + \phi\delta}\right)$$

$$\geq \frac{\alpha A}{c_1 \cdot \eta^2} \log\left(\frac{c_2}{1 - \phi + \phi\delta}\right)$$

$$= \frac{\alpha A \cdot 0.6^2 c_0^2}{c_1 \cdot 64 e^8 \varepsilon^2 \alpha^2} \log\left(\frac{c_2}{\frac{c_0}{8e^4} + \delta}\right)$$

for some constant $c_2$.

Let $c_0$ be a small constant such that $\frac{c_0}{8e^4} < \delta$. Let $w(s_n)$ denote the probability of visiting $s_n$ in an episode, and this probability is the same for all policies. Let $\tau$ denote the number of trajectories required by $\mathcal{A}$ to be $(\varepsilon, \delta)$-correct. Then, $\tau$ must satisfy

$$\tau \geq \frac{A \cdot 0.6^2 c_0^2}{c_1 \cdot 64 e^8 \varepsilon^2 \alpha \cdot w(s_n)} \log\left(\frac{c_2}{\frac{c_0}{8e^4} + \delta}\right)$$

$$= \Omega\left(\frac{A}{\varepsilon^2 \alpha \cdot w(s_n)} \log\left(\frac{1}{\delta}\right)\right).$$

Recall that $n = H - 1$. Thus, in the constructed instance (Figure 7), we have that $w(s_n) = \alpha^{n-1} = \alpha^{H-2}$, and

$$\tau = \Omega\left(\frac{A}{\varepsilon^2 \alpha \cdot \alpha^{H-2} \log\left(\frac{1}{\delta}\right)}\right)$$

$$= \Omega\left(\frac{A}{\varepsilon^2 \alpha^{H-1} \log\left(\frac{1}{\delta}\right)}\right).$$

$\square$

# F    PROOFS FOR WORST PATH RL

In this section, we provide the proofs of regret upper and lower bounds (Theorems 4 and 6) for Worst Path RL.

## F.1    PROOFS OF REGRET UPPER BOUND

### F.1.1    CONCENTRATION

Recall that for any $k > 0$ and $(s, a) \in \mathcal{S} \times \mathcal{A}$, $n_k(s, a)$ is the number of times that $(s, a)$ was visited before episode $k$. For any $k > 0$ and $(s', s, a) \in \mathcal{S} \times \mathcal{S} \times \mathcal{A}$, let $n_k(s', s, a)$ denote the number of times that $(s, a)$ was visited and transitioned to $s'$ before episode $k$.

For any policy $\pi$ and $(s, a) \in \mathcal{S} \times \mathcal{A}$, let $\upsilon_\pi(s, a)$ and $\upsilon_\pi(s)$ denote the probabilities that $(s, a)$ and $s$ are visited at least once in an episode under policy $\pi$, respectively.

**Lemma 18.** *It holds that*

$$\Pr\left[n_k(s,a) \geq \frac{1}{2}\sum_{k'=1}^{k-1} \upsilon_{\pi^{k'}}(s,a) - \log\left(\frac{SA}{\delta'}\right), \ \forall k > 0, \ \forall (s,a) \in \mathcal{S} \times \mathcal{A}\right] \geq 1 - \delta'$$

*Proof of Lemma 18.* For any $k$ and $(s,a) \in \mathcal{S} \times \mathcal{A}$, conditioning on the filtration generated by episodes $1, \ldots, k-1$, whether the algorithm visited $(s,a)$ at least once in episode $k$ is a Bernoulli random variable with success probability $\upsilon_{\pi^k}(s,a)$. Then, using Lemma F.4 in (Dann et al., 2017), we can obtain this lemma. $\qquad\square$

**Lemma 19.** *It holds that*

$$\Pr\left\{n_k(s',s,a) \geq \frac{1}{2} \cdot n_k(s,a) \cdot p(s'|s,a) - 2\log\left(\frac{SA}{\delta'}\right),\right.$$

$$\left. \forall k > 0, \ \forall (s',s,a) \in \mathcal{S} \times \mathcal{S} \times \mathcal{A}\right\} \geq 1 - \delta'.$$

*Proof of Lemma 19.* For any $k$, $h \in [H]$ and $(s,a) \in \mathcal{S} \times \mathcal{A}$, conditioning on the event $\{s_h^k = s, a_h^k = a\}$, the indicator $\mathbb{1}\left\{s_{h+1}^k = s'\right\}$ is a Bernoulli random variable with success probability $p(s'|s,a)$. Then, using Lemma F.4 in (Dann et al., 2017), we can obtain this lemma. $\qquad\square$

To summarize, we define some concentration events which will be used in the following proof.

$$\mathcal{G}_1 := \left\{n_k(s,a) \geq \frac{1}{2}\sum_{k'=1}^{k-1} \upsilon_{\pi^{k'}}(s,a) - \log\left(\frac{SA}{\delta'}\right), \ \forall k > 0, \ \forall (s,a) \in \mathcal{S} \times \mathcal{A}\right\}$$

$$\mathcal{G}_2 := \left\{n_k(s',s,a) \geq \frac{1}{2} \cdot n_k(s,a) \cdot p(s'|s,a) - 2\log\left(\frac{SA}{\delta'}\right), \ \forall k > 0, \ \forall (s',s,a) \in \mathcal{S} \times \mathcal{S} \times \mathcal{A}\right\}$$

$$\mathcal{G} := \mathcal{G}_1 \cap \mathcal{G}_2$$

**Lemma 20.** *Letting $\delta' = \frac{\delta}{2}$, it holds that*

$$\Pr[\mathcal{G}] \geq 1 - \delta.$$

*Proof of Lemma 20.* This lemma can be obtained by combining Lemmas 18 and 19. $\qquad\square$

### F.1.2 OVERESTIMATION AND GOOD STAGE

Recall that in Worst Path RL, for any $k > 0$, $h \in [H]$ and $(s,a) \in \mathcal{S} \times \mathcal{A}$, $Q_h^*(s,a) := r(s,a) + \min_{s' \sim p(\cdot|s,a)}(V_{h+1}^*(s'))$ and $V_h^*(s) := \max_{a \in \mathcal{A}} Q_h^*(s,a)$. In addition, $\hat{Q}_h^k(s,a) := r(s,a) + \min_{s' \sim \hat{p}^k(\cdot|s,a)}(\hat{V}_{h+1}^k(s'))$ and $\hat{V}_h^k(s) := \max_{a \in \mathcal{A}} \hat{Q}_h^k(s,a)$.

**Lemma 21** (Overestimation). *For any $k > 0$, $h \in [H]$ and $(s,a) \in \mathcal{S} \times \mathcal{A}$, $\hat{Q}_h^k(s,a) \geq Q_h^*(s,a)$ and $\hat{V}_h^k(s) \geq V_h^*(s)$.*

**Remark.** Lemma 21 shows that if the Q-value of some state-action pair is not accurately estimated, it can only be overestimated (not underestimated). This feature is due to the $\min$ metric in the Worst Path RL formulation (Eq. (2)).

*Proof of Lemma 21.* We prove this lemma by induction.

For any $k > 0$ and $s \in \mathcal{S}$, $\hat{V}_{H+1}^k(s) = V_{H+1}^*(s) = 0$.

For any $k > 0$, $h \in [H]$ and $(s,a) \in \mathcal{S} \times \mathcal{A}$, since $r(s,a)$ is known and $\hat{V}_{h+1}^k(s) \geq V_{h+1}^*(s)$ (due to the induction hypothesis), if $\hat{p}^k(\cdot|s,a)$ has detected all successor states, then $\hat{Q}_h^k(s,a) \geq Q_h^*(s,a)$. Otherwise, if $\hat{p}^k(\cdot|s,a)$ has not detected all successor states, due to the property of $\min$, we also have $\hat{Q}_h^k(s,a) \geq Q_h^*(s,a)$. Therefore, we have $\hat{V}_h^k(s) \geq V_h^*(s)$, which completes the proof. $\qquad\square$

**Lemma 22** (Non-increasing Estimated Value). *For any $k_1, k_2 > 0$ such that $k_1 \leq k_2$, $h \in [H]$ and $(s,a) \in \mathcal{S} \times \mathcal{A}$, $\hat{Q}_h^{k_1}(s,a) \geq \hat{Q}_h^{k_2}(s,a)$ and $\hat{V}_h^{k_1}(s) \geq V_h^{k_2}(s)$.*

*Proof of Lemma 22.* We prove this lemma by induction.

For any $k_1, k_2 > 0$ such that $k_1 \leq k_2$ and $s \in \mathcal{S}$, $\hat{V}_{H+1}^{k_1}(s) = V_{H+1}^{k_2}(s) = 0$.

For any $k_1, k_2 > 0$ such that $k_1 \leq k_2$, $h \in [H]$ and $(s,a) \in \mathcal{S} \times \mathcal{A}$, since $r(s,a)$ is known and $\hat{V}_{h+1}^{k_1}(s) \geq V_{h+1}^{k_2}(s)$ (due to the induction hypothesis), if $\hat{p}^{k_1}(\cdot|s,a)$ has detected all successor states, then $\hat{Q}_h^{k_1}(s,a) \geq \hat{Q}_h^{k_2}(s,a)$. Otherwise, if $\hat{p}^{k_1}(\cdot|s,a)$ has not detected all successor states, due to the $\min$ metric and that $\hat{p}^{k_2}(\cdot|s,a)$ will detect more (or the same) successor states than $\hat{p}^{k_1}(\cdot|s,a)$, we also have $\hat{Q}_h^{k_1}(s,a) \geq \hat{Q}_h^{k_2}(s,a)$. Therefore, we have $\hat{V}_h^{k_1}(s) \geq V_h^{k_2}(s)$, which completes the proof. $\square$

**Remark.** Combining Lemmas 21 and 22, we have that as the episode $k$ increases, the estimated value $\hat{Q}_h^k(s,a)$ ($\hat{V}_h^k(s)$) will decrease to its true value $Q_h^*(s,a)$ ($V_h^*(s)$) or keep the same.

Let $\mathcal{S}_* := \{s \in \mathcal{S} : v_{\pi^*}(s) > 0\}$ denote the set of states which are reachable for an optimal policy.

**Lemma 23** (Good Stage). *If there exists some episode $\bar{k} > 0$ which satisfies that for any $h \in [H]$ and $s \in \mathcal{S}_*$, $\hat{V}_h^{\bar{k}}(s) = V_h^*(s)$ and $\pi_h^{\bar{k}}(s)$ suggests an optimal action, then we have that for any $k \geq \bar{k}$, $h \in [H]$ and $s \in \mathcal{S}_*$, $\hat{V}_h^k(s) = V_h^*(s)$ and $\pi_h^k(s)$ suggests an optimal action, and thus for any $k \geq \bar{k}$, algorithm* MaxWP *takes an optimal policy.*

**Remark.** Lemma 23 reveals that if in some episode $\bar{k}$, for any step $h$ and state $s \in \mathcal{S}_*$, algorithm MaxWP estimates the V-value accurately and chooses an optimal action, then hereafter, algorithm MaxWP always takes an optimal policy.

We say algorithm MaxWP enters a *good stage*, if starting from some episode $\bar{k}$, for any $k \geq \bar{k}$, $h \in [H]$ and $s \in \mathcal{S}_*$, $\hat{V}_h^k(s) = V_h^*(s)$ and $\pi_h^k(s)$ suggests an optimal action.

*Proof of Lemma 23.* Suppose that in episode $\bar{k}$, we have that for any $h \in [H]$ and $s \in \mathcal{S}_*$, $\hat{V}_h^{\bar{k}}(s) = V_h^*(s)$ and $\pi_h^{\bar{k}}(s)$ suggests an optimal action. This is equivalent to the statement that in episode $\bar{k}$, for any $h \in [H]$ and $s \in \mathcal{S}_*$, for each optimal action $a$ (such that $Q_h^*(s,a) = V_h^*(s)$), $\hat{Q}_h^{\bar{k}}(s,a) = Q_h^*(s,a)$, and for each suboptimal action $a$ (such that $Q_h^*(s,a) < V_h^*(s)$), $Q_h^*(s,a) \leq \hat{Q}_h^{\bar{k}}(s,a) < V_h^*(s)$.

Using Lemmas 21 and 22, we have that for any $h \in [H]$ and $s \in \mathcal{S}_*$, as $k$ increases, $\hat{Q}_h^k(s,a)$ will either decrease to the true value $Q_h^*(s,a)$ or keep the same. Therefore, we have that for any $k \geq \bar{k}$, $h \in [H]$ and $s \in \mathcal{S}_*$, for each optimal action $a$, $\hat{Q}_h^{\bar{k}}(s,a) = Q_h^*(s,a)$, and for each suboptimal action $a$, $Q_h^*(s,a) \leq \hat{Q}_h^{\bar{k}}(s,a) < V_h^*(s)$, which completes the proof.

$\square$

### F.1.3 PROOF OF THEOREM 4

*Proof of Theorem 4.* Suppose that event $\mathcal{G}$ holds.

Let

$$\bar{T} := \sum_{(s,a)} \frac{1}{\min\limits_{\pi:\, v_\pi(s,a)>0} v_\pi(s,a) \cdot \min\limits_{s' \in \mathrm{supp}(p(\cdot|s,a))} p(s'|s,a)} \cdot 8\left(2\log\left(\frac{SA}{\delta}\right) + 1\right).$$

For any $(s,a) \in \mathcal{S} \times \mathcal{A}$, let

$$\bar{T}(s,a) := \frac{1}{\min\limits_{\pi:\, v_\pi(s,a)>0} v_\pi(s,a) \cdot \min\limits_{s' \in \mathrm{supp}(p(\cdot|s,a))} p(s'|s,a)} \cdot 8\left(2\log\left(\frac{SA}{\delta}\right) + 1\right).$$

It holds that $\bar{T} = \sum_{(s,a) \in \mathcal{S} \times \mathcal{A}} \bar{T}(s,a)$.

According to Lemma 23, in order to prove Theorem 4, it suffices to prove that in episode $\bar{T} + 1$, for any $h \in [H]$ and $s \in \mathcal{S}_*$, $\hat{V}_h^{\bar{T}+1}(s) = V_h^*(s)$ and $\pi_h^{\bar{T}+1}(s)$ suggests an optimal action, i.e., algorithm MaxWP has entered the good stage in episode $\bar{T} + 1$. We prove this statement by contradiction.

Suppose that in episode $\bar{T} + 1$, there exists some $h \in [H]$ and some $s \in \mathcal{S}_*$ which satisfy that $\hat{V}_h^{\bar{T}+1}(s) > V_h^*(s)$ (the value function can only be overestimated) or $\pi_h^{\bar{T}+1}(s)$ suggests a suboptimal action.

If the policy $\pi^{\bar{T}+1}$ taken in episode $\bar{T} + 1$ is optimal, then there exists some $h \in [H]$, some $s \in \mathcal{S}_*$ and some optimal action $a$ which satisfy that $v_{\pi^{\bar{T}+1}}(s) > 0$ and $\hat{Q}_h^{\bar{T}+1}(s, a) > Q_h^*(s, a)$. Otherwise, if the policy $\pi^{\bar{T}+1}$ taken in episode $\bar{T} + 1$ is suboptimal, then there exists some $h \in [H]$, some $s \in \mathcal{S}_*$ and some suboptimal action $a$ which satisfy that $v_{\pi^{\bar{T}+1}}(s) > 0$ and $\hat{Q}_h^{\bar{T}+1}(s, a) > Q_h^*(s, a)$. Hence, no matter which of the above cases happen, we have that there exists some $h \in [H]$ and some $(s, a) \in \mathcal{S} \times \mathcal{A}$ which satisfy that $v_{\pi^{\bar{T}+1}}(s, a) > 0$ and $\hat{Q}_h^{\bar{T}+1}(s, a) > Q_h^*(s, a)$.

Under the $\min$ metric in Worst Path RL, the overestimation of a Q-value comes from the following reasons: (i) Algorithm MaxWP has not detected the successor state with the lowest V-value. (ii) The V-values of successor states at the next step are overestimated.

If the overestimation of $\hat{Q}_h^{\bar{T}+1}(s, a)$ comes from the overestimation of the V-values at the next step (reason (ii)), then we have that at the next step, there exists some state-action pair whose Q-value is overestimated. Then, we can trace the overestimation from $(s, a)$ at step $h$ to some $(s', a')$ at some step $h' \geq h$, which satisfies that $\hat{Q}_{h'}^{\bar{T}+1}(s', a') > Q_{h'}^*(s', a')$ and $\hat{V}_{h'+1}^{\bar{T}+1}(x) = V_{h'+1}^*(x)$ for any $x \in \mathcal{S}$. In other words, the overestimation of $\hat{Q}_{h'}^{\bar{T}+1}(s', a')$ is purely due to that at $(s', a')$, algorithm MaxWP has not detected the successor state $x$ with the lowest value $V_{h'+1}^*(x)$.

For any $k > 0$ and $(s, a) \in \mathcal{S} \times \mathcal{A}$, let $\mathcal{T}^k(s, a) = \{k' < k : v_{\pi^{k'}}(s, a) > 0\}$ denote the set of episodes where $(s, a)$ is reachable before episode $k$. We consider the following two cases according to whether $|\mathcal{T}^{\bar{T}+1}(s', a')|$ is large enough to detect all successor states of $(s', a')$.

**Case (1):** If $|\mathcal{T}^{\bar{T}+1}(s', a')| \geq \bar{T}(s', a')$, using Lemma 18, we have

$$
\begin{aligned}
n_k(s', a') &\geq \frac{1}{2} \sum_{k'=1}^{k-1} v_{\pi^{k'}}(s', a') - \log\left(\frac{SA}{\delta'}\right) \\
&\geq \frac{1}{2} \cdot \bar{T}(s', a') \cdot \min_{\pi: v_\pi(s', a') > 0} v_\pi(s', a') - \log\left(\frac{SA}{\delta}\right) \\
&= \frac{4\left(2 \log\left(\frac{SA}{\delta}\right) + 1\right)}{\min_{s' \in \text{supp}(p(\cdot|s, a))} p(s'|s, a)} - \log\left(\frac{SA}{\delta}\right) \\
&\geq \frac{2\left(2 \log\left(\frac{SA}{\delta}\right) + 1\right)}{\min_{s' \in \text{supp}(p(\cdot|s, a))} p(s'|s, a)}
\end{aligned}
$$

Then, using Lemma 19, we have that for any $s \in \text{supp}(p(\cdot|s', a'))$,

$$
\begin{aligned}
n_k(s, s', a') &\geq \frac{1}{2} \cdot n_k(s', a') \cdot \min_{s \in \text{supp}(p(\cdot|s', a'))} p(s|s', a') - 2 \log\left(\frac{SA}{\delta}\right) \\
&\geq \frac{1}{2} \cdot 2\left(2 \log\left(\frac{SA}{\delta}\right) + 1\right) - 2 \log\left(\frac{SA}{\delta}\right) \\
&= 1
\end{aligned}
$$

which contradicts that $\hat{Q}_{h'}^k(s', a')$ is overestimated.

**Case (2):** If $|\mathcal{T}^{\bar{T}+1}(s', a')| < \bar{T}(s', a')$, we say the overestimation in episode $\bar{T} + 1$ is due to the insufficient visitation on $(s', a')$. Then, among episodes $1, \ldots, \bar{T}$, we exclude the episodes contained

in $\mathcal{T}^{\bar{T}+1}(s', a')$, i.e., we ignore the episodes where $(s', a')$ is reachable and can be the source of the overestimation. Then, the number of excluded episodes due to $(s', a')$ is $|\mathcal{T}^{\bar{T}+1}(s', a')| < \bar{T}(s', a')$.

According to Lemma 23, since episode $\bar{T}+1$ has not entered the good stage, for any $k \leq \bar{T}$, episode $k$ has also not entered the good stage. Then, for any $k \leq \bar{T}$, there exists some $h \in [H]$ and some $s \in \mathcal{S}_*$ which satisfy that $\hat{V}_h^k(s) > V_h^*(s)$ or $\pi_h^k(s)$ suggests a suboptimal action. This implies that there exists some $h \in [H]$ and some $(s, a) \in \mathcal{S} \times \mathcal{A}$ which satisfy that $v_{\pi^k}(s, a) > 0$ and $\hat{Q}_h^k(s, a) > Q_h^*(s, a)$.

Consider the last episode $k$ among episodes $1, \ldots, \bar{T}$ which is not excluded. Using the above argument, let $(s^k, a^k)$ denote the source of overestimation in episode $k$ which satisfies that $\hat{Q}_{h'}^k(s', a') > Q_{h'}^*(s', a')$ and $\hat{V}_{h'+1}^k(x) = V_{h'+1}^*(x)$ for any $x \in \mathcal{S}$. Since we have excluded all the episodes where $(s', a')$ is reachable among episodes $1, \ldots, \bar{T}$ and episode $k$ is not excluded, it holds that $(s^k, a^k) \neq (s', a')$. We repeat the above analysis on $\mathcal{T}^k(s^k, a^k)$. If Case (1) happens, i.e., $|\mathcal{T}^k(s^k, a^k)| \geq \bar{T}(s^k, a^k)$, then we can derive a contradiction and complete the proof. If Case (2) happens, i.e., $|\mathcal{T}^k(s^k, a^k)| < \bar{T}(s^k, a^k)$, we exclude episode $k$ and the episodes contained in $\mathcal{T}^k(s^k, a^k)$. Then, the number of excluded episodes due to $(s^k, a^k)$ among episodes $1, \ldots, \bar{T}$ is at most $|\mathcal{T}^k(s^k, a^k)| + 1 \leq |\bar{T}(s^k, a^k)|$.

We repeat the above procedure. Once Case (1) happens, we can derive a contradiction and complete the proof. Otherwise, if Case (2) keeps happening, we will exclude the episodes due to the reachability and possible overestimation of $(s, a)$ for all $(s, a) \in \mathcal{S} \times \mathcal{A}$, and the total number of excluded episodes is strictly smaller than $\sum_{(s,a) \in \mathcal{S} \times \mathcal{A}} \bar{T}(s, a) = \bar{T}$. Thus, there exists some episode $k_0$ among episodes $1, \ldots, \bar{T}$ which satisfies that for any $(s, a) \in \mathcal{S} \times \mathcal{A}$, $v_{\pi^{k_0}}(s, a) = 0$, which gives a contradiction. □

## F.2 REGRET LOWER BOUND

In the following, we establish a regret lower bound for Worst Path RL and give its proof.

To exclude trivial instance-specific algorithms and formally state our lower bound, we first define an $o(K)$-consistent algorithm as an algorithm which guarantees an $o(K)$ regret on any instance of Worst Path RL.

**Theorem 6.** *There exists an instance of Worst Path RL, for which the regret of any $o(K)$-consistent algorithm is at least*

$$\Omega\left(\max_{(s,a):\,\exists h,\ a \neq \pi_h^*(s)} \frac{H}{\min_{\pi:\,v_\pi(s,a)>0} v_\pi(s, a) \cdot \min_{s' \in \text{supp}(p(\cdot|s,a))} p(s'|s, a)}\right),$$

*where $\max_{(s,a):\exists h,\ a \neq \pi_h^*(s)}$ takes the maximum over all $(s, a)$ such that $a$ is sub-optimal in state $s$ at some step.*

The intuition behind this lower bound is as follows. For a critical but hard-to-reach state $s$, any $o(K)$-consistent algorithm must explore all actions $a$ in state $s$, in order to detect their induced successor states $s'$ and distinguish the optimal action. This process incurs a regret dependent on factors $\min_{\pi:v_\pi(s,a)>0} v_\pi(s, a)$ and $\min_{s' \in \text{supp}(p(\cdot|s,a))} p(s'|s, a)$, and hence the lower bound.

*Proof of Theorem 6.* Consider the instance $\mathcal{I}$ as shown in Figure 8:

The action space contains two actions, i.e., $\mathcal{A} = \{a_1, a_2\}$. The state space is $\mathcal{S} = \{s_1, s_2, \ldots, s_n, x_1, x_2, x_3\}$, where $n = S - 3$ and $s_1$ is the initial state. Let $H \gg S$ and $0 < \alpha < \frac{1}{4}$.

The reward functions are as follows. For any $a \in \mathcal{A}$, $r(x_1, a) = 1$, $r(x_2, a) = 0.8$ and $r(x_3, a) = 0.2$. For any $i \in [n]$ and $a \in \mathcal{A}$, $r(s_i, a) = 0$.

The transition distributions are as follows. For any $a \in \mathcal{A}$, $p(s_2|s_1, a) = \alpha$, $p(x_1|s_1, a) = 1 - 3\alpha$, $p(x_2|s_1, a) = \alpha$ and $p(x_3|s_1, a) = \alpha$. For any $i \in \{2, \ldots, n-1\}$ and $a \in \mathcal{A}$, $p(s_{i+1}|s_i, a) = \alpha$ and $p(x_1|s_i, a) = 1 - \alpha$. $x_1$, $x_2$ and $x_3$ are absorbing states, i.e., for any $a \in \mathcal{A}$, $p(x_1|x_1, a) = 1$, $p(x_2|x_2, a) = 1$ and $p(x_3|x_3, a) = 1$. The state $s_n$ is a bandit state, which has an optimal action and a suboptimal action. Let $a_*$ denote the optimal action in state $s_n$, which is uniformly drawn from

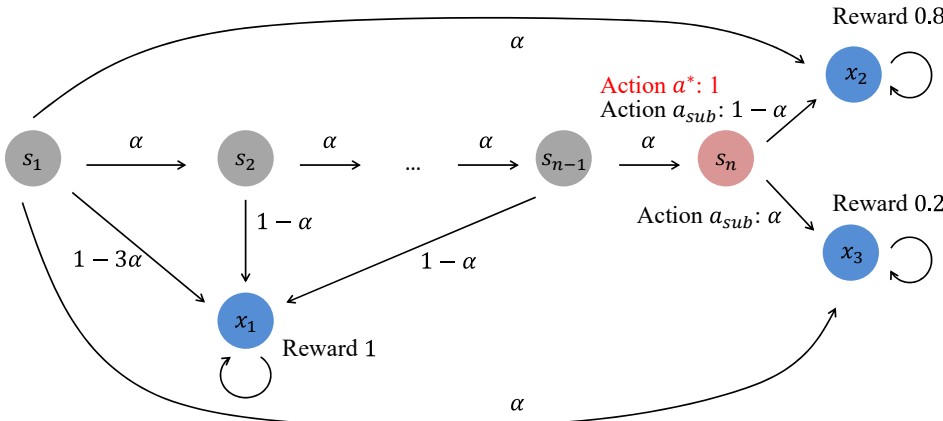

Figure 8: The instance for lower bound under the $\min$ metric (Theorem 6).

$\{a_1, a_2\}$, and let $a_{sub}$ denote the other sub-optimal action in state $s_n$. For the optimal action $a_*$, $p(x_2|s_n, a_*) = 1$. For the sub-optimal action $a_{sub}$, $p(x_2|s_n, a_{sub}) = 1 - \alpha$ and $p(x_3|s_n, a_{sub}) = \alpha$.

Fix an $o(K)$-consistent algorithm $\mathcal{A}$, which guarantees a sub-linear regret on any instance of Worst Path RL. We have that $\mathcal{A}$ needs to observe the transition from $(s_n, a_{sub})$ to $x_3$ at least once. Otherwise, without any observation of the transition from $(s_n, a_{sub})$ to $x_3$, $\mathcal{A}$ can only trivially choose $a_1$ or $a_2$ in state $s_n$, and no matter $\mathcal{A}$ chooses $a_1$ or $a_2$, it will suffer a linear regret in the counter instance where the unchosen action is optimal.

Thus, any $o(K)$-consistent algorithm must observe the transition from $(s_n, a_{sub})$ to $x_3$ at least once, and needs at least

$$\frac{1}{v_{\pi_{sub}}(s_n, a_{sub}) \cdot p(x_3|s_n, a_{sub})}$$

episodes with sub-optimal policies. Here $\pi_{sub}$ denotes a policy which chooses $a_{sub}$ in state $s_n$, and $v_{\pi_{sub}}(s_n, a_{sub})$ denotes the probability that $(s_n, a_{sub})$ is visited at least once in an episode under policy $\pi_{sub}$.

Once the agent takes a sub-optimal policy in an episode, she will suffer regret $0.6(H - n)$ in this episode.

Therefore, $\mathcal{A}$ needs to suffer at least

$$\Omega\left(\frac{1}{v_{\pi_{sub}}(s_n, a_{sub}) \cdot p(x_3|s_n, a_{sub})} \cdot 0.6(H - n)\right)$$

regret in expectation.

Since in the constructed instance (Figure 8)

$$\max_{(s,a):\, \exists h,\, a \neq \pi_h^*(s)} \frac{1}{\min\limits_{\pi:\, v_\pi(s,a)>0} v_\pi(s, a) \cdot \min\limits_{s' \in \mathrm{supp}(p(\cdot|s,a))} p(s'|s, a)} = \frac{1}{v(s_n, a_{sub}) \cdot p(x_3|s_n, a_{sub})},$$

we have that $\mathcal{A}$ needs to suffer at least

$$\Omega\left(\max_{(s,a):\, \exists h,\, a \neq \pi_h^*(s)} \frac{H}{\min\limits_{\pi:\, v_\pi(s,a)>0} v_\pi(s, a) \cdot \min\limits_{s' \in \mathrm{supp}(p(\cdot|s,a))} p(s'|s, a)}\right)$$

regret. $\qquad\square$

## G   TECHNICAL TOOL

In this section, we present a useful technical tool.

**Lemma 24** (Lemma 13 in (Ménard et al., 2021)). *Let $A, B, C, D, E$ and $\beta$ be positive scalars such that $1 \leq B \leq E$ and $\beta \geq e$. If $T \geq 0$ satisfies*

$$T \leq C \sqrt{T \left( A \log \left( \beta T \right) + B \log^2 \left( \beta T \right) \right)} + D \left( A \log \left( \beta T \right) + E \log^2 \left( \beta T \right) \right),$$

*then we have*

$$T \leq C^2 \left( A + B \right) C_1^2 + \left( D + 2 \sqrt{D} C \right) \left( A + E \right) C_1^2 + 1,$$

*where*

$$C_1 = \frac{8}{5} \log \left( 11 \beta^2 \left( A + E \right) \left( C + D \right) \right).$$

