# OpenReview forum: "Provably Efficient Risk-Sensitive Reinforcement Learning: Iterated CVaR and Worst Path"
_ICLR.cc/2023/Conference — ICLR 2023 poster_

### Official Review · Reviewer_rNfK · 2022-10-24

**Confidence:** 3
**Correctness:** 4
**Technical Novelty And Significance:** 3
**Empirical Novelty And Significance:** Not applicable
**Recommendation:** 8

**Clarity, Quality, Novelty And Reproducibility:**

- Clarity. The paper is well-written and easy to follow.
- Quality. The claims made in the paper are correct and justified.
- Novelty. Please see the previous paragraph on novelty.
- Reproducible. The proofs and simulation results are reproducible.

**Strength And Weaknesses:**

Strength
- The paper studies a setting that is novel and intriguing, offering new insights that are crucial to wider adoption of RL.
- The results obtained in the paper are comprehensive, covering both the best policy identification setting and the online learning setting.
- The regret decomposition used to obtain the main results is novel and not found in existing literature.

Weakness
- Algorithmically the work is not sufficiently novel. The two proposed algorithms bear strong resemblance to their non-risk aware counterparts. Moreover, while the concentration analysis leverages novel decomposition techniques, the lemmas used to prove the results (e.g. Lemma D.1, which shows the concentration behavior of the empirical CVaR estimate) can be found in existing literature. Overall the proof structure still seems to largely follow the optimism principle in provably efficient reinforcement learning. It would be great if the authors could further elaborate on how their analysis is not a combination of existing techniques in RL and existing concentration results from CVaR literature.

------ Post response update --------

I have read the authors' response and my concerns are addressed.

**Summary Of The Paper:**

The work studies iterated CVaR RL, and its limiting case worst path RL, as a representative provably efficient risk-sensitive RL problem. Two metrics, namely regret and best policy identification, are proposed and provably efficient algorithms with nearly matching upperbounds are provided.

**Summary Of The Review:**

The paper studies an interesting problem, risk-sensitive RL, and provides comprehensive results. The analyses used to obtain these results feature novel techniques. While some aspects of the proof depend on existing work, to me it does not outweigh the contributions made by the paper.

---

> ### Author Response · Authors · 2022-11-15
> **Response to Reviewer rNfK - Part 1/2**
>
> Thank you very much for your time and efforts in reviewing our paper! Following your suggestions, we have incorporated more discussion on the novelty of algorithms and analysis in our revision.
>
> **1. Novelty in Algorithm Design.**
>
> **(1).**
> Our best arm identification (BPI) algorithm ICVaR-BPI introduces a novel distorted (conditional) empirical transition probability to construct estimation error, which effectively assigns more attention to bad situations and well fits the main focus of the Iterated CVaR criterion.
>
> Different from existing BPI algorithms which typically use empirical transition probability $\hat{p}^k(\cdot|s,a)$ to construct estimation error, ICVaR-BPI establishes the estimation error by a conditional empirical transition probability $\hat{\beta}^{k;\alpha,\underline{V}^{k}_{h+1}}(\cdot|s,a)$. Equipped with this transition measure, ICVaR-BPI delicately places more emphasis on the estimated bad successor states, and adaptively fits the Iterated CVaR criterion.
>
>
> **(2).**
> Our Worst Path RL algorithm MaxWP establishes a simple and efficient empirical Q-value function and achieves a constant regret (independent of the number of episodes $K$), which is novel in the RL literature.
>
> Unlike classic RL algorithms which usually use optimistic exploration bonuses, MaxWP constructs an efficient empirical Q-value function, which makes full use of the unique feature of Worst Path RL and ensures that the Q-value function can only be over-estimated.
> This algorithm design combines the exploration of over-estimated actions (which lead to undetected successor states) and the exploitation of current best actions, and further enables a constant regret.
>
>
> **2. Novelty in Analysis.**
>
>
>
> **(1).**
> We develop a new analytical technique to bound the change of CVaR when the true value function shifts to an optimistic value function, which is new to the CVaR concentration literature and can find applications in other CVaR-based RL problems.
>
> In classic RL analysis, the change of the expected reward when the true value function shifts to an optimistic value function can be easily analyzed via transition probabilities. In contrast, under Iterated CVaR RL, the relationship between the change of CVaR and the shift of value functions is not straightforward. To handle this challenge, we develop a new technique to bound the change of CVaR by employing conditional transition probabilities (Eq. (3) in Section 4.1, Lemma 11 in Appendix D.1.2). This technique enables us to further conduct a fine-grained regret decomposition for Iterated CVaR RL (Eq. (4) in Section 4.1, Eq. (13) in Appendix D.1.3).
>
>
> **(2).**
> We establish a novel regret decomposition for Iterated CVaR RL via a distorted (conditional) visitation distribution, and quantify the deviation between this conditional visitation distribution and the original visitation distribution.
>
> In Iterated CVaR RL, the contribution of a state-action pair to the regret is not proportional to its visitation probability as in classic RL. Thus, under Iterated CVaR RL, the regret cannot be decomposed into the estimation error of state-action pairs via visitation distribution, which imposes huge challenges to the regret analysis.
>
> To overcome this difficulty, we decompose the regret via a carefully defined conditional visitation distribution (Eq. (4) in Section 4.1, Eq. (13) in Appendix D.1.3). Furthermore, we bound the distance between this conditional visitation distribution and the original visitation distribution by an MDP-intrinsic measure and a risk-level dependent factor, which is critical to derive a tight regret bound (Lemma 9 in Appendix D.1.2, Appendix D.1.3).
>
> **(3).**
> We take advantage of the unique problem feature of Worst Path RL and derive a tight constant regret analysis, which is novel to the RL literature.
>
> Most of existing RL analysis is based on the idea of upper confidence bounds, which induce a $O(\sqrt{K})$ regret. In contrast, under Worst Path RL, we make full use of the problem feature that, Worst Path RL only cares the successor state with the minimum value. Utilizing this feature, we prove several important properties, e.g., the Q-value function can only be over-estimated but not under-estimated (Lemma 21 in Appendix F.1.2), the empirical Q-value function is non-increasing as the episodes proceed (Lemma 22 in Appendix F.1.2), and algorithm MaxWP will not choose a suboptimal action once it has an accurate estimation of state connectivity (Lemma 23 in Appendix F.1.2). Building upon these properties, we adopt a contradiction argument to derive a constant regret which is independent of $K$ (Appendix F.1.3).
>
> We have included more discussion on the novelty of algorithms and analysis in Sections 4.1, 5.1 and 6.1 of our revision.

---

> > ### Author Response · Authors · 2022-11-15
> > **Response to Reviewer rNfK - Part 2/2**
> >
> > Finally, we want to thank the reviewer again for the insightful comments for our work!
> > If our response resolves your concerns at a satisfactory level, we hope that the reviewer can kindly raise the rating of our work. Certainly, we are more than happy to address any further questions you may have during the discussion period.

---

> > > ### Author Response · Authors · 2022-11-28
> > > **Dear Reviewer rNfK, We Are Wondering If Your Concerns Are Addressed**
> > >
> > > Dear Reviewer rNfK,
> > >
> > > We are wondering if you have gotten a chance to look over our responses and revisions and if your concerns are resolved.
> > >
> > > If our response resolves your concerns at a satisfactory level, we would like to ask you to kindly consider raising the score for our work. In addition, we are more than happy to answer any further questions you may have during the discussion period.
> > >
> > > Thank you very much!
> > >
> > > Best regards,
> > > Authors

---

### Official Review · Reviewer_U4MP · 2022-10-27

**Confidence:** 3
**Correctness:** 3
**Technical Novelty And Significance:** 3
**Empirical Novelty And Significance:** 2
**Recommendation:** 6

**Clarity, Quality, Novelty And Reproducibility:**

Clarity: The main paper is clear overall. It is not that easy to follow, but it's due to the nature of such a dense technical discussion. I think some discussions like comparison with CVaR MDP and Fei et al., should be mentioned in more detail in the main paper rather than the appendix, and some proof sketches can be deferred on the other hand.

Quality: The technical results (upper bounds and lower bounds) are solid.

Novelty: It is the first work to introduce iterated CVaR to RL settings. The main techniques it used in the analysis follow the method in Zanette & Brunskill, 2019, and some adjustments to deal with the new challenges with iterated CVaR.

Reproducibility: Proof and code are included.


**Strength And Weaknesses:**

Strength:
1. This paper proposed a new risk-sensitive RL formulation: iterated CVaR RL. Though the idea of iterated CVaR is not new, it is the first use case in RL to my knowledge. This paper gives a novel analysis and an alternative to standard CVaR MDP formulation.
2. The paper thoroughly discusses the theoretical results of iterated CVaR RL under different settings: risk minimization, the sample complexity of best policy identification, limitation case when \alpha goes to zero, and a lower bound for each case.

Weakness:
1. I think it could be better to have more discussion on the pros and cons of iterated CVaR RL with CVaR of return. With the current description in the main paper and comparison in Appendix C, it is not very convincing the iterated CVaR is more suitable for risk-sensitive RL than CVaR MDP.
 - Iterated CVaR takes the alpha CVaR over each transition step. That means in an H-step problem, only the worst $\alpha^H$ portion of the outcome matter to this quantity. As it decreases exponentially, it's less convincing to me that such a quantity is a good metric to distinguish different algorithms. For a reasonable $H$, $1-\alpha^H$ portion could be more meaningful even for the risk-sensitive analysis since it could consider more catastrophic returns.
 - Based on the last point, it is more concerning that an action with better performance in the worst $\alpha^H$ portion could actually have worse performance in the rest (not only for overall performance but even under risk-sensitive metrics). An example could be constructed by considering different CDF curves of the full return.
2. The term $1/\min_{\pi,s, a} w(s, a)$ in Theorem 1 seems meaningless to me. It can be arbitrarily large in an MDP, as long as there exists one state such that it has zero probability to be visited by at least one action. This looks like a very weak condition so please correct me if I'm wrong. Otherwise, the bound only depends on $\alpha^H$ in almost all MDPs. The lower bound construction verified this, as the $min_{\pi,s, a} w(s, a) = \Theta(\alpha^H) $. If authors want to show that there is a real dependency on $min_{\pi,s, a} w(s, a)$, it needs to give a construction where $w$ terms can be changed freely without changing $\alpha$ or $H$. (Like the proof of regret lower bound on $S$ and $H$.)


Some minor writing issues and related work:
 - I didn't find a formal definition of $w_{\pi, h}(s, a)$ in Theorem 1.
 - There exists work on efficient exploration with the CVaR of return (as CVaR MDP) in unknown MDP. E.g. Keramati et al., 2020, Being Optimistic to Be Conservative: Quickly Learning a CVaR Policy.


**Summary Of The Paper:**

This paper proposed a new risk-sensitive reinforcement learning formulation, namely iterated CVaR RL. It defines the value as the CVaR of the next step's value function with respect to the randomness in the states' transition. Under this framework, this paper proposes a regret minimization algorithm and a best policy identification algorithm, with regret and sample complexity upper bound respectively. The author also gives a lower bound of regret as well. Finally, this paper also studies the case of CVaR as the risk level $\alpha$ goes to zero, namely the worst path case, and gives a regret analysis.

**Summary Of The Review:**

This paper gives a new formulation of risk-sensitive RL, iterated CVaR. It's a different definition than most current work, which takes CVaR over the return variable. The analysis in this paper under iterated CVaR is thorough. The main concerns are about the significance of some technical results, and the comparison with the current CVaR MDP.

---

> ### Author Response · Authors · 2022-11-15
> **Response to Reviewer U4MP - Part 1/5**
>
> **Overall Reply.**
>
> Thank you very much for your time and efforts in reviewing our paper!
> We believe that your technical concerns can be easily addressed by noticing that $\min_{\pi,h,(s,a):\ w_{\pi,h}(s,a)>0} w_{\pi,h}(s,a)$ only takes the minimum over reachable state-action pairs, and our new lower bound construction. In addition, we believe that the concern on motivation can also be effectively addressed by noting that Iterated CVaR is not equivalent to the worst $\alpha^H$-portion, and our included discussions and comparisons with current CVaR MDP in the revision. We will elaborate on these points in detail below.
>
>
> **1. Motivation of Iterated CVaR RL.**
>
> Iterated CVaR RL concerns the worst $\alpha$-portion situations *at each step*, and aims to maximize the worst $\alpha$-portion tail of the reward-to-go at each step.
> Iterated CVaR RL is most suitable for  safety-critical applications where there is a fatal failure probability that leads to catastrophic states at each decision stage. Our goal is to find a policy that guarantees  safety even when failures (bad situations) can happen at each stage.
>
> For example, consider an unmanned helicopter control task, e.g., [Johnson \& Kannan, 2002], where one flies an unmanned helicopter to complete some task. There is a small probability that, at each time during execution, the helicopter encounters a sensing or control failure and does not take the scheduled action.
> In order to guarantee the safety of the helicopter and surrounding workers, it is important to guarantee that even if the sensing or control failure occurs, the taken policy ensures that the helicopter does not crash and cause fatal damage.
>
> As another example, consider a clinical treatment planning scenario, e.g., [Wang et al., 2019], where there is a treatment failure probability (e.g., surgical failures) at each therapy phase, and we want to guarantee the safety of patients, taking into consideration that treatment  can fail at each phase.
> In this case, Iterated CVaR RL allows us to pursue a safe and effective policy against the risk of treatment failures and prevents us from getting into catastrophic states.
>
> CVaR MDP aims to maximize the worst $\alpha$-portion of the *total* reward.
> CVaR MDP takes more cumulative reward into account, and prefers actions which have better performance in general, but can have larger probabilities of getting into catastrophic states. Thus, CVaR MDP is more suitable for scenarios where bad situations lead to a higher cost instead of fatal damage, e.g., finance.
> In contrast, Iterated CVaR RL prefers actions which **have smaller probabilities of getting into catastrophic states**.
> Hence, Iterated CVaR RL is suitable for the safety-critical applications, where **catastrophic states are unacceptable and need to be carefully avoided**, e.g., clinical treatment planning.
>
> We emphasize that Iterated CVaR is **not equivalent to** simply taking the worst $\alpha^H$-portion of the total reward. In fact, the good $(1-\alpha^H)$-portion of the total reward also contributes to Iterated CVaR. This is because Iterated CVaR considers bad situations for all states (both good and bad states) in its iterated computation, instead of just considering bad situations upon bad states.
>
> Below we provide an illustrating example of clinical treatment planning. Here we interpret the objective as cost minimization for ease of understanding, and set the risk level $\alpha=0.05$.

---

> > ### Author Response · Authors · 2022-11-15
> > **Response to Reviewer U4MP - Part 2/5**
> >
> > **Table 1. Illustrating example for the comparison between Iterated CVaR RL and the current CVaR MDP.**
> >
> > | layer 1  |   |   |   |   |  $s_1$ |   |   |   | |
> > |:------------:|:------------:|:------------:|:------------:| :------------: | :------------:| :------------:|:------------:| :------------: |:------------:|
> > |   |   |   |  $\downarrow$ ($a_1$) |   |  |   |  $\downarrow$ ($a_2$) |   |  |
> > | layer 2  |   |  |  $s_2$  |   |   |   |  $s_3$  |   | |
> > |   |   |  $\downarrow$ (0.05)  |   |  $\downarrow$ (0.95)  |   |  $\downarrow$ (0.01)  |   | $\downarrow$ (0.99)   | |
> > | layer 3  |   |  $s_4$ |   |  $s_5$ | | $s_6$  |  |  $s_7$  |  |
> > |    | $\downarrow$ (0.05)  |  $\downarrow$ (0.95)  |  $\downarrow$ (0.05)  |  $\downarrow$ (0.95)  |  |  $\downarrow$ (0.01)  | $\downarrow$  (0.99)   | $\downarrow$  (0.01)  |  $\downarrow$ (0.99)  |
> > |  layer 4 | $s_8$  |  $s_9$  |  $s_{10}$ |  $s_{11}$ | | $s_{12}$  |  $s_{13}$ | $s_{14}$  | $s_{15}$  |
> > | cost: | 1  |  0.4  |  0.4 |  0 | | 1  |  0.5 | 0.5  | 0  |
> >
> > Consider a 4-layered binary tree-structured MDP as shown in Table 1. The state sets in layers 1, 2, 3 and 4 are $\\{s_1\\}$, $\\{s_2,s_3\\}$, $\\{s_4,\dots,s_7\\}$ and $\\{s_8,\dots,s_{15}\\}$,  respectively. There are two actions $a_1, a_2$ in each state, and $a_1, a_2$ have the same transition distribution in all states except the initial state $s_1$. Thus, a policy is to decide whether to choose $a_1$ or $a_2$ in state $s_1$, which leads to different subsequent costs.
> >
> >
> > The agent starts from the initial state $s_1$ in layer 1.
> > If the agent takes action $a_1$, she will transition to state $s_2$ deterministically, and goes into the left sub-tree. On the other hand, if the agent takes action $a_2$ in state $s_1$, she will transition to state $s_3$ deterministically, and enters the right sub-tree.
> >
> > If the agent goes into the left sub-tree (state $s_2$) in layer 2, she will transition to $s_4$ and $s_5$ in layer 3 with probabilities $0.05$ and $0.95$, respectively.
> > Then, if she starts from state $s_4$ in layer 3, she will transition to $s_8$ and $s_9$ in layer 4 with probabilities $0.05$ and $0.95$, respectively.
> > Otherwise, if she starts from state $s_5$ in layer 3, she will transition to $s_{10}$ and $s_{11}$ in layer 4 with probabilities $0.05$ and $0.95$, respectively.
> >
> > On the other hand, if the agent goes into the right sub-tree (state $s_3$) in layer 2, she will transition to $s_6$ and $s_7$ in layer 3 with probabilities $0.01$ and $0.99$, respectively.
> > Then, if she starts from state $s_6$ in layer 3, she will transition to $s_{12}$ and $s_{13}$ in layer 4 with probabilities $0.01$ and $0.99$, respectively.
> > Otherwise, if she starts from state $s_7$ in layer 3, she will transition to $s_{14}$ and $s_{15}$ in layer 4 with probabilities $0.01$ and $0.99$, respectively.
> >
> >
> > The costs are state-dependent, and only the states in layer 4 produce non-zero costs. To be concrete, we use the clinical trial example, and the costs represent the patient status. Specifically, in layer 4, $s_8$ and $s_{12}$ give costs 1, which denote **death**. $s_{13}$ and $s_{14}$ produce costs 0.5, which mean that the patient **gets better**. $s_9$ and $s_{10}$ induce costs 0.4, which denote that the patient **gets much better**. $s_{11}$ and $s_{15}$ produce costs 0, which stand for that the patient is **fully cured**.
> >
> >
> >
> > Under the CVaR criterion, we have that
> > $$
> > Q^{CVaR, \alpha}(s_1,a_1)=\frac{0.0025}{0.05} \cdot 1+\frac{0.05-0.0025}{0.05} \cdot 0.4=0.43
> > $$
> > and
> > $$
> > Q^{CVaR, \alpha}(s_1,a_2)=\frac{0.0001}{0.05} \cdot 1+\frac{0.05-0.0001}{0.05} \cdot 0.5=0.501
> > $$
> >
> > Thus, CVaR MDP will choose action $a_1$ (and goes into the left sub-tree), since $a_1$ leads to better medium states $s_9$ and $s_{10}$, which give a lower cost $0.4$ than the cost $0.5$ produced by the right sub-tree.
> >
> > On the other hand, under the Iterated CVaR criterion, we have that
> > $$
> > Q^{ICVaR, \alpha}(s_1,a_1)=\frac{0.05}{0.05} \cdot Q^{ICVaR, \alpha}(s_4,\cdot)=\frac{0.05}{0.05} \cdot \left( \frac{0.05}{0.05} \cdot Q^{ICVaR, \alpha}(s_8,\cdot) \right)=\frac{0.05}{0.05} \cdot \left( \frac{0.05}{0.05} \cdot 1 \right)=1
> > $$
> > and
> > $$
> > Q^{ICVaR, \alpha}(s_1,a_2) = \frac{0.01}{0.05} \cdot Q^{ICVaR, \alpha}(s_6,\cdot) + \frac{0.05-0.01}{0.05} \cdot Q^{ICVaR, \alpha}(s_7,\cdot)
> > $$
> > $$
> > = \frac{0.01}{0.05} \cdot \left( \frac{0.01}{0.05} \cdot Q^{ICVaR, \alpha}(s_{12},\cdot) + \frac{0.05-0.01}{0.05} \cdot Q^{ICVaR, \alpha}(s_{13},\cdot) \right) + \frac{0.05-0.01}{0.05} \cdot \left( \frac{0.01}{0.05} \cdot Q^{ICVaR, \alpha}(s_{14},\cdot) + \frac{0.05-0.01}{0.05} \cdot Q^{ICVaR, \alpha}(s_{15},\cdot) \right)
> > $$
> > $$
> > = \frac{0.01}{0.05} \cdot \left( \frac{0.01}{0.05} \cdot 1 + \frac{0.05-0.01}{0.05} \cdot 0.5 \right) + \frac{0.05-0.01}{0.05} \cdot \left( \frac{0.01}{0.05} \cdot 0.5 + \frac{0.05-0.01}{0.05} \cdot 0 \right)
> > $$
> > $$
> > = 0.2
> > $$

---

> > > ### Author Response · Authors · 2022-11-15
> > > **Response to Reviewer U4MP - Part 3/5**
> > >
> > > Thus, Iterated CVaR RL will instead choose action $a_2$, because $a_2$ has a smaller probability of going into the bad left direction (which leads to the catastrophic state $s_{12}$).
> > >
> > >
> > > The above example shows that, Iterated CVaR RL prefers actions with a smaller probability of getting into catastrophic states. In contrast, CVaR MDP favors actions with  better average therapeutic effects, but has a larger probability of causing death.
> > >
> > >
> > > Note that the above example also demonstrates that Iterated CVaR is **not equivalent to** the worst $\alpha^{H}$-portion of the total cost. To see this, we have that (here we consider $\alpha^{3}$ because  there are $3$ transition steps):
> > >
> > > $$
> > > Q^{CVaR, \alpha^{3}}_1(s_1,a_2)=\frac{0.0001}{0.000125} \cdot 1+\frac{0.000125-0.0001}{0.000125} \cdot 0.5=0.9
> > > $$
> > >
> > > and
> > >
> > > $$
> > > Q^{ICVaR, \alpha}(s_1,a_2) =  \frac{0.01}{0.05} \cdot \left( \frac{0.01}{0.05} \cdot 1 + \frac{0.05-0.01}{0.05} \cdot 0.5 \right) + \frac{0.05-0.01}{0.05} \cdot \left( \frac{0.01}{0.05} \cdot 0.5 + \frac{0.05-0.01}{0.05} \cdot 0 \right)
> > > $$
> > >
> > > $$
> > > = 0.2
> > > $$
> > >
> > > In addition, one can see that, the good state which gives cost 0 (i.e., $s_{15}$) also contributes to $Q^{ICVaR, \alpha}(s_1,a_2)$, which shows that the good $(1-\alpha^H)$-portion of the total cost also matters for Iterated CVaR.
> > >
> > >
> > > In fact, the Iterated CVaR criterion has been **widely considered** in the risk-sensitive MDP literature (which studies the known-transition setting), e.g., [Hardy \& Wirch, 2004; Ruszczyński, 2010; Chu \& Zhang, 2014; Bauerle \& Glauner, 2022].
> > > Our work expands the theory of Iterated CVaR MDP from the known-transition setting to more practical unknown-transition (RL) model.
> > >
> > > We have included these discussions and comparisons in Sections 1, 2 and Appendix C.2 of our revision.
> > >
> > >
> > >
> > > ---
> > >
> > > References:
> > > Eric Johnson, Suresh Kannan. Adaptive flight control for an autonomous unmanned helicopter.
> > > AIAA Guidance, Navigation, and Control Conference and Exhibit, 2002.
> > > Chunhao Wang, Xiaofeng Zhu, Julian C Hong, Dandan Zheng. Artificial intelligence in radiotherapy treatment planning: present and future. Technology in Cancer Research \& Treatment, 2019.
> > > Mary R Hardy, Julia L Wirch. The iterated CTE: a dynamic risk measure. North American
> > > Actuarial Journal, 2004.
> > > Andrzej Ruszczyński. Risk-averse dynamic programming for Markov decision processes. Mathematical Programming, 2010.
> > > Shanyun Chu, Yi Zhang. Markov decision processes with iterated coherent risk measures. International Journal of Control, 2014.
> > > Nicole Bauerle, Alexander Glauner. Markov decision processes with recursive risk measures.
> > > European Journal of Operational Research, 2022.
> > >
> > > **2. Significance of Technical Results.**
> > >
> > >
> > > **2-1. $\min_{\pi,h,s:\  w_{\pi,h}(s)>0} w_{\pi,h}(s)$ in Theorem 1.**
> > >
> > >
> > > $\min_{\pi,h,(s,a):\ w_{\pi,h}(s,a)>0} w_{\pi,h}(s,a)$ or $\min_{\pi,h,s:\  w_{\pi,h}(s)>0} w_{\pi,h}(s)$ only takes the minimum over the state-actions or states which have **positive** probabilities to be visited under policy $\pi$.
> > > Thus, $\min_{\pi,h,(s,a):\ w_{\pi,h}(s,a)>0} w_{\pi,h}(s,a)$ or $\min_{\pi,h,s:\  w_{\pi,h}(s)>0} w_{\pi,h}(s)$ will **never be zero**, and the factor $\min_{\pi,h,s:\  w_{\pi,h}(s)>0} w_{\pi,h}(s)$ in Theorem 1 can dominate the bound (i.e., it is non-vacuous).
> > > This point has been highlighted in Remark 1 of our original submission.
> > >
> > > Here for any policy $\pi$, $h \in [H]$, $(s,a) \in \mathcal{S} \times \mathcal{A}$ and $s \in \mathcal{S}$, $w_{\pi,h}(s,a)$ and $w_{\pi,h}(s)$ denote the probabilities of visiting $(s,a)$ and $s$ at step $h$ under policy $\pi$, respectively.
> > >
> > >
> > > **2-2. Constructions of Lower Bound (Theorem 2).**
> > >
> > > **(1). Original Lower Bound Construction.**
> > >
> > >
> > > In our original lower bound construction (Figure 1), $\min_{\pi,h,s:\  w_{\pi,h}(s)>0} w_{\pi,h}(s)=\alpha^{n-1}$, where $n < H$. Thus, $\min_{\pi,h,s:\  w_{\pi,h}(s)>0} w_{\pi,h}(s)$ **is larger than** $\Theta(\alpha^H)$, and the lower bound (Theorem 2) does depend on $\min_{\pi,h,s:\  w_{\pi,h}(s)>0} w_{\pi,h}(s)$.

---

> > > > ### Author Response · Authors · 2022-11-15
> > > > **Response to Reviewer U4MP - Part 4/5**
> > > >
> > > > **(2). New Lower Bound Construction.**
> > > >
> > > > Following the reviewer's suggestion, below we  provide a new lower bound construction, where $\min_{\pi,h,s:\  w_{\pi,h}(s)>0} w_{\pi,h}(s)$ does not depend on $\alpha$ and $H$.
> > > >
> > > >
> > > > Specifically, in Figure 1, we change the transition probabilities of $s_1 \rightarrow s_2$, $s_2 \rightarrow s_3$, ..., $s_{n-1} \rightarrow s_n$, $s_1 \rightarrow x_2$, $s_1 \rightarrow x_3$ from $\alpha$ to $\mu$, change the transition probabilities of  $s_2 \rightarrow x_1$, $s_3 \rightarrow x_1$, ..., $s_{n-1} \rightarrow x_1$ from $1-\alpha$ to $1-\mu$, and change the transition probability of $s_1 \rightarrow x_1$ from $1-3\alpha$ to $1-3\mu$. Here $n<\frac{1}{2}H$, and $\mu$ is an *arbitrary* parameter which satisfies $0<\alpha<\mu<\frac{1}{3}$. Under this new construction, Theorem 2 still holds.
> > > >
> > > > In this new instance, $\min_{\pi,h,s:\  w_{\pi,h}(s)>0} w_{\pi,h}(s)=\mu^{n-1}$, which does not depend on $\alpha$ and $H$, and is larger than $\Theta(\alpha^H)$. Thus, the lower bound does depend on $\min_{\pi,h,s:\  w_{\pi,h}(s)>0} w_{\pi,h}(s)$, which demonstrates that the factor $\min_{\pi,h,s:\  w_{\pi,h}(s)>0} w_{\pi,h}(s)$ is inevitable for Iterated CVaR RL in general.
> > > >
> > > >
> > > > The proof idea is as follows.
> > > > When $\mu>\alpha$, the path $s_1 \rightarrow s_2 \rightarrow ... \rightarrow s_{n-1} \rightarrow s_n \rightarrow x_2/x_3$ gives the worst $\alpha$-portion of the reward-to-go at each step. Thus, the value function primarily depends on this path. In addition, $s_n$ is the state with the minimum probability to be visited, and is a bandit state (i.e., has an optimal action and suboptimal actions). Therefore, in order to learn the optimal policy, any algorithm needs to sufficiently visit state $s_n$ to identify the optimal action, and the algorithm will pay a regret due to collecting information from $s_n$.
> > > > Therefore, the regret lower bound will depend on the probability of visiting $s_n$, which is equal to $\min_{\pi,h,s:\  w_{\pi,h}(s)>0} w_{\pi,h}(s)$.
> > > >
> > > > In this new construction, for fixed $\alpha$ and $H$, $\min_{\pi,h,s:\  w_{\pi,h}(s)>0} w_{\pi,h}(s)=\mu^{n-1}$ can be freely changed under the conditions $n<\frac{1}{2}H$ and $\alpha<\mu<\frac{1}{3}$. Note that the condition $\alpha<\mu<\frac{1}{3}$ is not limited here. In fact, it is common in the risk-sensitive online learning and RL literature that the parameters in lower bound constructions have problem-dependent range limits, see e.g.,  [Agrawal et al., 2021; Fei et al., 2020; Fei \& Xu, 2022].
> > > >
> > > >
> > > > We have incorporated this new lower bound construction and its complete proof in Section 4.2 and Appendix D.2 of our revision.
> > > >
> > > > ---
> > > >
> > > > References:
> > > > Shubhada Agrawal, Wouter M. Koolen, Sandeep Juneja. Optimal best-arm identification methods for tail-risk measures. NeurIPS, 2021.
> > > > Yingjie Fei, Zhuoran Yang, Yudong Chen, Zhaoran Wang, Qiaomin Xie. Risk-sensitive reinforcement learning: Near-optimal risk-sample tradeoff in regret. NeurIPS, 2020.
> > > > Yingjie Fei, Ruitu Xu. Cascaded Gaps: Towards Logarithmic Regret for Risk-Sensitive Reinforcement Learning. ICML, 2022.
> > > >
> > > >
> > > > **3. Minor Writing Issues and Related Work.**
> > > >
> > > > **3-1. Notation $w_{\pi,h}(s,a)$.**
> > > >
> > > > For any policy $\pi$, $h \in [H]$ and $(s,a) \in \mathcal{S} \times \mathcal{A}$, $w_{\pi,h}(s,a)$ denotes the probability of visiting $(s,a)$ at step $h$ under policy $\pi$.
> > > > We have added this definition in Section 4.1 of our revision.
> > > >
> > > > **3-2. Related Work [Keramati et al., 2020].**
> > > >
> > > > Thank you for pointing to this preference. We have cited and discussed [Keramati et al., 2020] in Section 2 and Appendix B of our revision.
> > > >
> > > > ---
> > > >
> > > > Reference:
> > > > Ramtin Keramati, Christoph Dann, Alex Tamkin, Emma Brunskill. Being optimistic to be conservative: Quickly learning a CVaR policy. AAAI, 2020.

---

> > > > > ### Author Response · Authors · 2022-11-15
> > > > > **Response to Reviewer U4MP - Part 5/5**
> > > > >
> > > > > Finally, we thank the reviewer again for the helpful comments for our work!
> > > > >
> > > > > We would like to highlight our contributions again.
> > > > >
> > > > >
> > > > > - We propose the first sample-efficient algorithms and  the first regret/sample complexity guarantees for Iterated CVaR RL and Worst Path RL (an interesting limiting case). Our results expand the theory of the widely-considered Iterated CVaR MDP problem [Ruszczyński, 2010; Chu \& Zhang, 2014; Bauerle \& Glauner, 2022] from the  known-transition setting to more practical unknown-transition (RL) model.
> > > > >
> > > > > - We develop several novel techniques for bounding the CVaR gap due to the value function shift and decomposing the regret via a distorted visitation distribution, which can find applications in other CVaR-based RL problems.
> > > > >
> > > > > - We also provide experiments to validate our theoretical bounds (Appendix A). The code and implementation instructions are uploaded to guarantee the reproducibility.
> > > > >
> > > > >
> > > > > We believe that our work makes solid technical contributions to the RL literature.
> > > > > This is also supported by other reviewers, e.g., **“This paper conducts a novel and comprehensive study”** by Reviewer zHz7, **“The results are interesting, new, and come with a clear presentation”** by Reviewer gBVT, and **“The paper studies an interesting problem, risk-sensitive RL, and provides comprehensive results”** by Reviewer rNfK.
> > > > >
> > > > >
> > > > >
> > > > > We believe that the reviewer's concerns can be effectively addressed by our included discussions and new lower bound construction, and the score 3 does not accurately reflect the quality of our work, as the reviewer also appreciates the novelty and thoroughness of our theoretical analysis.  If our response resolves your concerns at a satisfactory level, we would like to ask the reviewer to kindly consider raising the rating of our paper.
> > > > > Certainly, we are more than happy to answer any further questions you may have during the discussion period.
> > > > >
> > > > > ---
> > > > >
> > > > > References:
> > > > > Andrzej Ruszczyński. Risk-averse dynamic programming for Markov decision processes. Mathematical Programming, 2010.
> > > > > Shanyun Chu, Yi Zhang. Markov decision processes with iterated coherent risk measures. International Journal of Control, 2014.
> > > > > Nicole Bauerle, Alexander Glauner. Markov decision processes with recursive risk measures.
> > > > > European Journal of Operational Research, 2022.

---

> > > > > > ### Author Response · Authors · 2022-11-21
> > > > > > **Dear Reviewer U4MP, We Wonder If Your Concerns Are Resolved?**
> > > > > >
> > > > > > Dear Reviewer U4MP,
> > > > > >
> > > > > > We were wondering if you have gotten a chance to look over our responses and revisions and if your concerns are resolved.
> > > > > >
> > > > > > We believe that the score 3 does not accurately reflect the quality of our work, as you also appreciate the novelty and thoroughness of our theoretical analysis. If our response resolves your concerns at a satisfactory level, we would like to ask you to kindly consider raising the rating of our paper. In addition, we are more than happy to answer any further questions you may have during the discussion period.
> > > > > >
> > > > > > Thank you!
> > > > > >
> > > > > > Best regards,
> > > > > > Authors

---

> ### Author Response · Authors · 2022-11-26
> **Dear Reviewer U4MP, We Are Wondering If Your Concerns Are Addressed**
>
> Dear Reviewer U4MP,
>
> Since there are only two weeks left for the discussion period, we are wondering if you have gotten a chance to look over our responses and revisions and if your concerns are addressed. Certainly, we are more than happy to answer any further questions you may have during the discussion period.
>
> Best regards,
> Authors

---

> > ### Comment · Reviewer_U4MP · 2022-11-30
> > **Thanks for update!**
> >
> > I appreciate the author's effort in this very detailed explanation. It helped me understand more on the behavior of iterated CVaR (my first concern). So I'm raising my score.
> >
> > I still have one questions that I'd like to discuss regarding part 2.1 in your response. I didn't express my concern well in my initial review. It's not about the min w to be zero. It is about when we take the min over all possible (stochastic) policies, we can set the probability to be arbitrarily small but still positive.
> >
> >  - Say, there exist a state s1 and it is reachable from state s2. In s2, there exists action a2 going to s1 with 0 probability, and some other actions leading to a1. For the state s1 and any $\epsilon > 0$, there exists policy $\pi$ such that $0 < w(s_1) < \epsilon$, by setting $\pi(a1|s1) = 1-\epsilon$. In fact s1 can be a set of state, instead of a single state. It seems a structure like this exists in any simple example MDP such as tree, chain, grid-world.
> >
> >  - It seems less intuitive when $w_{\pi,h}(s)$, as a function of $(\pi,h,s)$, has values only in $\{0\} \cup [c, \infty)$ for some $c>0$.

---

> > > ### Author Response · Authors · 2022-12-01
> > > **Thank You for Raising Your Score! We Are Happy to Answer Your Question**
> > >
> > > Thank you very much for raising your score for our work!
> > >
> > > The reviewer's question may be effectively addressed by noting that the policies $\pi$ in the factor $\min_{\pi,h,s:\  w_{\pi,h}(s)>0} w_{\pi,h}(s)$ are **deterministic policies instead of stochastic policies**. Since there always exists a deterministic optimal policy for Iterated CVaR RL [Chu \& Zhang, 2014], in our paper, we consider deterministic policies rather than stochastic policies, i.e., we define a policy $\pi$ as a collection of $H$
> > > functions $\pi:=\\{\pi_h:\mathcal{S} \mapsto \mathcal{A}\\}_{h \in [H]}$ as done in the RL theory literature [Azar et al., 2017; Jin et al., 2018; Zanette \& Brunskill, 2019] (mentioned in Section 3).
> > >
> > > Therefore, for a given MDP, the factor $\min_{\pi,h,s:\  w_{\pi,h}(s)>0} w_{\pi,h}(s)$ **depends on the intrinsic transition distribution of this MDP**, which is fixed and **will not be arbitrarily small**. For example, in the MDP suggested by the reviewer, there are two states $s_1$ and $s_2$, and in state $s_2$, there are two actions $a_1$ and $a_2$ which can lead to $s_1$. In this case, the feasible policies we consider in state $s_2$ are $\pi(s_2)=a_1$ and $\pi(s_2)=a_2$. Then, the minimum visitation probability of $s_1$ is $w(s_1)=\min\{p(s_1|s_2,a_1),p(s_1|s_2,a_2)\}$, which depends on the intrinsic transition distribution of this MDP and will not be arbitrarily small.
> > >
> > > We will include more clarifications in our revision to avoid confusion. Please let us know if you have any further questions.
> > > Thank you very much again for your time in reviewing our paper!
> > >
> > > ---
> > >
> > > Reference:
> > > Shanyun Chu, Yi Zhang. Markov decision processes with iterated coherent risk measures. International Journal of Control, 2014.
> > > Mohammad Gheshlaghi Azar, Ian Osband, Remi Munos. Minimax regret bounds for reinforcement learning. ICML, 2017.
> > > Chi Jin, Zeyuan Allen-Zhu, Sebastien Bubeck, Michael I Jordan. Is Q-learning provably efficient? NeurIPS, 2018.
> > > Andrea Zanette, Emma Brunskill. Tighter problem-dependent regret bounds in reinforcement learning without domain knowledge using value function bounds. ICML, 2019.

---

> > > > ### Comment · Reviewer_U4MP · 2022-12-10
> > > > **Thanks for the update**
> > > >
> > > > Thank you for the update on the deterministic policy requirement in w. Therefore I agree it is more reasonable.

---

> > > > > ### Author Response · Authors · 2022-12-10
> > > > > **Thank You For Your Positive Comments!**
> > > > >
> > > > > Thank you very much for your positive comments and your time in reviewing our paper!

---

### Official Review · Reviewer_gBVT · 2022-10-28

**Confidence:** 3
**Correctness:** 4
**Technical Novelty And Significance:** 3
**Empirical Novelty And Significance:** 3
**Recommendation:** 8

**Clarity, Quality, Novelty And Reproducibility:**

Quality: The results are interesting, new, and come with a clear presentation.

Clarity: The structure of the paper is well organized. Related works have been discussed sufficiently. The novel technical part of the results is also well explained.

Originality: Both the results and the technical developments appear to be novel in the literature.

**Strength And Weaknesses:**

Strength:
1. I first find the technical results to be interesting. Especially the terms that have been singled out by the authors, e.g., the minimal visitation measure (Remark 1 in the paper). The authors have shown that this instance-dependent term not only appears in the regret upper bound, but also in the lower bound, demonstrating the importance of this term in CVaR RL problems.

2. I also appreciate the author's effort in interpreting the result after each main theorem, and the additional highlight in delineating the technical differences between solving the CVaR RL problems and standard RL problems. The proof overview is well-written and easy to follow.

Weakness:

There is one thing I recommend the authors to consider during revision. Namely, the value function and Q-function are defined in equations (i) and (ii), through recursion. Though I understand this might be concise and technically correct. This approach to some extent hides the true definition of the objective. Namely, how is it that this definition of value function reflects the goal of optimizing the tail? I find it hard to read the risk-averse nature of the problem simply from the DP equations.

Minor Question:

1. Looking into the author's explanation on the constant-regret of Worst Path RL, I was wondering is the technical analysis fundamentally different in the Worst Path RL than the CVaR case (when $\alpha > 0$)?


**Summary Of The Paper:**

This paper studies the RL problem with a CVAR objective, which concerns the tail of rewards instead of the expectation in standard RL. In the standard setting, the paper proposes two methods. The first method, named ICVaR-RM, focuses on regret minimization, and the paper also provides a matching lower bound. The second method, named ICVaR-BPI, focuses on returning an $\epsilon$-optimal policy. Finally, when the risk level $\alpha \to 0$, where the objective reduces to optimizing the worst-case cumulative reward, the authors propose MaxWP method that enjoys constant regret.

**Summary Of The Review:**

This paper studies the important problem of solving RL tasks with a risk-averse mindset. To this end, the authors propose new methods for solving CVaR RL problem, which performs efficient regret minimization (demonstrated by a matching upper and lower bound). The authors also additionally consider the limiting case, where the task is to optimize the worst-case cumulative reward, and a method with constant regret is proposed.

---

> ### Author Response · Authors · 2022-11-15
> **Response to Reviewer gBVT**
>
> Thank you very much for your valuable comments! According to your suggestions, we have included the expanded value function definitions and discussions in our revision.
>
> **1. Definitions of Value Functions.**
>
> Thank you for your insightful suggestions! The value function definition for Iterated CVaR RL, i.e., Eq. (i) in Section 3, can be expanded as follows. (Due to the formula display issue of OpenReview, here we use $s(h+1)$, $a(h+1)$ and $\pi(h+1)(\cdot)$ to denote $s_{h+1}$, $a_{h+1}$ and $\pi_{h+1}(\cdot)$, respectively, and the factor $(s_{h+1} \sim p(\cdot|s,a))$ in $CVaR^{\alpha}(s_{h+1} \sim p(\cdot|s,a))$ denotes the subscript of CVaR.)
>
>
> $$
> Q^{\pi}_h(s,a) = r(s,a) + CVaR^{\alpha}(s(h+1) \sim p(\cdot|s,a)) \bigg(r(s(h+1),\pi(h+1)(s(h+1))) + CVaR^{\alpha}(s(h+2) \sim p(\cdot|s(h+1),\pi(h+1)(s(h+1)))) \Big( \dots CVaR^{\alpha} (s_H \sim p(\cdot|s(H-1),\pi(H-1)(s(H-1)) )) (r(s_H,\pi_H(s_H) ) ) \Big) \bigg)
> $$
>
> $$
> V^{\pi}_h(s) = r(s, \pi_h(s)) + CVaR^{\alpha}(s(h+1) \sim p(\cdot|s,\pi_h(s))) \bigg(r(s(h+1),\pi(h+1)(s(h+1))) + CVaR^{\alpha}(s(h+2) \sim p(\cdot|s(h+1),\pi(h+1)(s(h+1)))) \Big(\dots CVaR^{\alpha}(s_H \sim p(\cdot|s(H-1),\pi(H-1)(s(H-1)))) (r(s_H,\pi_H(s_H))) \Big) \bigg)
> $$
>
> Similarly, the optimal value function definition, i.e., Eq. (ii) in Section 3, can be expanded as (here the factor $h$ in $Q^*h(s,a)$ denotes the subscript of the optimal Q-value function, and similar for that in the optimal V-value function)
>
> $$
>     Q^{*}h(s,a) = \max_{\pi} \Bigg\\{ r(s,a) + CVaR^{\alpha}(s(h+1) \sim p(\cdot|s,a)) \bigg(r(s(h+1),\pi(h+1)(s(h+1))) + CVaR^{\alpha}(s(h+2) \sim p(\cdot|s(h+1),\pi(h+1)(s(h+1)))) \Big( \dots CVaR^{\alpha} (s_H \sim p(\cdot|s(H-1),\pi(H-1)(s(H-1)) )) (r(s_H,\pi_H(s_H) ) ) \Big) \bigg)  \Bigg\\}
> $$
>
> $$
>     V^{*}h(s) = \max_{\pi} \Bigg\\{ r(s, \pi_h(s)) + CVaR^{\alpha}(s(h+1) \sim p(\cdot|s,\pi_h(s))) \bigg(r(s(h+1),\pi(h+1)(s(h+1))) + CVaR^{\alpha}(s(h+2) \sim p(\cdot|s(h+1),\pi(h+1)(s(h+1)))) \Big(\dots CVaR^{\alpha}(s_H \sim p(\cdot|s(H-1),\pi(H-1)(s(H-1)))) (r(s_H,\pi_H(s_H))) \Big) \bigg)  \Bigg\\}
> $$
>
>
> From the above definitions, we can see that, Iterated CVaR RL aims to maximize the worst $\alpha$-portion tail of the reward-to-go at each step (i.e., taking the CVaR operator on the reward-to-go at each step).
> In other words, Iterated CVaR RL wants to optimize the performance and avoid getting into bad situations  at each decision stage.
>
> We agree with the reviewer that including these definitions will make the problem formulation more clear. We have added the above definitions and discussions in Appendix C.1 of our revision (due to space limit), and mentioned and referred to them in the main text (Section 3).
>
>
>
> **2. Analysis of Worst Path RL.**
>
> The analysis for Worst Path RL ($\alpha \rightarrow 0$) is fundamentally different from that for Iterated CVaR RL ($\alpha>0$).
> This is due to the fact that the analysis for Iterated CVaR RL has a dependency on $\frac{1}{\alpha}$ in both upper and lower bounds. Thus, simply taking $\alpha \rightarrow 0$ there cannot obtain a tight constant regret for Worst Path RL.
>
> Instead, our analysis for Worst Path RL builds upon its unique $\min$ criterion in the Bellman equations.
> Under the $\min$ criterion, we show that the empirical Q-value function used in algorithm MaxWP enjoys several nice properties. For example, the Q-value function can only be over-estimated but not under-estimated (Lemma 21), the empirical Q-value function is non-increasing as the episodes proceed (Lemma 22), and algorithm MaxWP will not choose a suboptimal action  once it has an accurate estimation on state connectivity (Lemma 23). Building upon these properties, we adopt a novel contradiction argument to prove that, after a constant number of episodes, algorithm MaxWP always takes an optimal policy with high confidence. This constant regret analysis for Worst Path RL is new to the RL literature, and is very different from the Iterated CVaR analysis.

---

> > ### Comment · Reviewer_gBVT · 2022-11-21
> > **Thanks for clarifications**
> >
> > I thank the authors for responding in detail to my previous questions and suggestions. The responses are detailed and clear, and have sufficiently addressed my comments before.

---

> > > ### Author Response · Authors · 2022-11-21
> > > **Thank Reviewer gBVT**
> > >
> > > Thank you very much for your valuable suggestions and your time in reviewing our paper!

---

### Official Review · Reviewer_zHz7 · 2022-11-04

**Confidence:** 4
**Correctness:** 4
**Technical Novelty And Significance:** 3
**Empirical Novelty And Significance:** 3
**Recommendation:** 8

**Clarity, Quality, Novelty And Reproducibility:**

The paper is well-written and comprehensive. The extension from the traditional RL framework to CVaR RL is interesting.

**Strength And Weaknesses:**

Strength:
1. The paper studies the CVaR-based MDPs with the unknown transition.
2. Both upper and lower bounds results are provided.
3. The usage of concentration of conditional value at risk is interesting.

Weaknesses:
1. This paper does not use the Bernstein-type exploration bonuses. There could be potential improvements.
2. The lower bound results do not involve the size of state space, which seems loose.

I acknowledge the authors' response.

**Summary Of The Paper:**

This paper studies risk-sensitive reinforcement learning. The objective is to maximize the worst $\alpha$-percent tail of the reward-to-go at each step. This paper designs algorithms for regret minimization and best policy identification problems using the concentration of conditional value at risk. For the regret minimization problem, regret upper and lower bounds are provided; for the best policy identification problem, the sample complexity upper and lower bounds are provided. This paper also studies the limiting case when $\alpha$ is close to 0, i.e.the worst path RL.

**Summary Of The Review:**

This paper conducts a novel and comprehensive study for CVaR RL.

---

> ### Author Response · Authors · 2022-11-15
> **Response to Reviewer zHz7**
>
> Thank you very much for your time and efforts in reviewing our paper!
>
> **1. Bernstein-type Exploration Bonuses.**
>
> Thank you very much for suggesting this  interesting improvement direction!
> How to use the Bernstern-type exploration bonuses and analysis to improve the theoretical bounds is still an open problem in the risk-sensitive RL literature, e.g., [Tamar et al., 2015; Keramati et al., 2020; Fei et al., 2020; Fei et al., 2021], and will be a research topic of our future work.
>
>
>
>
> References:
> Aviv Tamar, Yonatan Glassner, Shie Mannor. Optimizing the CVaR via sampling. AAAI, 2015.
> Ramtin Keramati, Christoph Dann, Alex Tamkin, Emma Brunskill. Being optimistic to be conservative: Quickly learning a CVaR Policy. AAAI, 2020.
> Yingjie Fei, Zhuoran Yang, Yudong Chen, Zhaoran Wang, Qiaomin Xie. Risk-sensitive reinforcement learning: Near-optimal risk-sample tradeoff in regret. NeurIPS, 2020.
> Yingjie Fei, Zhuoran Yang, Yudong Chen, Zhaoran Wang. Exponential bellman equation and
> improved regret bounds for risk-sensitive reinforcement learning. NeurIPS, 2021.
>
>
> **2. Lower Bound and the Size of State Space.**
>
> Thank you for pointing out this problem!
> We believe that the factor $\min_{\pi,h,s:\  w_{\pi,h}(s)>0} w_{\pi,h}(s)$ in the lower bound may contain the information of the state space size $S$.
> In fact, existing risk-sensitive RL works, e.g., [Fei et al., 2020; Fei \& Xu, 2022], also do not have $S$ in their lower bounds.
> How to close this gap on $S$ is still an open problem in the risk-sensitive RL literature.
> We plan to further investigate this problem in our future work.
>
>
>
>
> References:
> Yingjie Fei, Zhuoran Yang, Yudong Chen, Zhaoran Wang, Qiaomin Xie. Risk-sensitive reinforcement learning: Near-optimal risk-sample tradeoff in regret. NeurIPS, 2020.
> Yingjie Fei, Ruitu Xu. Cascaded Gaps: Towards Logarithmic Regret for Risk-Sensitive Reinforcement Learning. ICML, 2022.

---

### Decision · Program_Chairs · 2023-01-20

**Decision:**

Accept: poster

**Justification For Why Not Higher Score:**

The main reason for not oral/spotlight is that the studied problem is relatively niche and is an incrementally different setting than what is previously studied in risk-sensitive RL, i.e. CVaR RL.

**Justification For Why Not Lower Score:**

The theoretical contribution is significant and there are new technical tools being develop that can be of independent interests.

**Metareview: Summary, Strengths And Weaknesses:**

This paper studies a new risk-sensitive RL setting called Iterated CVaR RL, where the goal is to maximize the $\alpha$-tail of the value function for each (s,a) pair instead of the expectation as in standard RL or the $\alpha$-tail of the overall value function in CVaR RL. Nearly tight upper and lower bounds are provided for both regret minimization and best policy identification in the tabular MDP setting.
There is a consensus among the reviewers that the paper is well-written and the technical contribution is solid. The authors have also provided an improved lower-bound construction in response to reviewer U4MP's comments. Overall, I believe this is a very solid paper and recommend acceptance.

**Note From Pc:**

if the above contains the word "oral" or "spotlight" please see: "oral" presentation means -> notable-top-5% and "spotlight" means -> notable-top-25%. As stated in our emails, we are disassociating presentation type from AC recommendations